# An optimal regulation of fluxes dictates microbial growth in and out of steady state

**Griffin Chure\*, Jonas Cremer\***

Department of Biology, Stanford University, Stanford, United States

**Abstract** Effective coordination of cellular processes is critical to ensure the competitive growth of microbial organisms. Pivotal to this coordination is the appropriate partitioning of cellular resources between protein synthesis via translation and the metabolism needed to sustain it. Here, we extend a low-dimensional allocation model to describe the dynamic regulation of this resource partitioning. At the core of this regulation is the optimal coordination of metabolic and translational fluxes, mechanistically achieved via the perception of charged- and uncharged-tRNA turnover. An extensive comparison with $\approx$ 60 data sets from *Escherichia coli* establishes this regulatory mechanism's biological veracity and demonstrates that a remarkably wide range of growth phenomena in and out of steady state can be predicted with quantitative accuracy. This predictive power, achieved with only a few biological parameters, cements the preeminent importance of optimal flux regulation across conditions and establishes low-dimensional allocation models as an ideal physiological framework to interrogate the dynamics of growth, competition, and adaptation in complex and ever-changing environments.

## Editor's evaluation

This valuable study provides a synthesis of sector models for cellular resource partitioning in microbes and shows how a simple flux balance model can quantitatively explain growth phenomena from numerous published experimental data sets. The evidence is convincing, and the study should be of interest to the microbial physiology community.

**\*For correspondence:**
gchure@stanford.edu (GC);
jbcremer@stanford.edu (JC)

**Competing interest:** The authors declare that no competing interests exist.

## Introduction

Growth and reproduction is central to life. This is particularly true of microbial organisms where the ability to quickly accumulate biomass is critical for competition in ecologically diverse habitats. Understanding which cellular processes are key in defining growth has thus become a fundamental goal in the field of microbiology. Pioneering physiological and metabolic studies throughout the 20th century laid the groundwork needed to answer this question (*Monod, 1935*; *Monod, 1937*; *Monod, 1941*; *Monod, 1947*; *Monod, 1966*; *Campbell, 1957*; *Schaechter et al., 1958*; *Kjeldgaard et al., 1958*; *Cooper and Helmstetter, 1968*; *Donachie et al., 1976*; *Jun et al., 2018*), with the extensive characterization of cellular composition across growth conditions at both the elemental (*Heldal et al., 1985*; *Loferer-Krößbacher et al., 1998*; *Lawford and Rousseau, 1996*) and molecular (*Schaechter et al., 1958*; *Kjeldgaard et al., 1958*; *Watson, 1976*; *Britten and Mcclure, 1962*) levels showing that the dry mass of microbial cells is primarily composed of proteins and RNA. Seminal studies further revealed that the cellular RNA content is strongly correlated with the growth rate (*Schaechter et al., 1958*; *Kjeldgaard et al., 1958*; *Gausing, 1977*), an observation which has held for many microbial species (*Karpinets et al., 2006*). As the majority of RNAs are ribosomal, these observations suggested that

protein synthesis via ribosomes is a major determinant of biomass accumulation in nutrient replete conditions (*Koch, 1988*; *Hernandez and Bremer, 1993*; *Magasanik et al., 1959*). Given that the cellular processes involved in biosynthesis, particularly those of protein synthesis, are well conserved between species and domains (*Doris et al., 2015*; *Davidovich et al., 2009*; *Bruell et al., 2008*), these findings have inspired hope that fundamental principles of microbial growth can be found despite the enormous diversity of microbial species and the variety of habitats they occupy.

The past decade has seen a flurry of experimental studies further establishing the importance of protein synthesis in defining growth. Approaches include modern '-omics' techniques with molecular-level resolution (*Taniguchi et al., 2010*; *Bennett et al., 2009*; *Schmidt et al., 2016*; *Valgepea et al., 2013*; *Peebo et al., 2015*; *Li et al., 2014*; *Balakrishnan et al., 2021b*; *Mori et al., 2021*; *Belliveau et al., 2021*; *Metzl-Raz et al., 2017*; *Paulo et al., 2015*; *Paulo et al., 2016*; *Xia et al., 2021*; *Jahn et al., 2018*), measurements of many core physiological processes and their coordination (*Dai et al., 2016*; *Basan et al., 2015*; *You et al., 2013*; *Wu et al., 2022*; *Di Bartolomeo et al., 2020*; *Li et al., 2018*; *Jahn et al., 2018*; *Zavřel et al., 2019*; *Parker et al., 2020*), and the perturbation of major cellular processes like translation (*Scott et al., 2010*; *Hui et al., 2015*; *Dai et al., 2016*; *Towbin et al., 2017*). Together, these studies advanced a more thorough description of how cells allocate their ribosomes to the synthesis of different proteins depending on their metabolic state and the environmental conditions they encounter, called *ribosomal allocation*. Tied to the experimental studies, different theoretical *ribosomal allocation models* have further been formulated to dissect how ribosomal allocation influences growth (*Molenaar et al., 2009*; *Karr et al., 2012*; *Scott et al., 2014*; *Weiße et al., 2015*; *Maitra and Dill, 2015*; *Giordano et al., 2016*; *Mori et al., 2017*; *Erickson et al., 2017*; *Towbin et al., 2017*; *Mori et al., 2017*; *Korem Kohanim et al., 2018*; *Macklin et al., 2020*; *Hu et al., 2020*; *Dourado and Lercher, 2020*; *Roy et al., 2021*; *Mori et al., 2021*; *Serbanescu et al., 2020*; *Balakrishnan et al., 2021a*; *Balakrishnan et al., 2021b*). For example, high-dimensional models have been formulated which simulate hundreds to thousands of biological reactions (*Karr et al., 2012*; *Macklin et al., 2020*) providing a detailed view of the emergence of distinct internal physiological states and the underlying processes which sustain them. Alternatively, other theoretical considerations follow coarse-grained approaches of moderate dimensionality which group different classes metabolic reactions together and mathematizicing their dynamics (*Roy et al., 2021*; *Hu et al., 2020*). Distinct from these is an array of extremely low-dimensional models, pioneered by *Molenaar et al., 2009*, which have been developed to describe growth phenomena in varied conditions and physiological limits that rely on only a few parameters (*Molenaar et al., 2009*; *Scott et al., 2014*; *Bosdriesz et al., 2015*; *Giordano et al., 2016*; *Towbin et al., 2017*; *Korem Kohanim et al., 2018*; *Erickson et al., 2017*; *Mairet et al., 2021*; *Balakrishnan et al., 2021a*) (a more detailed overview of the different modeling approaches is provided in Appendix 1 - Allocation models to study microbial growth).

In this work, we build on low-dimensional allocation models (*Scott et al., 2014*; *Giordano et al., 2016*; *Bosdriesz et al., 2015*; *Dourado and Lercher, 2020*; *Hu et al., 2020*) and the results from dozens of experimental studies to synthesize a self-consistent and quantitatively predictive description of resource allocation and growth. At the core of our model is the dynamic reallocation of resources between the translational and metabolic machinery, which is sensitive to the metabolic state of the cell. We demonstrate how 'optimal allocation'—meaning an allocation towards ribosomes which contextually maximizes the steady-state growth rate—emerges when the flux of amino acids through translation to generate new proteins and the flux of uncharged-tRNA through metabolism to provide charged-tRNA required for translation are mutually maximized, given the environmental conditions and corresponding physiological constraints. This regulatory scheme, which we term *flux-parity regulation*, can be mechanistically achieved by a global regulator (e.g., guanosine tetraphosphate, ppGpp, in bacteria) capable of simultaneously measuring the turnover of charged- and uncharged-tRNA pools and routing protein synthesis. The explanatory power of the flux-parity regulation circuit is confirmed by extensive comparison of model predictions with ≈ 60 data sets from *Escherichia coli*, spanning more than half a century of studies using varied methodologies. This comparison demonstrates that a simple argument of flux-sensitive regulation is sufficient to predict bacterial growth phenomena in and out of steady state and across diverse physiological perturbations. The accuracy of the predictions, coupled with the minimalism of the model, establishes the optimal regulation and cements the centrality of protein synthesis in defining microbial growth. The mechanistic nature of the theory—predicated on

a minimal set of biologically meaningful parameters—provides a low-dimensional framework that can be used to explore complex phenomena at the intersection of physiology, ecology, and evolution without requiring extensive characterization of the myriad biochemical processes which drive them.

## A simple allocation model describes translation-limited growth

We begin by formulating a simplified model of growth which follows the flow of mass from nutrients in the environment to biomass by building upon and extending the general logic of low-dimensional resource allocation models (*Molenaar et al., 2009*; *Scott et al., 2010*; *Scott et al., 2014*; *Dai et al., 2016*; *Giordano et al., 2016*). Specifically, we focus on the accumulation of *protein* biomass, as protein constitutes the majority of microbial dry mass (*Churchward et al., 1982*; *Feijó Delgado et al., 2013*) and peptide bond formation commonly accounts for ≈80% of the cellular energy budget (*Stouthamer, 1973*; *Belliveau et al., 2021*). Furthermore, low-dimensional allocation models utilize a simplified representation of the proteome where proteins can be categorized into only a few functional classes (*Molenaar et al., 2009*; *Scott et al., 2014*; *Hui et al., 2015*; *Maitra and Dill, 2015*; *Dourado and Lercher, 2020*). In this work, we consider proteins to be either ribosomal (i.e., a structural component of the ribosome, excluding ternary complex members like EF-Tu), metabolic (i.e., enzymes catalyzing synthesis of charged-tRNA molecules from environmental nutrients), or being involved in all other biological processes (e.g., lipid synthesis, DNA replication, energy generation, and chemotaxis) *Molenaar et al., 2009*; *Scott et al., 2010*; *Scott et al., 2014*; *Hui et al., 2015*; *Figure 1—figure supplement 1*; in Appendix 1 What makes the fraction of 'other' proteins?, we outline in more detail how individual protein species are partitioned between the 'metabolic' and 'other' sectors depending on their functional annotations. Simple allocation models further do not distinguish between different cells but only consider the overall turnover of nutrients and biomass. To this end, we explicitly consider a well-mixed batch culture growth as reference scenario where the nutrients are considered to be in abundance. This low-dimensional view of living matter may at first seem like an unfair approximation, ignoring the decades of work interrogating the multitudinous biochemical and biophysical processes of cell-homeostasis and growth (*Macklin et al., 2020*; *Karr et al., 2012*; *Hui et al., 2015*; *Grigaitis et al., 2021*; *Noree et al., 2019*). However, at least in nutrient replete conditions, many of these processes appear not to impose a fundamental limit on the rate of growth in the manner that protein synthesis does (*Belliveau et al., 2021*). In Appendix 1 The major simplifications of low-dimensional allocation models and why they might work we discuss this along with other simplifications in more detail.

To understand protein synthesis and biomass growth within the low-dimensional allocation framework, consider the flux diagram (*Figure 1A*, *Molenaar et al., 2009*; *Giordano et al., 2016*; *Belliveau et al., 2021*; *Balakrishnan et al., 2021b*; *Scott et al., 2014*) showing the masses of the three protein classes, precursors which are required for protein synthesis (including charged-tRNA molecules, free amino acids, cofactors, etc.), nutrients which are required for the synthesis of precursors, and the corresponding fluxes through the key biochemical processes (arrows). This diagram emphasizes that growth is autocatalytic in that the synthesis of ribosomes is undertaken by ribosomes which imposes a strict speed limit on growth (*Dill et al., 2011*; *Belliveau et al., 2021*; *Kafri et al., 2016*). While this may imply that the rate of growth monotonically increases with increasing ribosome abundance, it is important to remember that metabolic proteins are needed to supply the ribosomes with the precursors needed to form peptide bonds. Herein lies the crux of ribosomal allocation models: the abundance of ribosomes is constrained by the need to synthesize other proteins and growth is a result of how new protein synthesis is partitioned between ribosomal, metabolic, and other proteins. How is this partitioning determined, and how does it affect growth?

To answer these questions, we must understand how these different fluxes interact at a quantitative level and thus must mathematize the biology underlying the boxes and arrows in *Figure 1A*. Taking inspiration from previous models of allocation (*Molenaar et al., 2009*; *Scott et al., 2010*; *Scott et al., 2014*; *Giordano et al., 2016*; *Dourado and Lercher, 2020*), we enumerate a minimal set of coupled differential equations which captures the flow of mass through metabolism and translation (*Figure 1B*, with the dimensions and value ranges of the parameters listed in *Figure 1C* and *Supplementary file 1*). While we present a step-by-step introduction of this model in 'Methods,' we here focus on a summary of the underlying biological intuition and implications of the approach.

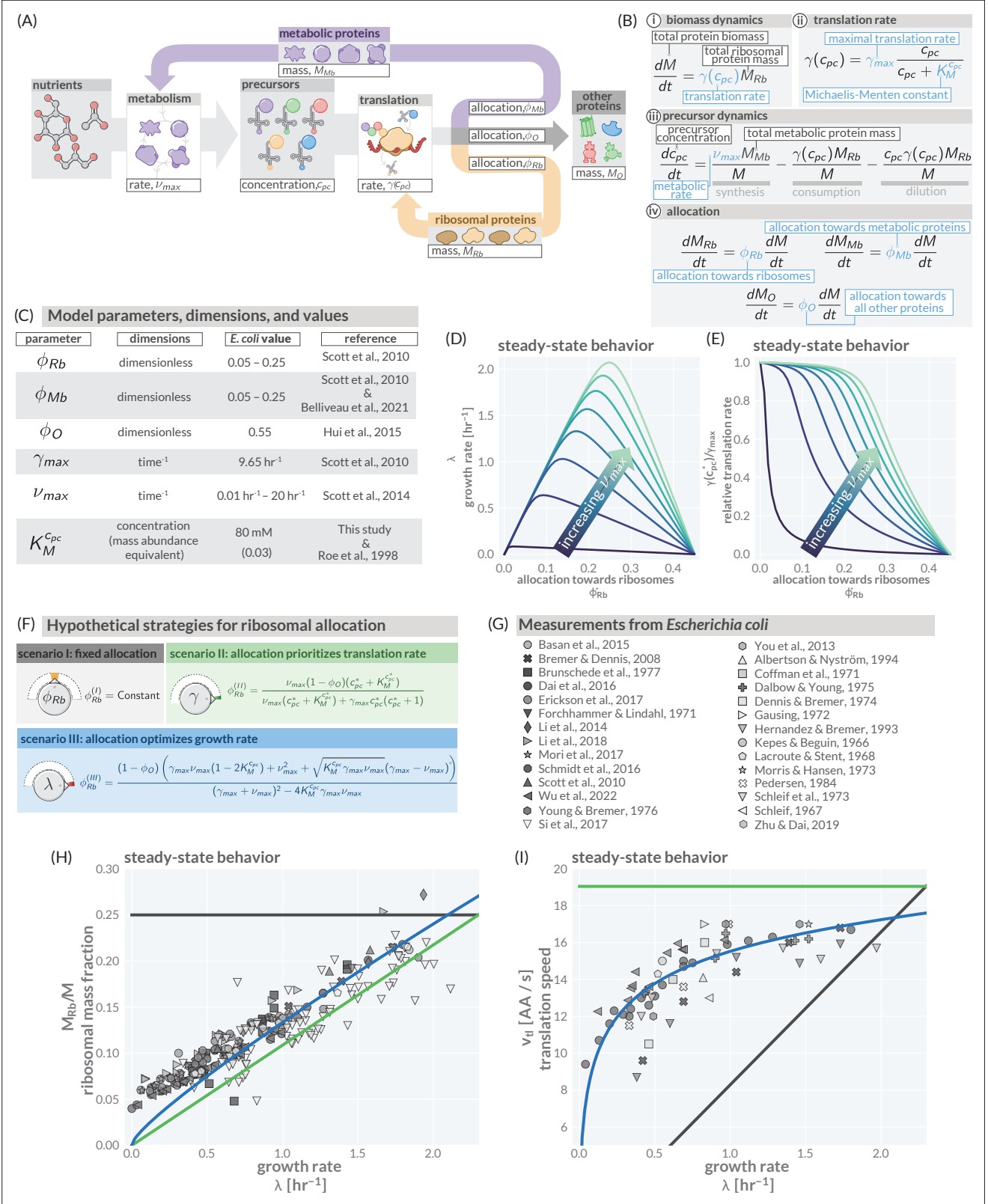

**Figure 1.** A simple model of ribosomal allocation and hypothetical regulatory strategies. (**A**) The flow of mass through the self-replicating system. Biomolecules and biosynthetic processes are shown as gray and white boxes, respectively. Nutrients in the environment passed through cellular metabolism to produce 'precursor' molecules which are then consumed through the process of translation to produce new protein biomass, either as metabolic proteins (purple arrow), ribosomal proteins (gold arrow), or 'other' proteins (gray arrow). (**B**) Annotated equations of the model with key

*Figure 1 continued on next page*

*Figure 1 continued*

parameters highlighted in blue. An interactive figure where these equations can be numerically integrated is provided on paper website (cremerlab. github.io/flux_parity). (**C**) Key model parameters, their units, typical values in *E. coli*, and their appropriate references. This is also provided as **Supplementary file 1**. The steady-state values of (**D**) the growth rate $\lambda$ and (**E**) the relative translation rate $\gamma(c_{pc}^*)/\gamma_{max}$, are plotted as functions of the allocation towards ribosomes for different metabolic rates (colored lines). (**F**) Analytical solutions for candidate scenarios for regulation of ribosomal allocation with fixed allocation, allocation to prioritize translation rate, and allocation to optimal growth rate highlighted in gray, green, and blue respectively. (**G**) A list of collated data sets of *E. coli* ribosomal allocation and translation speed measurements spanning 55 years of research. Details regarding these sources and method of data collation is provided in **Supplementary file 2**. A comparison of the observations with predicted growth-rate dependence of ribosomal allocation (**H**) and translation speeds (**I**) for the three allocation strategies. An interactive version of the panels allowing the free adjustment of parameters is available on the associated paper website (cremerlab.github.io/flux_parity).

The online version of this article includes the following source data and figure supplement(s) for figure 1:

**Source data 1.** Collated measurements of ribosomal mass fractions in *E. coli*.

**Source data 2.** Collated measurements of translation speeds per ribosome in *E. coli*.

**Figure supplement 1.** Coarse-grained description of biomass and the proteome.

**Figure supplement 2.** Precursor synthesis and growth when nutrients are not saturating.

**Figure supplement 3.** Modeling predictions of steady growth behavior.

**Figure supplement 4.** Three different allocation scenarios.

---

We begin by codifying the assertion that protein synthesis is key in determining growth. The synthesis of new total protein mass $M$ depends on the total proteinaceous mass of ribosomes $M_{Rb}$ present in the system and their corresponding average translation rate $\gamma$ (**Figure 1Bi**). As ribosomes rely on precursors to work, it is reasonable to assert that this translation rate must be dependent on the concentration of precursors $c_{pc}$ such that $\gamma \equiv \gamma(c_{pc})$ (**Scott et al., 2014**; **Giordano et al., 2016**), for which a simple Michaelis–Menten relation is biochemically well motivated (**Figure 1Bii**). With changing precursor concentrations, the translation rate $\gamma$ varies between a maximum value $\gamma_{max}$, representing rapid synthesis, and a minimum value $\gamma_{min}$, representing the slowest achievable translation rate. In our model, this minimum rate $\gamma_{min}$ is zero and corresponds to the condition where there are no available precursors to support translation. The standing precursor concentration $c_{pc}$ is set by a combination of processes (**Figure 1Biii**), namely the production of new precursors through metabolism (synthesis), their degradation through translation (consumption), and their dilution as the total cell volume grows. The synthesis is driven by the abundance of metabolic proteins $M_{Mb}$ in the system and the speed by which they convert nutrients into novel precursors. As the metabolic networks at play are complex, low-dimensional allocation models describe the process of metabolism using an average metabolic rate $\nu$ in lieu of mathematicizing the network's individual components. As such, the metabolic rate is difficult to directly measure but generally depends on the quality and concentration of nutrients in the environment (see below, **Figure 1—figure supplement 2** and 'Methods'). In the following, we focus on a growth regime in which nutrient concentrations are saturating. In such a scenario, metabolism operates at a nutrient-specific maximal metabolic rate $\nu \equiv \nu_{max}$. Finally, the relative magnitude of the ribosomal, metabolic, and 'other' protein masses is dictated by $\phi_{Rb}$, $\phi_{Mb}$, and $\phi_O$, three *allocation parameters* which range between zero and one to describe the fraction of ribosomes being utilized in synthesizing the corresponding protein pools. Importantly, as ribosomes only translate one protein at a time, the allocation parameters follow the constraint $\phi_{Rb} + \phi_{Mb} + \phi_O = 1$ (**Figure 1Biv**). For readers familiar with allocation models, we emphasize that we here use $\phi_X$ to denote allocation parameters rather than mass fractions, $M_X/M$; both quantities are only equivalent in the steady-state regime. Together, the introduced equations provide a full mathematicization of the mass flow diagram shown in **Figure 1A**.

For constant allocation parameters $(\phi_{Rb}^*, \phi_{Mb}^*)$, a steady-state regime emerges from this system of differential equation. Particularly, the precursor concentration is stationary in time ($c_{pc} = c_{pc}^*$), meaning the rate of synthesis is exactly equal to the rate of consumption and dilution. Furthermore, the translation rate $\gamma(c_{pc}^*)$ is constant during steady-state growth and the mass abundances of ribosomes and metabolic proteins are equivalent to the corresponding allocation parameters, e.g. $\frac{M_{Rb}}{M} \equiv \phi_{Rb}^*$. As a consequence, biomass is increasing exponentially $\frac{dM}{dt} = \lambda M$, with the growth rate $\lambda = \gamma(c_{pc}^*)\phi_{Rb}^*$. The emergence of a steady state and analytical solutions describing steady growth are further discussed in **Figure 1—figure supplement 2** and **Figure 1—figure supplement 3**. Notably, dilution is important

to obtain a steady state as has been highlighted previously by *Giordano et al., 2016* and *Dourado and Lercher, 2020* but is often neglected (Appendix Precursors concentrations and the importance of dilution by cell growth).

*Figure 1D and E* show how the steady-state growth rate $\lambda$ and translation rate $\gamma(c_{pc}^*)$ are dependent on the allocation towards ribosomes $\phi_{Rb}^*$. The figures also show the dependence on the metabolic rate $\nu_{max}$ which we here assert to be a proxy for the 'quality' of the nutrients in the environment (with increasing $\nu_{max}$, less metabolic proteins are required to obtain the same synthesis of precursors). The non-monotonic dependence of the steady-state growth rate on the ribosome allocation and the metabolic rate poses a critical question: What biological mechanisms determine the allocation towards ribosomes in a particular environment and what criteria must be met for the allocation to ensure efficient growth?

## Different strategies for regulation of allocation predicts different phenomenological behavior

While cells might employ many different ways to regulate allocation, we here consider three specific allocation scenarios to illustrate the importance of allocation on growth. These candidate scenarios either strictly maintain the total ribosomal content (scenario I), maintain a high rate of translation (scenario II), or optimize the steady-state growth rate (scenario III). We derive analytical solutions for these scenarios (as has been previously performed for scenario III; *Giordano et al., 2016*; *Dourado and Lercher, 2020*; *Figure 1F* and 'Methods'), and ultimately compare these predictions to observations with *E. coli* to show this organisms' optimal allocation of resources.

The simplest and perhaps most näive regulatory scenario is one in which the allocation towards ribosomes is completely fixed and independent of the environmental conditions. This strategy (scenario I in *Figure 1F*, gray) represents a locked-in physiological state where a specific constant fraction of all proteins is ribosomal. This imposes a strict speed limit for growth when all ribosomes are translating close to their maximal rate, $\gamma(c_{pc}^*) \approx \gamma_{max}$. If the fixed allocation is low (e.g., $\phi_{Rb}^{(I)} = 0.2$), then this speed limit could be reached at moderate metabolic rates.

A more complex regulatory scenario is one in which the allocation towards ribosomes is adjusted to prioritize the translation rate. This strategy (scenario II in *Figure 1F*, green) requires that the ribosomal allocation is adjusted such that a constant internal concentration of precursors $c_{pc}^*$ is maintained across environmental conditions, irrespective of the metabolic rate. In the case where this standing precursor concentration is large ($c_{pc}^* \gg K_M^{c_{pc}}$), all ribosomes will be translating close to their maximal rate.

The third and final regulatory scenario is one in which the allocation towards ribosomes is adjusted such that the steady-state growth rate is maximized. The analytical solution which describes this scenario (scenario III in *Figure 1F*) resembles previous analytical solutions by *Giordano et al., 2016*; *Dourado and Lercher, 2020*. More illustratively, the strategy can be thought of as one in which the allocation towards ribosomes is tuned across conditions such that the observed growth rate rests at the peak of the curves in *Figure 1D*. Notably, this does not imply that the translation rate is constantly high across conditions (as in scenario II). Rather, the translation rate is also adjusted and approaches its maximal value $\gamma_{max}$ only in very rich conditions (high metabolic rates). All allocation scenarios and their consequence on growth are discussed in further detail in *Figure 1—figure supplement 4* and the corresponding interactive figure on the paper website (cremerlab.github.io/flux_parity).

## *E. coli* regulates its ribosome content to optimize growth

Thus far, our modeling of microbial growth has remained 'organism agnostic' without pinning parameters to the specifics of any one microbe's physiology. To probe the predictive power of this simple allocation model and test the plausibility of the three different strategies for regulation of ribosomal allocation, we performed a systematic and comprehensive survey of data from a vast array quantitative studies of the well-characterized bacterium *E. coli*. This analysis, consisting of 26 studies spanning 55 years of research (listed in *Supplementary file 2* and as *Figure 1—source data 1* and *Figure 1—source data 2*) using varied experimental methods, goes well beyond previous attempts to compare allocation models to data (*Scott et al., 2010*; *Hui et al., 2015*; *Erickson et al., 2017*; *Giordano et al., 2016*; *Bosdriesz et al., 2015*; *Hu et al., 2020*; *Dourado and Lercher, 2020*; *Serbanescu et al., 2020*; *Hu et al., 2020*; *Roy et al., 2021*; *Maitra and Dill, 2015*; *Weiße et al., 2015*).

These data, shown in *Figure 1H and I* (markers), present a highly consistent view of *E. coli* physiology where the allocation towards ribosomes (equivalent to ribosomal mass fraction in steady-state balanced growth) and the translation rate demonstrate a strong dependence on the steady-state growth rate in different carbon sources. The pronounced correlation between the allocation towards ribosomes and the steady-state growth rate immediately rules out scenario I, where allocation is constant, as a plausible regulatory strategy used by *E. coli*, regardless of its precise value. Similarly, the presence of a dependence of the translation speed on the growth rate rules out scenario II, where the translation rate is prioritized across growth rates and maintained at a constant value. The observed phenomenology for both the ribosomal allocation *and* the translation speed is only consistent with the logic of regulatory scenario III where the allocation towards ribosomes is tuned to optimize growth rate.

This logic is quantitatively confirmed when we compute the predicted dependencies of these quantities on the steady-state growth rate for the three scenarios diagrammed in *Figure 1F* based on literature values for key parameters (outlined in *Supplementary file 1*). Deviations from the prediction for scenario III are only evident for the ribosomal content at very slow steady growth ($\lambda \leq 0.5$ hr$^{-1}$), which are hardly observed in any ecologically relevant conditions and can be attributed to additional biological and experimental factors, including protein degradation (*Calabrese et al., 2021*) and cultures which have not yet reached steady state, factors we discuss in Appendix 1 – Additional considerations relevant at slow growth. The inactivation of ribosomes is another such explanation, though a growth rate-independent inactive fraction is not sufficient to explain the observations, Appendix 1 —Inactive ribosomes.

Importantly, the agreement between theory and observations works with a minimal number of parameters and does not require the inclusion of fitting parameters. All fixed model parameters such as the maximum translation rate $\gamma_{max}$ and the Michaelis–Menten constant for translation $K_M^{c_{pc}}$ have distinct biological meaning and can be either directly measured or inferred from data (*Supplementary file 1*). Furthermore, we discuss the necessity of other parameters such as the 'other protein sector' $\phi_O$ (Appendix 1— What makes the fraction of 'other' proteins?), its degeneracy with the maximum metabolic rate $\nu_{max}$, and inclusion of ribosome inactivation and minimal ribosome content (Appendix 1— Inactive ribosomes). We, furthermore, provide an interactive figure on the paper website (cremerlab.github.io/flux_parity) where the parametric sensitivity of these regulatory scenarios and the agreement/disagreement with data can be directly explored. Notably there is no combination of parameter values that would allow scenario I or II to adequately describe both the ribosomal allocation and the translation speed as a function of growth rate. These findings are in line with a recent higher-dimensional modeling study (*Hu et al., 2020*), which, based on the optimization of a reaction network with >200 components, rationalized the variation in translation speed with growth as a manifestation of efficient protein synthesis. Together, these results confirm that scenario III can accurately describe observations over a very broad range of conditions, in strong support of the popular but often questioned presumption that *E. coli* optimally tunes its ribosomal content to promote fast growth (*Giordano et al., 2016*; *Bosdriesz et al., 2015*; *Towbin et al., 2017*).

In Appendix 1 – Application of the model to Saccharomyces cerevisiae, we present a similar analysis for yeast, which, in line with previous studies (*Metzl-Raz et al., 2017*; *Xia et al., 2021*; *Paulo et al., 2015*; *Paulo et al., 2016*; *Kostinski and Reuveni, 2021*), suggests that this eukaryote likely follows a similar optimal allocation strategy, although data for ribosomal content and the translation rate is scarce. The strong correlation between ribosome content and growth rate has further been reported for other microbial organisms in line with an optimal allocation (*Karpinets et al., 2006*; *Jahn et al., 2018*; *Zavřel et al., 2019*; *Jahn et al., 2021*), though the absence of translation rate measurements precludes confirmation. An interesting exception is the methanogenic archaeon *Methanococcus maripaludis*, which appears to maintain constant allocation, in agreement with scenario I (*Müller et al., 2021*). The presented analysis thus suggests that *E. coli* and possibly many other microbes closely follow an optimal ribosome allocation behavior to support efficient growth. Moreover, the good agreement between experiments and data establishes that a simple low-dimensional allocation model can describe growth with notable quantitative accuracy. However, this begs the question: how do cells coordinate their complex machinery to ensure optimal allocation?

## Optimal allocation results from a mutual maximization of translational and metabolic flux

To optimize the steady-state growth rate, cells must have some means of coordinating the flow of mass through metabolism and protein synthesis. In the ribosomal allocation model, this reduces to a regulatory mechanism in which the allocation parameters ($\phi_{Rb}$ and $\phi_{Mb}$) are dynamically adjusted such that the metabolic flux to provide new precursors ($\nu(c_{nt})\phi_{Mb}$) and translational flux to make new proteins ($\gamma(c_{pc}^*)\phi_{Rb}$, equivalent to the steady-state growth rate $\lambda$) are not only equal, but are mutually maximized. Such regulation therefore requires a mechanism by which both the metabolic and translational flux can be simultaneously sensed.

Thus far, we have referred to the end product of metabolism as ambiguous 'precursors' which are used by ribosomes to create new proteins. In reality, these precursors are tRNAs charged with their cognate amino acids. One can think of metabolism as a two-step process where (i) an amino

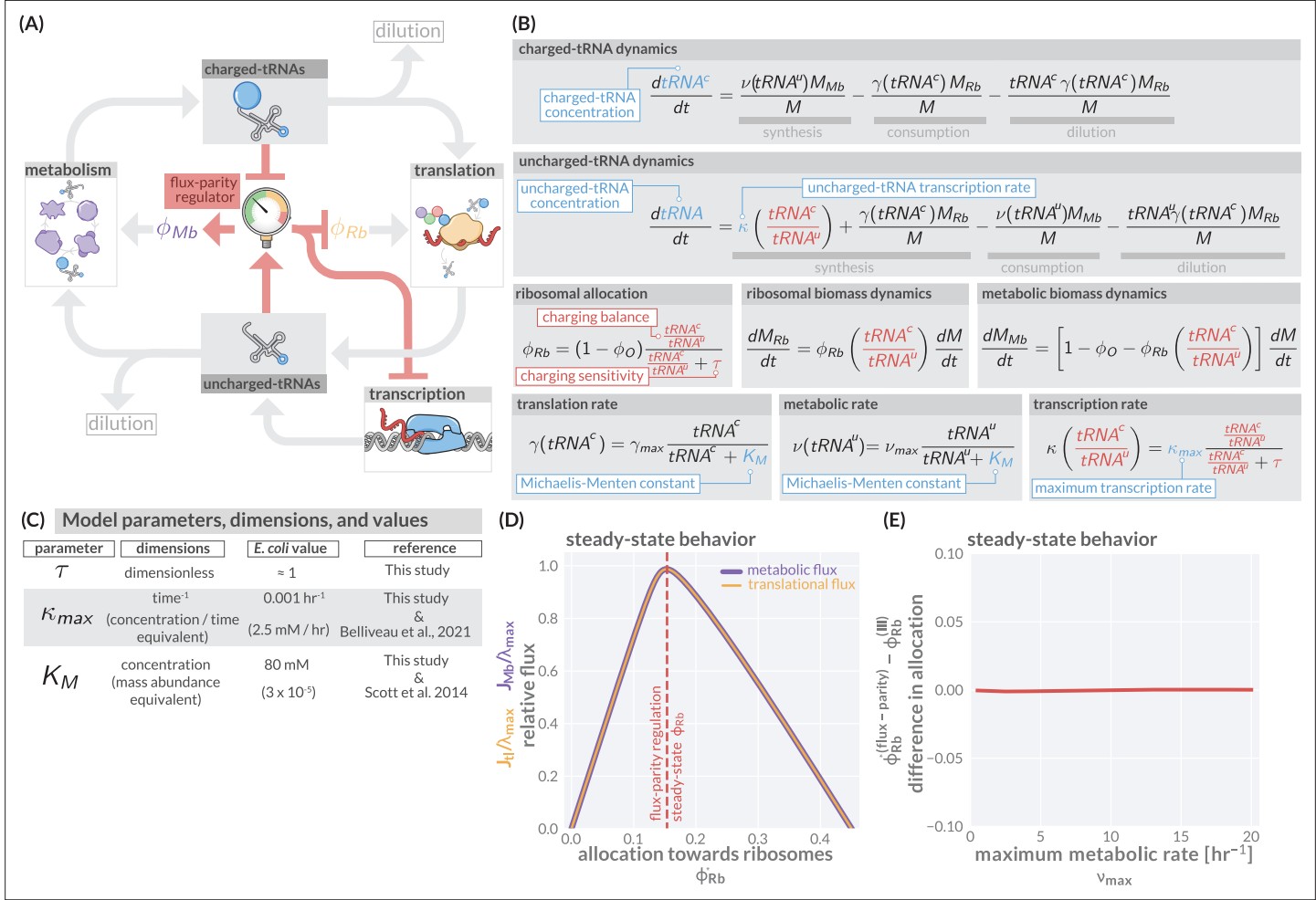

**Figure 2.** The regulation of ribosome allocation via a flux-sensing mechanism. (**A**) A circuit diagram of interactions between metabolic and translational fluxes with flux-parity regulatory connections highlighted in red. The fluxes are connected via a positive feedback loop through the generation of mutual starting materials (uncharged- or charged-tRNAs, respectively). The rates of each flux exhibit semi-autoregulatory behavior in that flux through each process reduces the standing pool of tRNAs. (**B**) The governing dynamics of the flux-parity regulatory circuit with key parameters highlighted in blue and flux-parity regulatory components highlighted in red. (**C**) Parameters, dimensions, values, and references for each component of the flux-parity regulatory circuit. (**D**) The steady-state meabolic (purple) and translational (gold) fluxes plotted as a function of the ribosomal allocation under the simple allocation model. Vertical red line indicates the steady-state solution of the flux-parity model under physiological parameter regimes. (**E**) The difference in allocation towards ribosomes in steady state between the flux-parity model and optimal allocation ($\phi_{Rb}^{*(flux-parity)} - \phi_{Rb}^{(III)}$) plotted as a function of the maximal metabolic rate, $\nu_{max}$.

The online version of this article includes the following figure supplement(s) for figure 2:

**Figure supplement 1.** Flux-parity directs allocation parameters towards an optimum.

acid is synthesized from environmental nutrients and (ii) an amino acid is attached to the appropriate uncharged-tRNA. As we assume that nutrients are in excess in the environment, we make the approximation that nutrients in the environment are saturating such that $c_{nt} \gg K_M^{c_{nt}}$ and the metabolic rate $\nu$ now depends solely on the concentration of uncharged-tRNAs $\nu(\text{tRNA}^u)$. This enforces some level of regulation of metabolism; if the uncharged tRNA concentration is too low, the rate of metabolism slows and does not add to the already large pool of charged tRNA. But when charged-tRNA is available, translation occurs at a rate $\gamma(\text{tRNA}^c)$, forming new protein biomass and converting a charged-tRNA back to an uncharged state. This process is shown by gray arrows in *Figure 2A*.

To describe the state-dependent adjustment of the allocation parameters ($\phi_{Rb}$ and $\phi_{Mb}$), we further include in this feedback loop a regulatory system we term a 'flux-parity regulator' (*Figure 2A*, red), which controls the allocation parameters in response to relative changes in the concentrations of the two tRNA species. Together, the arrows in *Figure 2* represent a more fine-grained view of a proteinaceous self replicating system, yet maintains much of the structural minimalism of the simple ribosomal allocation model without requiring explicit consideration of different types of amino acids (*Bosdriesz et al., 2015*), inclusion of their myriad synthesis pathways (*Hu et al., 2020*), or reliance on observed phenomenology (*Wu et al., 2022*).

The boxes and arrows of *Figure 2A* can be mathematized to arrive at a handful of ordinary differential equations (*Figure 2B*) structurally similar to those in *Figure 1B*. At the center of this model is the ansatz that the ribosomal allocation $\phi_{Rb}$ is dependent on the ratio of charged- and uncharged-tRNA pools and has the form

$$\phi_{Rb}\left(\frac{\text{tRNA}^c}{\text{tRNA}^u}\right) = (1 - \phi_O)\frac{\frac{\text{tRNA}^c}{\text{tRNA}^u}}{\frac{\text{tRNA}^c}{\text{tRNA}^u} + \tau}, \tag{1}$$

where the ratio $\frac{\text{tRNA}^c}{\text{tRNA}^u}$ represents the 'charging balance' of the tRNA and $\tau$ is a dimensionless sensitivity parameter which defines the charging balance at which the allocation towards ribosomes is half-maximal. Additionally, we make the assertion that the synthesis rate of new uncharged-tRNA via transcription $\kappa$ is coregulated with ribosomal proteins (*Skjold et al., 1973*; *Dong et al., 1996*) and has a similar form of

$$\kappa\left(\frac{\text{tRNA}^c}{\text{tRNA}^u}\right) = \kappa_{max}\frac{\frac{\text{tRNA}^c}{\text{tRNA}^u}}{\frac{\text{tRNA}^c}{\text{tRNA}^u} + \tau}, \tag{2}$$

where $\kappa_{max}$ represents the maximal rate of tRNA transcription relative to the total biomass.

Numerical integration of this system of equations reveals that the flux-parity regulation is capable of optimizing the allocation towards ribosomes, $\phi_{Rb}$, such that the metabolic and translation fluxes are mutually maximized (*Figure 2D*), thus achieving optimal allocation. Importantly, the optimal behavior inherent to this regulatory mechanism can be attained across a wide range of parameter values for the charging sensitivity $\tau$ and the transcription rate $\kappa_{max}$, the two key parameters of flux-parity regulation (*Figure 2C*). Moreover, the emergent optimal behavior of this regulatory scheme occurs across conditions without the need for any fine-tuning between the flux-parity parameters and other parameters. For example, the control of allocation via flux-parity regulation matches the optimal allocation (scenario III above) when varying the metabolic rate $\nu_{max}$ (*Figure 2E* and Appendix 1 – Parameter dependence of the flux-parity model).

The theoretical analysis presented in *Figure 2* suggests that a flux-parity regulatory mechanism may be a simple way to ensure optimal ribosomal allocation that is robust to variation in the key model parameters. To test if such a scheme may be implemented in *E. coli*, we compared the behavior of the steady-state flux-parity regulatory circuit within physiological parameter regimes to steady-state measurements of ribosomal allocation and the translation rate as a function of the growth rate (*Figure 3A and B*). Remarkably, the predicted steady-state behavior of the flux-parity regulatory circuit describes the observed data with the same quantitative accuracy as the optimal behavior defined by scenario III, as indicated by the overlapping red and blue lines, respectively.

While the flux-parity regulation scheme appears to accurately describe the behavior of *E. coli*, how are metabolic and translational fluxes sensed at a mechanistic level? Many bacteria, including *E. coli*, utilize the small molecule guanosine tetraphosphate (ppGpp) as a molecular indicator of amino acid limitation and has been experimentally shown to regulate ribosomal, metabolic, and tRNA genes through many routes, including directly binding RNA polymerase (*Magnusson et al., 2005*; *Anderson*

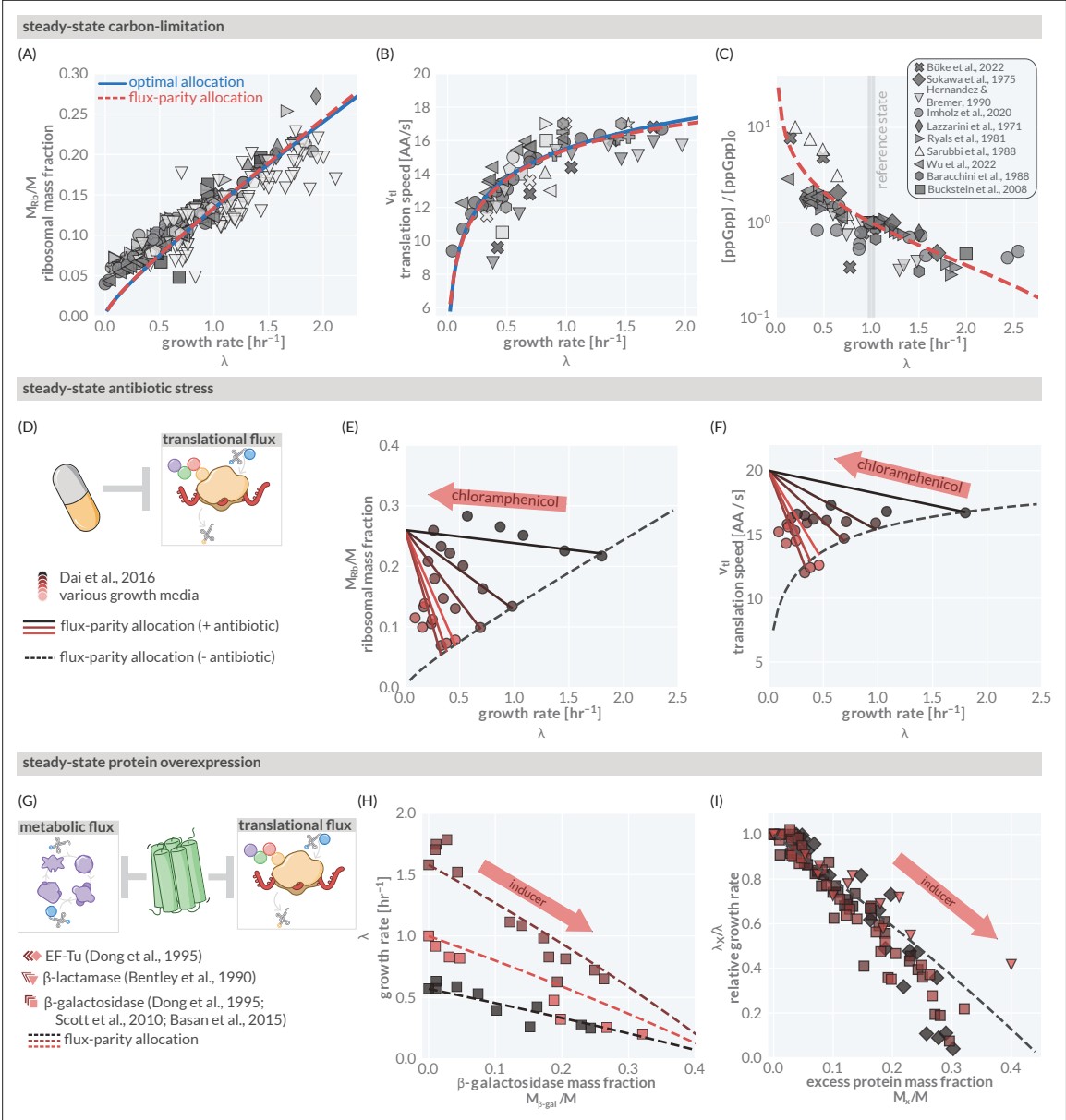

**Figure 3.** The predictive power of flux-parity regulation in steady state. Measurements of the (**A**) ribosomal allocation and the (**B**) translation rate are plotted alongside the steady-state behavior of the flux-parity regulatory circuit (red dashed line) and the optimal behavior of scenario III (solid blue line). Points and markers are the same as those shown in *Figure 1G*. (**C**) Measurements of intracellular ppGpp concentrations relative to a reference condition ($\lambda_0 \approx 1$ hr$^{-1}$) are plotted as a function of growth rate alongside the prediction emergent from the flux-parity regulatory circuit (red dashed line). (**D–F**) Inhibition of ribosome activity via antibiotic modeled repression of translational flux. Plots show comparison with data for different media (red shades) with the flux-parity model predictions (dashed lines). (**G–I**) Inhibition of metabolic and translational fluxes through excess gene expression. (**H**) shows data where β-galactosidase is expressed at different levels. Different shades of red correspond to different growth media. Right-hand panel shows collapse of the growth rates of overexpression of β-galactosidase (squares), β-lactamase (inverted triangles), and EF-Tu (diamonds) relative to the wild-type growth rate in different media conditions. The same set of model parameters listed in *Supplementary file 2* has been used to generate the predictions.

The online version of this article includes the following source data and figure supplement(s) for figure 3:

**Source data 1.** Collated measurements of relative ppGpp concentrations.

**Source data 2.** Collated measurements of excess protein mass fractions.

**Figure supplement 1.** Comparison of predictive capacity of flux-parity allocation between ppGpp scaling ansatzes.

*et al., 2021*; *Potrykus and Cashel, 2008*; *Potrykus et al., 2011*; *Imholz et al., 2020*) and plays an important role in other cellular processes, including cell size control (*Büke et al., 2022*). Mechanistically, ppGpp levels are enzymatically controlled depending on the metabolic state of the cell, with synthesis being triggered upon binding of an uncharged-tRNA into an actively translating ribosome. While many molecular details of this regulation remain unclear (*Magnusson et al., 2005*; *Anderson et al., 2021*; *Potrykus and Cashel, 2008*; *Wu et al., 2022*), the behavior of ppGpp meets all of the criteria of a flux-parity regulator. Rather than explicitly mathematicizing the biochemical dynamics of ppGpp synthesis and degradation, as has been undertaken previously (*Bosdriesz et al., 2015*; *Giordano et al., 2016*; *Wu et al., 2022*), we model the concentration of ppGpp being inversely proportional to the charging balance,

$$[\text{ppGpp}] \propto \frac{\text{tRNA}^u}{\text{tRNA}^c}, \tag{3}$$

encompassing the fact that processes beyond allocation use ppGpp as an effector molecule. This ratio, mathematically equivalent to the odds of a ribosome binding an uncharged-tRNA relative to binding a charged-tRNA, is one example of a biochemically motivated ansatz that can be considered ('Methods') and provides a relative measure of the metabolic and translational fluxes.

With this approach, the amount of ppGpp present at low growth rates, and therefore low ribosomal allocation, should be significantly larger than at fast growth rates where ribosomal allocation is larger and charged-tRNA are in abundant supply. While our model cannot make predictions of the *absolute* ppGpp concentration, we can compute the *relative* ppGpp concentration to a reference state $[\text{ppGpp}]_0$ as

$$\frac{[\text{ppGpp}]}{[\text{ppGpp}]_0} = \frac{\left(\text{tRNA}^u/\text{tRNA}^c\right)}{\left(\text{tRNA}_0^u/\text{tRNA}_0^c\right)}. \tag{4}$$

To test this, we compiled and rescaled ppGpp measurements of *E. coli* across a range of growth rates from various literature sources (*Figure 3C* and *Figure 3—source data 1*). The quantitative agreement between the scaling predicted by *Equation 4* and the experimental measurements strongly suggests that ppGpp assumes the role of a flux sensor and enforces optimal allocation through the discussed flux-parity mechanism.

## The flux-parity allocation model predicts *E. coli* growth behavior in and out of steady state

We find that the flux-parity allocation model is extremely versatile and allows us to quantitatively describe aspects of microbial growth in and out of steady state and under various physiological stresses and external perturbations with the same core set of parameters. Here, we demonstrate this versatility by comparing predictions to data for four particular examples using the same self-consistent set of parameters we have used thus far (*Supplementary file 1*). First, we examine the influence of translation-targeting antibiotics like chloramphenicol (*Figure 3D*) on steady-state growth in different growth media (*Scott et al., 2010*; *Dai et al., 2016*). By incorporating a mathematical description of ribosome inactivation via binding to chloramphenicol (described in 'Methods'), we find that the flux-parity allocation model quantitatively predicts the change in steady-state growth and ribosomal content with increasing chloramphenicol concentration (*Figure 3E*, red shades). Furthermore, the effect on the translation speed is qualitatively captured (*Figure 3F*, red shades). The ability of the flux-parity allocation model to describe these effects without readjustment of the model and its core parameters is notable and provides a mechanistic rationale for previously established phenomenological relations (*Scott et al., 2010*; *Dai et al., 2016*).

As a second perturbation, we consider the burden of excess protein synthesis by examining the expression of synthetic genes (*Figure 3G*). A decrease in growth rate results when cells are forced to synthesize different amounts of the lactose cleaving enzyme $\beta$-galactosidase in different media lacking lactose (*Figure 3H*, red shades). The flux-parity allocation model (dashed lines) quantitatively predicts the change in growth rate with the measured fraction of $\beta$-galactosidase without further fitting ('Methods'). The trends for different media (red shades) quantitatively collapse onto a single line (*Figure 3I* and *Figure 3—source data 2*) when comparing changes in relative growth rates, a relation which is also captured by the model (dashed black line) and is independent of the overexpressed protein (symbols). This collapse, whose functional form is derived in 'Methods,' demonstrates that

the flux-parity allocation model is able to describe excess protein synthesis in general, rather than at molecule- or media-specific level.

As the flux-parity regulatory circuit responds to changes in the metabolic and translational fluxes, it can be used to explore behavior in changing conditions. Consider a configuration where the starting conditions of a culture are tuned such that the ribosomal allocation $\phi_{Rb}$, the tRNA charging balance tRNA$^c$/tRNA$^u$, and the ribosome content $M_{Rb}/M$ are set to be above or below the appropriate level for steady-state growth in the environment (*Figure 4A*). As the culture grows, the observed ribosomal content $M_{Rb}/M$ is steadily adjusted until the steady-state level is met where it directly matches the optimal allocation (*Figure 4B*). This adaptation of the ribosomal content is controlled by dynamic adjustment of the allocation parameters via the flux-parity regulatory circuit (*Figure 4C*). To further test the flux-parity allocation model, we examine how accurately this system can predict growth behavior under nutritional shifts (*Figure 4D–F*) and the entry to starvation (*Figure 4G–I*).

We first consider a nutrient shift where externally supplied low-quality nutrients are instantaneously exchanged with rich nutrients. *Figure 4E* shows three examples of such nutritional upshifts (markers), all of which are well described by the flux-parity allocation theory (dashed lines). The precise values of the growth rates before, during, and after the shift will depend on the specific carbon sources involved. However, by relating the growth rates before and immediately after the shift to the total shift magnitude (as shown in *Korem Kohanim et al., 2018*), one can collapse a large collection of data onto a single curve (*Figure 4F*, markers). The collapse emerges naturally from the model (dashed line) when decomposing the metabolic sector into needed and non-needed components ('Methods'), demonstrating that the flux-parity allocation model is able to quantitatively describe nutritional upshifts at a fundamental level.

Finally, we consider the growth dynamics during the onset of starvation, another non-steady-state phenomenon (*Figure 4G–I*). *Figure 4H* shows the growth of batch cultures where glucose is provided as the sole carbon source in different limiting concentrations (*Bren et al., 2013*) (markers). The cessation of growth coincides with a rapid, ppGpp-mediated increase in expression of metabolic proteins (*Magnusson et al., 2005*; *Dennis et al., 2004*). *Bren et al., 2013* demonstrated that expression from a glucose-specific metabolic promoter (PtsG) rapidly, yet temporarily, increases with the peak occurring at the moment where growth abruptly stops (*Figure 4I*, solid gray lines). The flux-parity allocation model again predicts this behavior (*Figure 4I*, red lines) without additional fitting ('Methods'), cementing the ability of the model to describe growth far from steady state.

## Discussion

Microbial growth results from the orchestration of an astoundingly diverse set of biochemical reactions mediated by thousands of protein species. Despite this enormous complexity, experimental and theoretical studies alike have shown that many growth phenotypes can be captured by relatively simple correlations and models which incorporate only a handful of parameters (*Schaechter et al., 1958*; *Molenaar et al., 2009*; *Scott et al., 2010*; *Scott et al., 2014*; *Erickson et al., 2017*; *Korem Kohanim et al., 2018*; *Bosdriesz et al., 2015*; *Giordano et al., 2016*; *Dai et al., 2016*). Through re-examination of these works, we relax commonly invoked approximations and assumptions, include a generalized description of global regulation, and integrate an extensive comparison with data to establish a self-consistent, low-dimensional model of protein synthesis that is capable of quantitatively describing complex growth behaviors in and out of steady state.

Growth emerges as in previous allocation models (*Molenaar et al., 2009*; *Scott et al., 2010*; *Giordano et al., 2016*) as a consequence of protein synthesis and the allocation of ribosome activity towards (i) making new ribosomes, (ii) making the metabolic proteins which sustain the precursors ribosomes require to translate, and (iii) making other proteins cells require to operate. An *optimal allocation* which yields the fastest growth in a given condition is reached when the synthesis of precursors (metabolic flux) and the consumption of precursors (translational flux) are mutually maximized, a process we term *flux-parity regulation*. We analyze how such regulation can be mechanistically achieved by the relative sensing of charged- and uncharged-tRNA via the abundance of a global regulator (such as ppGpp) which diametrically affects the expression of ribosomal and metabolic genes. Through extensive comparison with 61 data sets from 46 studies, we show that the flux-parity model predicts the fundamental growth behavior of *E. coli* with quantitative accuracy. Beyond describing the growth-rate dependent ribosomal content and translation speed for steady growth across various

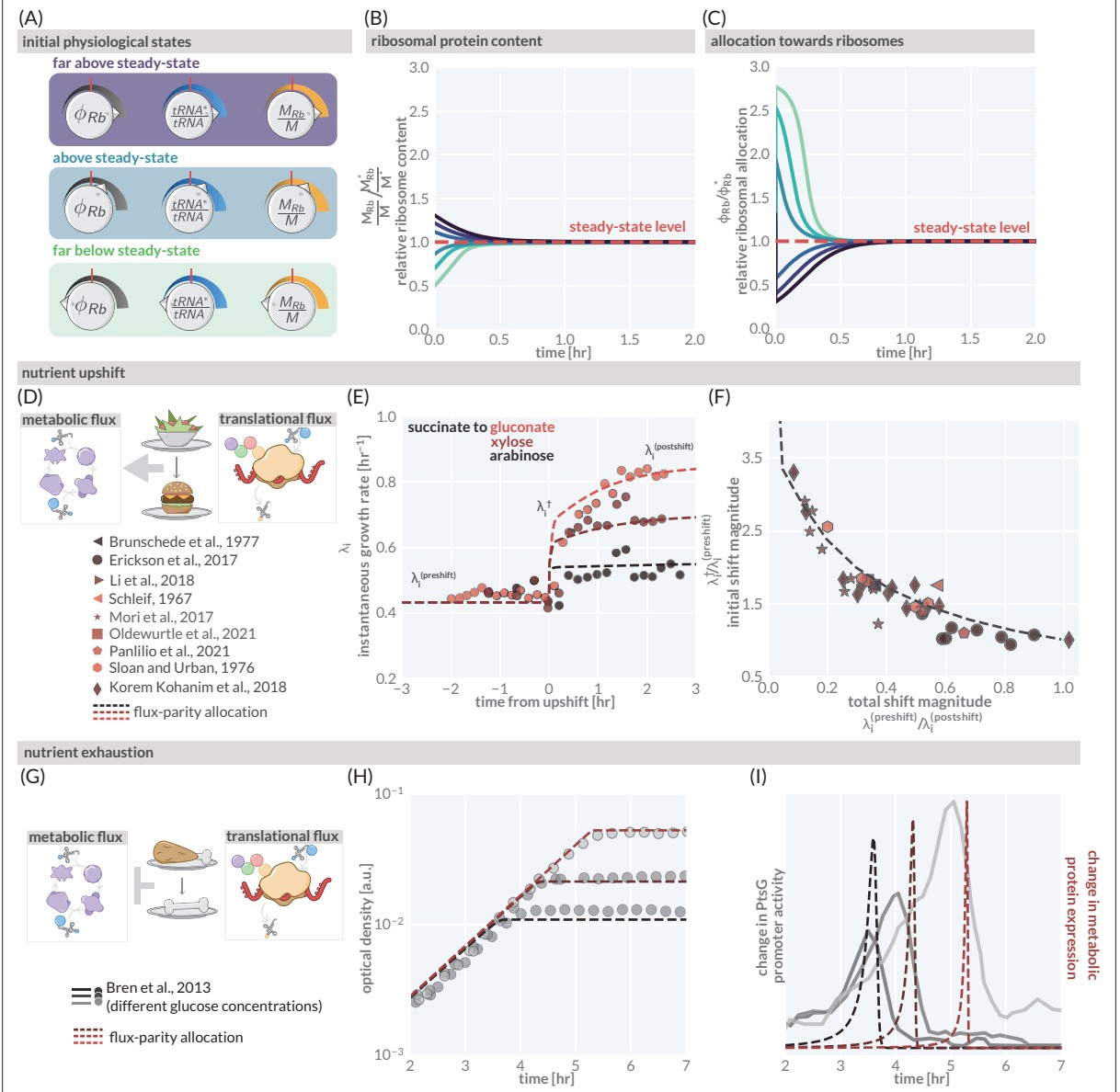

**Figure 4.** The predictive power of flux-parity regulation out of steady state. (**A**) Hypothetical initial configurations of model parameters and variables before begining numerical integration. (**B**) The equilibration of the ribosomal protein content ($M_{Rb}/M$). (**C**) Dynamic adjustment of the ribosomal allocation parameter in response to the new environment. Green and purple colored lines correspond to the initial conditions of the culture from well above to well below the steady-state values, respectively. Dashed red line indicates the steady-state solution. (**D, E**) Nutrient upshifts with increased metabolic flux. (**E**) The instantaneous growth rate $\lambda_i$ for shifts from succinate to gluconate (bright red), xylose (dark red), or arabinose (black) (**Erickson et al., 2017**). (**F**) Collapse of instantaneous growth rate measurements immediately after the shift (relative to the preshift-growth rate) as a function of the total shift magnitude. (**G–I**) Exhaustion of nutrients in the environment yields a decrease in the metabolic flux, promoting expression of more metabolic proteins. (**H**) Growth curve measurements in media with different starting concentrations of glucose (0.22 mM, 0.44 mM, and 1.1 mM glucose from light to dark, respectively) overlaid with flux-parity predictions. (**I**) The change in total metabolic protein synthesis in the flux-parity model (dashed lines) overlaid with the change in expression of a fluorescent reporter from a PtsG promoter (solid lines).

The online version of this article includes the following source data for figure 4:

**Source data 1.** Collated measurements of relative nutrient upshift magnitudes.

carbon sources, the flux-parity model quantitatively captures phenomena out of steady state (including nutrient upshifts and response to starvation) and under externally applied physiological perturbations (such as antibiotic stress or expression of synthetic genes). Notably, the broad agreement across data sets is obtained using a single core parameter set which does not require any adjustment from one

scenario to the next. As such, the flux-parity model predicts the microbial 'growth laws,' providing a mechanistic explanation for previous phenomenological models formulated to understand them (*Scott et al., 2010*; *Scott et al., 2014*; *Molenaar et al., 2009*). The finding that these predictions hold so well despite the overwhelmingly complex nature of the cell further highlights that biological systems are not irreducibly complex but can be distilled to a small number of fundamental components sufficient to capture the core behavior of the system.

As proteins commonly account for the majority of biomass in microbial organisms and the core processes of protein synthesis are universally conserved among them, it is likely that protein synthesis is a fundamental growth constraint across many organisms. Accordingly, flux-parity regulation may be a very general scheme which ensures the efficient coordination of metabolic and translational fluxes across many microbial organisms. And as our modeling approach is organism agnostic, it should be transferable to a variety of microbes growing in nutrient-replete conditions. Indeed, other organisms including *S. cerevisiae* exhibit a strict interdependence between growth rate and ribosome content (*Karpinets et al., 2006*; *Metzl-Raz et al., 2017*), as is predicted by the flux-parity model. However, more quantitative data on ribosomal content, translation speeds, upshift dynamics, and more need to be acquired to fully examine the commonality of flux-parity regulation in the microbial world.

A common interpretation of previous allocation models is that cells maximize their growth rate in whatever conditions they encounter (*Bosdriesz et al., 2015*; *Towbin et al., 2017*). Rather, we believe flux-parity regulation only ensures optimal coordination between metabolic and translational fluxes. It does not imply that the growth rate itself is maximized or directly sensed. In particular, the flux-parity model does not assume that the pool of metabolic proteins is tailored to maximize the metabolic flux and thus growth in the encountered conditions. This is in agreement with an expanding body of evidence which shows that microbes frequently synthesize metabolic and other proteins which are not directly needed in the encountered condition and thus impede growth. *E. coli*, for example, synthesizes a plethora of different transport proteins when exposed to poor growth conditions even if the corresponding substrates are not available, collectively occupying a significant portion of the proteome (*Belliveau et al., 2021*; *Schmidt et al., 2016*; *Hui et al., 2015*; *Balakrishnan et al., 2021a*). Accordingly, it has been observed that cells stop synthesizing these proteins when evolving over many generations in the absence of those sugars (*Leiby and Marx, 2014*; *Favate et al., 2021*).

But why, then, do we observe an optimal allocation between metabolic and ribosomal proteins when the pool of metabolic proteins itself shows this apparent non-optimal behavior? We posit here that both behaviors emerge from the adaptation to fluctuating conditions: in contrast to the well-defined static conditions of laboratory experiments, the continuous ebb and flow of nutrients in natural environments precludes any sense of stability. Accordingly, the machinery of the cell should be predominantly adapted to best cope with the fluctuating conditions microbial organisms encounter in their natural habitats (*Koch, 1971*). A complex regulation of metabolic proteins is thus expected, including, for example, the diverse expression of nutrient transporters which promote growth in anticipated conditions, rather than synthesizing only those specific to nutrients that are present in the moment (*Balakrishnan et al., 2021a*).

However, in those fluctuating conditions, flux-parity regulation promotes rapid growth. To illustrate this point, we consider again a nutrient upshift in which there is an instantaneous improvement in the nutrient conditions. We compare the predicted response via flux-parity (*Figure 5A*, red box) with that predicted by a simpler step-wise regulation where the allocation solely depends on the environmental condition (and not the internal fluxes) and immediately adjusts to the new steady value at the moment of the shift (*Figure 5A*, blue box). The dynamic reallocation by flux-parity facilitates a sharp increase in the allocation towards ribosomes (*Figure 5B*), resulting in a rapid increase in instantaneous growth rate compared to the step-wise reallocation mechanism (*Figure 5C*), suggesting that flux-parity is advantageous in fluctuating environments. As its regulation solely depends on the internal state of the cell (particularly, the relative abundance of charged- to uncharged-tRNA), it holds independently of the encountered conditions. This stands in contrast to the regulation of metabolic proteins, where both the external and internal states dictate what genes are expressed. As a result, optimal coordination between metabolic and translational fluxes occurs ubiquitously across conditions and not only in those that occur in natural habitats and drive adaptation. These broader conditions include steady-state growth within the laboratory, with the 'growth laws' observed under those conditions emerging as a serendipitous consequence.

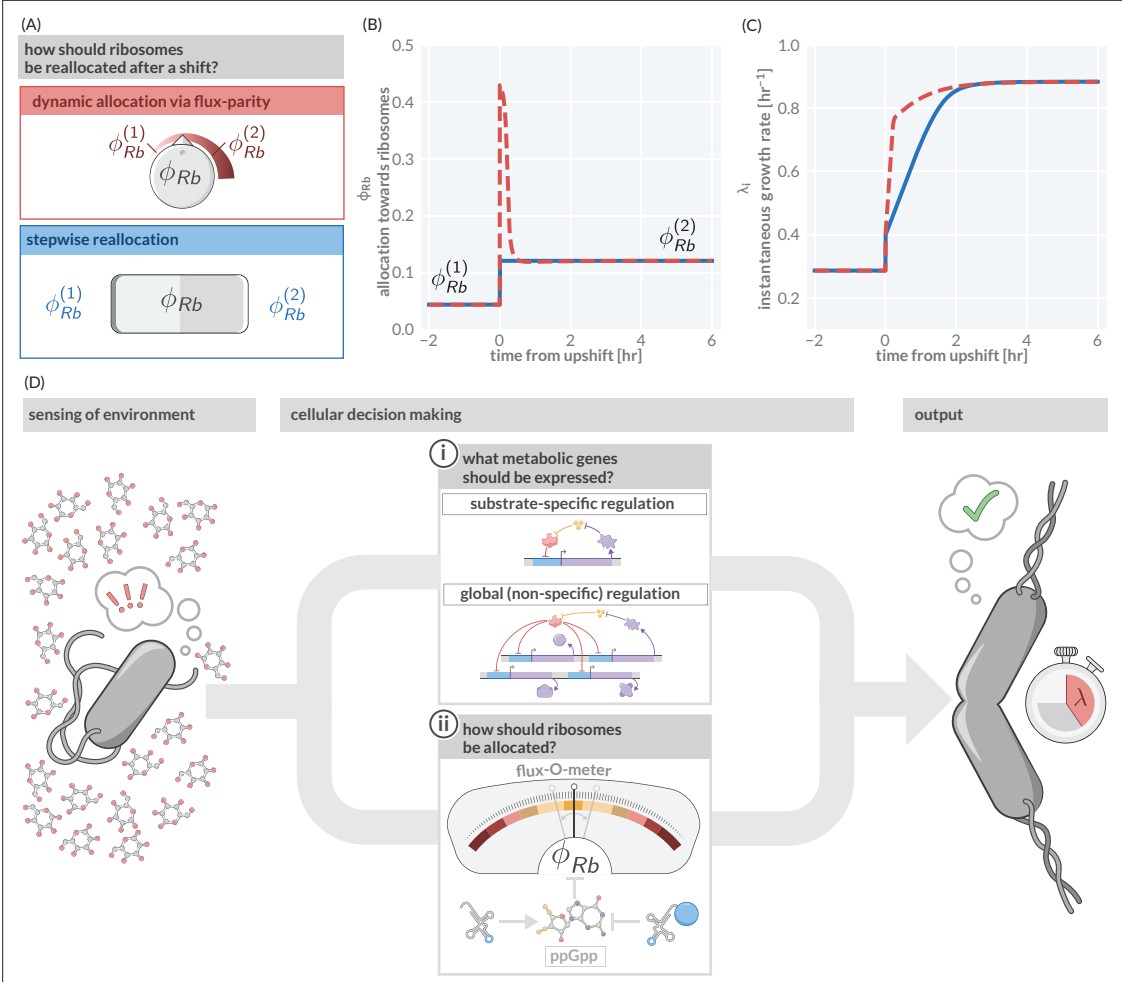

**Figure 5.** Flux-parity allocation as a strategy to adapt to fluctuating conditions. (**A**) Ribosome reallocation strategies upon a nutrient upshift. After a nutrient upshift, cells either dynamically reallocate their ribosomes given flux-parity regulation (top, red) or they undergo stepwise reallocation from one steady-state value to the next (bottom, blue). (**B**) The allocation dynamics for both strategies in response to a nutrient upshift. (**C**) The instantaneous growth rate for both strategies over the course of the shift. Dashed red and solid blue lines correspond to model predictions for optimal allocation and flux-parity regulation, respectively. (**D**) Cellular decision making in fluctuating environments. Upon sensing features of the environment, cells undergo a two-component decision making protocol defining what metabolic genes should be expressed (top) and how the allocation towards ribosomes should be adjusted to maintain flux-parity. The combination of these processes yield an increase of biomass at a given characteristic growth rate.

In summary, we view the process of cellular decision making as having two major components (***Figure 5D***): (i) determining what metabolic genes should be expressed given the environmental and physiological state and (ii) determining how ribosomes should be allocated given the metabolic and translational fluxes. Flux-parity regulation can explain the latter but many details of the former remain enigmatic. Additional studies are thus required to understand how the regulation of metabolic genes depends on encountered conditions and how it is shaped by adaptation to specific habitats. However, the ability of this theory to predict complex phenotypes across scales suggests that it can also act as a basis to answer these questions, and thereby galvanize an integrative understanding of microbial life connecting physiology, ecology, and evolution.

## Methods
### Formulating the allocation model

Here we present a step-by-step derivation of the low-dimensional allocation model we use to describe bacterial growth. We provide additional biological motivation for its construction and highlight the different assumptions and simplifications invoked. To maintain consistency with the literature, we

largely follow the notational scheme introduced by *Scott et al., 2014* and define each symbol as it is introduced.

## Synthesis of proteins

The rate of protein synthesis is determined by two quantities: the total number of ribosomes $N_{Rb}$ and the speed $v_{tl}$ at which they are translating. The latter depends on the concentration of precursors needed for peptide bond formation, such as tRNAs, free amino acids, and energy sources like ATP and GTP. Taking the speed $v_{tl}$ as a function of the concentration of the collective precursor pool $c_{pc}$, the increase in protein biomass $M$ follows as

$$\frac{dM}{dt} = v_{tl}(c_{pc})N_{Rb}. \tag{5}$$

There exists a maximal speed at which ribosomes can operate, $v_{tl}^{max}$, that is reached under optimal conditions when precursors are highly abundant, in *E. coli* approximately 20 amino acids (AA)/second (s) (*Forchhammer and Lindahl, 1971*). Conversely, the translation speed falls when precursor concentrations $c_{pc}$ get sufficiently small. Simple biochemical considerations support a Michaelis–Menten relation (*Ehrenberg and Kurland, 1984*; *Klumpp et al., 2013*; *Belliveau et al., 2021*) as good approximation of this behavior with the specific form

$$v_{tl}(c_{pc}) = v_{tl}^{max}\left(\frac{c_{pc}}{c_{pc}+K_M^{c_{pc}}}\right), \tag{6}$$

where $K_M^{c_{pc}}$ is a Michaelis–Menten constant with the maximum speed $v_{tl}^{max}$ only observed for $c_{pc} \gg K_M^{c_{pc}}$. The number of ribosomes $N_{Rb}$ can be approximated given knowledge of the total mass of ribosomal proteins $M_{Rb}$ and the proteinaceous mass of a single ribosome $m_{Rb}$ via $N_{Rb} \approx M_{Rb}/m_{Rb}$ (more details in Appendix 1 Estimating the number of ribosomes within the cell). The increase in protein biomass (*Equation 5*) is thus

$$\frac{dM}{dt} = v_{tl}(c_{pc})\frac{M_{Rb}}{m_{Rb}} \equiv \gamma(c_{pc})M_{Rb}. \tag{7}$$

The *translation rate* $\gamma(c_{pc}) \equiv v_{tl}(c_{pc})/m_{Rb}$ describes the rate at which ribosomes generate new protein.

The maximal translation rate $\gamma_{max} \equiv v_{tl}^{max}/m_{Rb}$ imposes a firm upper limit (*Dill et al., 2011*; *Belliveau et al., 2021*; *Kafri et al., 2016*) of how rapidly biomass can accumulate, unrealistically assuming the system would consist of only ribosomes translating at maximum rate. Notably, however, this upper limit is not much faster than the fastest growth observed, highlighting the importance of protein synthesis in defining the timescale of growth. For example, the maximal translation rate for *E. coli* is $\approx 10$ hr$^{-1}$ and thus only $\approx 4$ times higher than the growth rates in rich LB media ($\lambda \approx 2.5$ hr$^{-1}$). Including the synthesis of rRNA, another major component of the cellular dry mass, lowers this theoretical limit only marginally (*Kostinski and Reuveni, 2020*), further supporting our sole consideration of protein synthesis in defining growth. The difference between measured growth rates and the theoretical limits can be mostly attributed to the synthesis of metabolic proteins which generate the precursors required for protein synthesis, which we consider next.

## Synthesis of precursors

Microbial cells are generally capable of synthesizing precursors from nutrients available in the environment, such as sugars or organic acids. This synthesis is undertaken by a diverse array of metabolic proteins ranging from those which transport nutrients across the cell membrane, to the enzymes involved in energy generation (such as those of fermentation or respiration), and the enzymes providing the building blocks for protein synthesis (such as those involved in the synthesis of amino acids). While these enzymes vary in their abundance and kinetics, we group them all into single set of metabolic proteins with a mass $M_{Mb}$ which cooperate to synthesize the collective pool of precursors from nutrients required for protein synthesis. We make the approximation that these metabolic proteins generate precursors at an effective *metabolic rate* $\nu$. In general, this rate depends on the concentration of nutrients $c_{nt}$ in the environment. This relation is canonically described by a Monod (Michaelis–Menten) relation

$$\nu(c_{nt}) = \nu^{max}\left(\frac{c_{nt}}{c_{nt}+K_M^{c_{nt}}}\right),\tag{8}$$

where $\nu_{max}$ is the maximum metabolic rate describing how fast the metabolic proteins can synthesize precursors, and $K_M^{c_{nt}}$ is the Monod constant describing the concentration below which nutrient utilization slows (**Monod, 1949**). Novel precursors are thus supplied with a total rate of $\nu(c_{nt})M_{Mb}$ and consumed via protein synthesis at a rate $\gamma(c_{pc})M_{Rb}$. Translation relies on precursors and, as introduced above, the translation rate $\gamma(c_{pc})M_{Rb}$ thus depends on the concentration of precursors in the cell, $c_{pc}$. As we do not explicitly model cell division, we here approximate this cellular concentration as the relative mass abundance of precursors to total protein biomass. This approximation is justified by the observation that cellular mass density and total protein content is approximately constant across a wide range of conditions (**Belliveau et al., 2021**; **Martínez-Salas et al., 1981**; **Kubitschek et al., 1983**). The dynamics of precursor concentration follows from the balance of synthesis, consumption, and dilution as the total biomass grows:

$$\frac{dc_{pc}}{dt} = \overbrace{\frac{\nu(c_{nt})M_{Mb}}{M}}^{\text{production via metabolism}} - \underbrace{\frac{\gamma(c_{pc})M_{Rb}}{M}}_{\text{consumption via protein synthesis}} - \overbrace{\frac{c_{pc}\gamma(c_{pc})M_{Rb}}{M}}^{\text{dilution via growth}}.\tag{9}$$

While the dilution term is often assumed to be negligible, this term is critical to describe growth and derive analytical expressions. Furthermore, we note that the precursor concentration is defined such that the consumption of one precursor yields the addition of one amino acid to the biomass $M$. As we measure proteins in units of amino acids, there is thus no conversion factor needed when describing the consumption of precursors by protein synthesis.

## Simplification of saturating nutrients

The introduced dynamics simplifies when the nutrient concentration in the environment $c_{nt}$ well exceeds the Monod constant $K_M^{c_{nt}}$ as $\nu(c_{nt})$ simplifies to $\nu_{max}$. Steady growth for which biomass increases exponentially readily emerges. This is the scenario we focus on in in the first half of this work. It should be noted, however, that biologically such a scenario can only be realized temporarily as the nutrient supply required by the exponentially growing biomass can only be sustained by the environment for a limited amount of time. In general, the nutrient levels vary.

## Consumption of nutrients in batch culture growth

The synthesis of novel precursors relies on the availability of nutrients which changes depending on the environment. In *Figure 1—figure supplement 2*, we consider specifically a 'batch culture' scenario in which nutrients are provided only at the beginning of growth and are never replenished. Therefore, growth of the culture continues until all of the nutrients have been consumed. The concentration of nutrients in the environment is thus given as

$$\frac{dc_{nt}}{dt} = -\frac{\nu(c_{nt})M_{Mb}}{Y},\tag{10}$$

where $Y$ is the yield coefficient which describes how many nutrient molecules are needed to produce one unit of precursors.

## Ribosomal allocation of protein synthesis

As final step of the model definition, we must describe how cells direct their protein synthesis towards making ribosomes, metabolic proteins, or all other proteins that make up the cell (colored arrows in *Figure 1A*). We do so by introducing three *allocation parameters* $\phi_{Rb}$, $\phi_{Mb}$, and $\phi_O$ (such that $\phi_{Rb} + \phi_{Mb} + \phi_O = 1$) which define how novel protein synthesis is partitioned among these categories:

$$\frac{dM_{Rb}}{dt} = \phi_{Rb}\frac{dM}{dt}; \frac{dM_{Mb}}{dt} = \phi_{Mb}\frac{dM}{dt}; \frac{dM_O}{dt} = \phi_O\frac{dM}{dt}.\tag{11}$$

These equations are summarized in *Figure 1B* and *Figure 1*, *Figure 1—figure supplement 2* and define the accumulation of biomass, from nutrient uptake to protein synthesis.

## Approximating concentration via relative abundance

In addition to maintaining the *total* macromolecular densities, cells also maintain an approximately constant protein density (*Bremer and Dennis, 2008*). This observation allows for a major simplification when formulating the allocation model, namely the approximation of concentrations as relative mass abundances. The rate $\gamma$ at which ribosomes can synthesize protein is dependent on the abundance of precursors, $c_{pc}$, in the cell. To compute the concentration and/or density in typical units (e.g. µM, or mass/volume), we would require some measure of the total cellular volume, $V_{cell}$, such that the concentration follows

$$c_{pc} = \frac{M_{pc}}{V_{cell}},\qquad(12)$$

with $M_{pc}$ denoting the total mass of the precursor pool. By making the experimentally supported assertion that the protein density $\rho$ is constant, we can say that

$$\rho = \frac{M}{V_{cell}} = \text{Constant},\qquad(13)$$

where $M$ is the total protein biomass. Thus, the total cellular volume $V_{cell}$ can be computed as

$$V_{cell} = \frac{M}{\rho}.\qquad(14)$$

Plugging this result into *Equation 12*, we arrive at the approximation

$$c_{pc} = \rho\frac{M_{pc}}{M} \approx \frac{M_{pc}}{M}.\qquad(15)$$

In this work, we neglect $\rho$ as a multiplicative constant and treat $c_{pc}$ as being dimensionless. We direct the reader to *Scott et al., 2010* and *Milo, 2013* for a further discussion of the conversion between concentration and relative abundance.

## Derivation of analytical expressions

In the first section of this work, we present several analytical relations pertinent to steady-state growth. These relations follow from the simple allocation model and describe (i) how the growth rate depends on model parameters (*Figure 1C*) and (ii) how ribosome content depends on other model parameters for the three different regulation scenarios we discuss (*Figure 1F*). Here, we introduce a step-by-step derivation of these expressions.

## Deriving the steady-state growth rate

We begin with deriving an expression for the steady-state growth rate $\lambda$ which is similar to previous approaches taken by *Giordano et al., 2016* and *Dourado and Lercher, 2020*. As discussed in *Figure 1—figure supplement 2*, steady-state conditions are satisfied when two conditions are met. First, the dynamics of the precursor concentration is constant (i.e., $\frac{dc_{pc}}{dt} = 0$) and the composition of the proteome matches the allocation parameters (i.e., $\frac{M_{Rb}^*}{M^*} = \phi_{Rb}^*$ and $\frac{M_{Mb}^*}{M^*} = \phi_{Mb}^*$). Furthermore, we assume that in steady-state growth, the concentration of nutrients in the environment is saturating ($c_{nt} \gg K_M^{c_{nt}}$), meaning that $\nu(c_{nt}) \approx \nu_{max}$. With these conditions satisfied, we can rewrite *Equation 9* as

$$\frac{dc_{pc}}{dt} = \nu_{max}\phi_{Mb} - \gamma(c_{pc}^*)\phi_{Rb} - c_{pc}\gamma(c_{pc}^*)\phi_{Rb} = 0,\qquad(16)$$

where $c_{pc}^*$ is the steady-state precursor concentration.

Noting that in steady-state conditions the total biomass increases exponentially at a rate $\lambda \equiv \gamma(c_{pc})\phi_{Rb}^*$, *Equation 16* can be simplified to

$$\frac{dc_{pc}}{dt} = \nu_{max}\phi_{Mb}^* - \lambda(1 + c_{pc}) = 0.\qquad(17)$$

We can therefore solve for the steady-state precursor concentration $c_{pc}^*$ to yield

$$c_{pc}^* = \frac{\nu_{max} \phi_{Mb}^*}{\lambda} - 1. \tag{18}$$

Assuming a Michaelis–Menten form for the translation rate $\gamma(c_{pc}^*)$, we can now define it as a function of the growth rate $\lambda$ as

$$\gamma(c_{pc}^*) = \frac{\gamma_{max}}{1 + \frac{K_M^{c_{pc}}}{c_{pc}}} = \frac{\gamma_{max}}{1 + \frac{K_M^{c_{pc}} \lambda}{\nu_{max} \phi_{Mb}^* - \lambda}}. \tag{19}$$

Knowing that the growth rate $\lambda \equiv \gamma(c_{pc^*}) \phi_{Rb}^*$, and $\phi_{Mb}^* = 1 - \phi_{Rb}^* - \phi_O^*$, we say that

$$\lambda = \frac{\gamma_{max} \phi_{Rb}^*}{1 + \frac{K_M^{c_{pc}} \lambda}{\nu_{max}(1 - \phi_{Rb}^* - \phi_O^*) - \lambda}}. \tag{20}$$

This can be algebraically manipulated to yield a quadratic equation of the form

$$\lambda^2 \left(1 - K_M^{c_{pc}}\right) + \lambda \left(\nu_{max}(1 - \phi_{Rb}^* - \phi_O^*) + \gamma_{max} \phi_{Rb}^*\right) - \gamma_{max} \phi_{Rb}^* \nu_{max}(1 - \phi_{Rb}^* - \phi_{Mb}^*) = 0, \tag{21}$$

which has one positive root of

$$\lambda = \frac{\nu_{max}(1 - \phi_{Rb}^* - \phi_O^*) + \gamma_{max} \phi_{Rb}^* - \sqrt{(\nu_{max}(1 - \phi_{Rb}^* - \phi_O^*) + \gamma_{max} \phi_{Rb}^*)^2 - 4(1 - K_M^{c_{pc}}) \gamma_{max} \phi_{Rb}^* \nu_{max}(1 - \phi_{Rb}^* - \phi_O^*)}}{2\left(1 - K_M^{c_{pc}}\right)}. \tag{22}$$

For notational simplicity, we can define the maximum metabolic output and the maximum translational output as $N = \nu_{max}(1 - \phi_{Rb} - \phi_O)$ and $\Gamma = \gamma_{max} \phi_{Rb}$, respectively, and substitute them into *Equation 22* to generate

$$\lambda = \frac{N + \Gamma - \sqrt{(N + \Gamma)^2 - 4\left(1 - K_M^{c_{pc}}\right) N \Gamma}}{2\left(1 - K_M^{c_{pc}}\right)}, \tag{23}$$

## Defining $\phi_{Rb}$ for scenarios II and III

In *Figure 1F*, we provide a description of three plausible regulatory scenarios microbes may employ to regulate their ribosomal content. Scenario I assumes just a constant, arbitrary allocation parameter $\phi_{Rb} \in [0, 1 - \phi_O]$. Here, we provide a short derivation for the more complicated relations describing ribosomal content under scenarios II and III.

## Scenario II: Constant translation rate

The second regulatory scenario assumes that the ribosomal content is adjusted to maintain a specific standing concentration of precursors, which we denote as $c_{pc}^*$. Noting that the growth rate $\lambda \equiv \gamma(c_{pc}^*) \phi_{Rb}^*$, we can restate *Equation 18* in the form

$$c_{pc}^* = \frac{\nu_{max}(1 - \phi_O^* - \phi_{Rb}^*)(c_{pc}^* + K_M^{c_{pc}})}{c_{pc}^* \gamma_{max} \phi_{Rb}^*}. \tag{24}$$

Some algebraic rearrangement allows us to solve for $\phi_{Rb}^*$, yielding

$$\phi_{Rb} = \frac{(1 - \phi_O^*) \nu_{max} \left(c_{pc}^* + K_M^{c_{pc}}\right)}{\nu_{max} \left(c_{pc}^* + K_M^{c_{pc}}\right) + \gamma_{max} c_{pc}^* \left(c_{pc}^* + 1\right)}. \tag{25}$$

This expression is equivalent to that shown for scenario II in *Figure 1F*. In evaluating this scenario, we considered the regime in which precursors were in abundance, meaning $c_{pc}^* \gg K_M^{c_{pc}^*}$. Under this regime, *Equation 25* simplifies further to

$$\phi_{Rb}^* \approx \frac{(1-\phi_O^*)\nu_{max}}{\gamma_{max}\left(c_{pc}^*+1\right)+\nu_{max}}. \tag{26}$$

This represents a strategy where the cell adjusts $\phi_{Rb}^*$ to maintain a translation rate very close to $\gamma_{max}$.

## Scenario III: Optimal allocation

In this work, we define the optimal allocation of ribosomes $\phi_{Rb}^*$ to be that which maximizes the growth rate in a given environment and at a given metabolic state. To determine the optimal $\phi_{Rb}^*$, we can differentiate *Equation 22* with respect to $\phi_{Rb}^*$ to yield the cumbersome expression

$$\frac{\partial\lambda}{\partial\phi_{Rb}^*} = \frac{1}{2\left(1+K_M^{c_{pc}}\right)} \times \tag{27}$$

$$\left[\gamma_{max}-\nu_{max}-\frac{2\gamma_{max}\nu_{max}\left(1-K_M^{c_{pc}}\right)\left(2\phi_{Rb}^*+\phi_O^*-1\right)+(\gamma_{max}-\nu_{max})\left(\gamma_{max}\phi_{Rb}^*+\nu_{max}\left(1-\phi_O^*-\phi_{Rb}^*\right)\right)}{\sqrt{\left(\gamma_{max}\phi_{Rb}^*+\nu_{max}\left(1-\phi_O^*-\phi_{Rb}^*\right)\right)^2-4\left(1-K_M^{c_{pc}}\right)\gamma_{max}\nu_{max}\phi_{Rb}^*\left(1-\phi_O^*-\phi_{Rb}^*\right)}}\right] \tag{28}$$

Setting this expression equal to zero and solving for $\phi_{Rb}$ results in

$$\phi_{Rb} = \frac{(1-\phi_O^*)\left(\gamma_{max}\nu_{max}\left(1-2K_M^{c_{pc}}\right)+\nu_{max}^2+\sqrt{K_M^{c_{pc}}\gamma_{max}\nu_{max}(\gamma_{max}-\nu_{max})}\right)}{(\gamma_{max}+\nu_{max})^2-4K_M^{c_{pc}}\gamma_{max}\nu_{max}} \tag{29}$$

which is the optimal allocation towards ribosomes as presented in *Figure 1F*.

## Implementing flux-parity regulation via ppGpp

Here we expand upon and derive the equations defining the flux-parity allocation model shown schematically in *Figure 2A* and explore its dependence on parameter values.

### Formulation of model

To include ppGpp signaling into the ribosomal allocation model, we must perform two tasks. First, we must explicitly model the dynamics of both charged- and uncharged-tRNAs. Secondly, we must tie the relative abundances of these tRNAs to the allocation parameters such that when charged-tRNAs are limiting and uncharged-tRNAs in abundance, the system reacts by adjusting the allocation parameters towards ribosomal proteins and away from metabolic proteins ($\phi_{Rb}$ and $\phi_{Mb}$).

We consider there to be two pools of tRNAs: those charged with an amino acid (denoted as $tRNA^c$) and those that are uncharged ($tRNA^u$). Rather than keeping track of the copy numbers of these tRNAs, we instead model their concentration as relative mass abundances (relative to the total protein biomass $M$), treating each tRNA to have an effective mass of one amino acid as each tRNA can in principle be charged. Much as for consideration of precursors in the simpler model we can model the concentration dynamics of these pools of tRNAs by considering three processes: the generation of the tRNAs, the consumption of the tRNAs, and the effect of dilution as the biomass grows.

We begin first with modeling the dynamics of the charged-tRNA pool, $tRNA^c$. Here, we consider that charged-tRNAs are synthesized from one free amino acid and one uncharged-tRNA and further assume that the pool of free amino acids is abundant enough such that the tRNA pool is the rate limiting component. Making this assumption allows us to state that the conversion of one uncharged-tRNA to one charged-tRNA via the metabolic machinery proceeds at a rate $\nu(tRNA^u)$, itself dependent on the uncharged-$tRNA^u$ concentration. Likewise, we consider that the conversion of one charged-tRNA to an uncharged-tRNA is only possible via protein synthesis, which proceeds at a rate $\gamma(tRNA^c)$ that is dependent on the *charged-tRNA* concentration. Finally, we must also consider how the mere fact of growing biomass effectively dilutes the charged-tRNA concentration. Together, these processes can be combined to enumerate the dynamics of the charged-tRNA pool as

$$\frac{dtRNA^c}{dt} = \overbrace{\frac{\nu(tRNA^u)M_{Mb}}{M}}^{\text{generation via metabolism}} - \underbrace{\frac{\gamma(tRNA^c)M_{Rb}}{M}}_{\text{consumption via protein synthesis}} - \overbrace{\frac{tRNA^c\gamma(tRNA^c)M_{Rb}}{M}}^{\text{reduction via dilution}}. \tag{30}$$

The dynamics for the pool of uncharged-tRNAs can be constructed in a similar manner, with the caveat that the generation of new uncharged-tRNAs occurs from both protein synthesis (converting one charged-tRNA into one uncharged-$tRNA^u$) and from transcription of the individual tRNA genes. We consider the latter to occur at a rate $\kappa$, which has dimensions of concentration per unit time. Using the same logic of mapping the productive and consumptive processes, we can enumerate the dynamics of the uncharged-tRNA pool as

$$\frac{dtRNA^u}{dt} = \overbrace{\kappa}^{\text{production via transcription}} + \underbrace{\frac{\gamma(tRNA^c)M_{Rb}}{M}}_{\text{occurance via protein synthesis}} - \overbrace{\frac{\nu(tRNA^u)M_{Mb}}{M}}^{\text{consumption via metabolism}} - \underbrace{\frac{tRNA\gamma(tRNA^c)M_{Rb}}{M}}_{\text{reduction via dilution}}. \tag{31}$$

These expressions comprehensively define the dynamics of the tRNA pool, from generation via transcription to their recycling between charged and uncharged states through metabolic and translational fluxes, respectively. As in the main text, we posit that the dynamics of the ribosomal $M_{Rb}$, metabolic $M_{Mb}$, and 'other' $M_O$ protein masses follow via the allocation parameters $\phi_{Rb}$, $\phi_{Mb}$, and $\phi_O$ respectively. However, in this treatment of the model, we consider these parameters, with the exception of $\phi_O$, to be dynamic and depending on the intracellular concentration of ppGpp. Mathematically, we state this as

$$\frac{dM_{Rb}}{dt} = \phi_{Rb}(\text{ppGpp})\frac{dM}{dt} \; ; \; \frac{dM_{Mb}}{dt} = [1 - \phi_O - \phi_{Rb}(\text{ppGpp})]\frac{dM}{dt} \; ; \; \frac{dM_O}{dt} = \phi_O\frac{dM}{dt}. \tag{32}$$

We are now tasked with (i) enumerating the dynamics of ppGpp and (ii) assigning a specific functional form to $\phi_{Rb}(\text{ppGpp})$. The biochemistry of ppGpp synthesis, degradation, and binding to the transcription machinery has been studied in *E. coli* among other prokaryotes, revealing the enzyme(s) important for this process, In *E. coli* RelA and SpoT. Many molecular details revealing how those enzymes control ppGpp levels in response to the abundance of tRNA levels are known but important details also remain puzzling (*Magnusson et al., 2005*; *Anderson et al., 2021*). Thus, while previous works have consider the dynamics of these specific proteins in more detail (*Bosdriesz et al., 2015*; *Giordano et al., 2016*), we here take a more coarse-grained view. Specifically, we first make the ansatz that the dynamics of ppGpp synthesis and degradation are sufficiently fast compared to the timescale of protein synthesis such that it can be treated as being in steady-state instantaneously. Secondly, we take the concentration of ppGpp to be inversely proportional to the relative abundance of charged- to uncharged-tRNAs,

$$\text{ppGpp} \propto \frac{1}{\frac{tRNA^c}{tRNA^u}}. \tag{33}$$

This is a well-motivated starting point as in *E. coli*, ppGpp is primarily synthesized via RelA when an uncharged-tRNA enters the A-site of a translating ribosome, forming a stalled complex. As binding of a charged-tRNA or an uncharged-tRNA is a competitive process, the probability of one or the other being bound is dependent on their relative concentrations, rather than the absolute concentrations of either species. However, other processes which affect ppGpp levels, including the synthesis and degradation by SpoT in relation to ribosome activity, are less well understood (*Srivatsan and Wang, 2008*). Accordingly, we consider our approach to describe ppGpp as inversely proportional to the relative abundance of charged- to uncharged-tRNAs as a motivated ansatz rather than a fully established biochemical relation. And we furthermore show below that this ansatz works much better for describing the experimental observations as a few different ones we probed.

Given the relation between ppGpp and tRNA charging ratio, *Equation 33*, we can now define the allocation towards ribosomes to be a function of the tRNA charging ratio, $\phi_{Rb}\left(\frac{tRNA^c}{tRNA^u}\right)$. To assign a specific functional form to this relation, we assume that the expression of ribosomal genes is in first

order described by a simple binding kinetics of ppGpp to the transcriptional machinery and the allocation towards ribosomes follows a form similar to that of a Michaelis–Menten relation,

$$\phi_{Rb}\left(\frac{tRNA^c}{tRNA^u}\right) = (1-\phi_O)\frac{\frac{tRNA^c}{tRNA^u}}{\frac{tRNA^c}{tRNA^u}+\tau}. \tag{34}$$

Here, the parameter $\tau$ represents the value of the charged- to uncharged-tRNA ratio where $\phi_{Rb}$ is at its half-maximal value. The maximal value itself depends on the magnitude of $\phi_O$, the allocation towards other proteins, which we are considering to be independent of ppGpp; $\phi_{Rb}^{(max)} = 1 - \phi_O$.

The transcription of tRNA genes towards novel tRNA synthesis has also been shown to be regulated with ppGpp, appearing to closely match the regulatory behavior of ribosomal proteins (**Jinks-Robertson et al., 1983**). We therefore model that the tRNA synthesis rate $\kappa$ (introduced in **Equation 31**) is similarly modulated by the charged- to uncharged-tRNA ratio,

$$\kappa\left(\frac{tRNA^c}{tRNA^u}\right) = \kappa_{max}\frac{\frac{tRNA^c}{tRNA^u}}{\frac{tRNA^c}{tRNA^u}+\tau}. \tag{35}$$

Here, $\kappa_{max}$ is the rate of tRNA transcription when all tRNA genes are fully saturated with RNA polymerase in rich growth conditions where gene dosage is high. Finally, we must establish functional forms for the tRNA dependencies on the metabolic and translation rate. Simple biochemical assumptions permit a formulation of a Michaelis–Menten function for each rate. Noting that the translation rate $\gamma$ is defined as $\gamma \equiv \frac{v_{tl}}{m_{Rb}}$, where $v_{tl}$ is the translation speed and $m_{Rb}$ is the proteinaceous mass of a single ribosome, we take $\gamma(tRNA^c)$ to be of the form

$$\gamma(tRNA^c) = \frac{v_{tl}^{(max)}}{m_{Rb}}\frac{tRNA^c}{tRNA^c+K_M^{(tRNA^c)}}, \tag{36}$$

where $v_{tl}^{(max)}$ is the maximum translation speed and $K_M^{(tRNA^c)}$ is the Michaelis–Menten constant. A similar argument can be made for the dependence of the metabolic rate $\nu$ on the uncharged-tRNA concentration,

$$\nu(tRNA^u) = \nu_{max}\frac{tRNA^u}{tRNA^u+K_M^{(tRNA^u)}}, \tag{37}$$

with $K_M^{(tRNA^u)}$ being another Michaelis–Menten constant. Together, **Equations 30–37** mathematically describe a model for ppGpp-dependent regulation of translational and metabolic fluxes.

In principle, an analytical solution for this system of ODEs can be found, though it precludes evaluation by hand and is computationally intensive. While we do not solve this system of ODEs analytically here, we can numerically integrate them to sufficiently approximate the steady-state behavior. Depending on the choice of parameter values, such an approach can yield an allocation scenario nearly indistinguishable from that of the optimal allocation scenario (scenario III) of the simple model (**Figure 1H and I**).

## Optimal allocation emerges from flux-parity regulation

While the previous section lays out the mathematics of the flux-parity model, we now discuss how this regulation scheme can lead to an optimal allocation. Towards this goal, we first discuss in more detail what we mean when we say 'flux-parity.' As described in the main text, we define flux-parity as a balance *and* mutual maximization of (i) the flux of uncharged-tRNAs through metabolism (termed the *metabolic flux* $J_{Mb}$) and (ii) the flux of charged-tRNAs through protein synthesis (termed the *translational flux* $J_{Tl}$). To demonstrate this point, assume that we can decouple the dependence of the allocation parameter $\phi_{Rb}$ from the ratio of charged- to uncharged-tRNAs. Mathematically speaking, we can define the metabolic flux as the collective action of metabolic proteins,

$$J_{Mb} = \nu(tRNA^u)\phi_{Mb} = \frac{\nu_{max}tRNA\left(1-\phi_O-\phi_{Rb}\right)}{tRNA^u+K_M^{tRNA^u}}. \tag{38}$$

Similarly, we can state that the translational flux is the collective action of ribosomal proteins,

$$J_{Tl} = \frac{\gamma_{max}tRNA^c\phi_{Rb}}{tRNA^c+K_M^{tRNA^c}} \tag{39}$$

So long as these fluxes are equivalent, a steady-state is satisfied. However, this steady-state is not necessarily the optimal value. This is illustrated in *Figure 2*, *Figure 2—figure supplement 1*. For example, if we consider that $\phi_{Rb}$ is too large for the given condition (*Figure 2—figure supplement 1*, left), a specific steady-state is realized (black point). If $\phi_{Rb}$ is further increased, the value of both the metabolic and translational fluxes (dashed lines) must decrease to reach a new steady state and growth rate thus declines. However, if $\phi_{Rb}$ is decreased, the value of both fluxes increase and growth-rate thus also increases as well. At optimum allocation (where growth is locally maximized, *Figure 2—figure supplement 1*, middle), any perturbation to $\phi_{Rb}$ will necessarily result in a decrease in the fluxes, indicating that at the optimal allocation the fluxes are *mutually maximized*.

As the concentrations of both tRNA species (*Equations 30 and 31*) are dependent on the allocation towards ribosomes $\phi_{Rb}$ in inverse ways, the ratio of their concentrations acts as an effective sensor of the magnitude of either flux. A large charged- to uncharged-tRNA ratio indicates that there is an abundance of charged-tRNAs, suggesting that the translational flux is too low. Conversely, a small charged- to uncharged-tRNA ratio indicates a translational flux that is too large, diminishing the metabolic flux. By tying the allocation towards ribosomes $\phi_{Rb}$ to this ratio, an allocation can emerge that optimizes the fluxes and thus growth.

## Assessing different assumptions of $\phi_{Rb}$ dependence on ppGpp

In *Equation 33*, we made the assumption that the concentration of ppGpp was inversely proportional to the charging balance of the tRNA pools. We put this forward as an ansatz with the motivation that the degree of tRNA charging should be related to the amount of ppGpp synthesized. However, there are other ansatzes that could be made relating the amount of ppGpp to the individual concentrations of the tRNAs, or other ratiometric definitions.

To test how sensitive our findings are to the particular ansatz used, we considered other ways in which the ppGpp concentration could be related to the tRNA pools. There is strong biochemical evidence that a primary route of ppGpp synthsesis is via the enzyme RelA, which becomes active when associated to a 'stalled' ribosome—one that is bound to an uncharged tRNA—though some details remain enigmatic. In manner similar to other works (*Giordano et al., 2016*; *Wu et al., 2022*; *Bosdriesz et al., 2015*), we can assert that the amount of ppGpp is proportional to the abundance of stalled ribosomes. Mathematically, we can define the ppGpp concentration as being proportional to the probability of a ribosome binding an uncharged tRNA. Assuming that the tRNA concentration (of both charged and uncharged forms) is sufficiently high that all ribosomes are complexed with a tRNA, this equates to

$$[ppGpp] \propto P_{\text{bound}}^{\text{(uncharged)}} \approx \frac{tRNA^u}{tRNA^c + tRNA^u}, \tag{40}$$

where $tRNA^c$ and $tRNA^u$ represent the absolute concentrations of charged and uncharged species, respectively. If the ppGpp concentration is inversely proportional to the allocation towards ribosomes, we can similarly make the argument that the ribosomal allocation $\phi_{Rb}$ will be proportional to the probability of a ribosome being bound to a *charged-tRNA*,

$$\phi_{Rb} = (1 - \phi_O) P_{\text{bound}}^{\text{(charged)}} = (1 - \phi_O) \frac{tRNA^c}{tRNA^u + tRNA^c}. \tag{41}$$

This equation mechanistically operates in a similar way as *Equation 34*—the allocation towards ribosomes depends on the relative amounts of charged- and uncharged-tRNAs. In the extreme limit where the total concentration of tRNA is fixed (for which there is conflicting evidence; *Dong et al., 1996*; *Skjold et al., 1973*; *Bremer and Dennis, 2008*; *Bosdriesz et al., 2015*; *Giordano et al., 2016*), *Equation 41* and *Equation 34* are mathematically equivalent. However, the predicted scaling dependence of ppGpp takes a different form.

In the main text, we noted that the concentration of ppGpp relative to a reference growth rate $\frac{[ppGpp]}{[ppGpp]_0}$ is equivalent to the inverse ratio of the charging balances. Under the ansatz that the ppGpp concentration is depending on the uncharged-tRNA binding probability, this relation takes the form

$$\frac{[ppGpp]}{[ppGpp]_0} = \frac{P_{\text{bound}}^{\text{(uncharged)}}}{P_{\text{bound}_0}^{\text{(uncharged)}}} = \frac{1 + \frac{tRNA_0^c}{tRNA_0}}{1 + \frac{tRNA^c}{tRNA^u}}, \tag{42}$$

where the subscript 0 denotes the reference state value. This distinction, coupled with experimental measurements of the relative ppGpp concentrations, allows us to test the validity of the two assumed forms for $\phi_{Rb}$.

*Figure 3—figure supplement 1* shows the predictive capacity of these two ansatzes with the simple binding (*Equation 41*) and ratiometric (*Equation 33*) predictions shown in solid-blue and dashed-red lines, respectively. While both of these assumptions are capable of predicting the scaling of the ribosome content and translation speed with quantitative equivalence, there is a distinct difference in the predicted behavior of the relative ppGpp concentrations. The simple binding ansatz predicts a significantly shallower dependence on the growth rate than is observed in the data and in the ratiometric prediction. Thus, it appears that relating [ppGpp] to the ratio of uncharged- to charged-tRNA concentrations accurately captures the behavior of *E. coli*, though there remain gaps in our understanding of this relationship at a biochemical level.

## Incorporating effects of ribosome-targeting antibiotics

To extend the flux-parity allocation model and incorporate the effects of antibiotic treatment, we must consider the mechanism of action of the antibiotic, specifically chloramphenicol. Chloramphenicol is a bacteriostatic antibiotic with tightly, but reversibly, binds to the ribosome. Once bound, the ribosome is unable to resume translation until chloramphenicol dissociates. Thus, we can model the effect of this drug by enumerating the probability that chloramphenicol is bound to a ribosome $P_{\text{bound}}$ at a given concentration $c_{cm}$. Mathematically, this can be stated as

$$P_{\text{bound}} = \frac{c_{cm}}{c_{cm} + K_D^{cm}},\tag{43}$$

where $K_D^{cm}$ is an effective dissociation constant of chloramphenicol to a unit of ribosomal mass accounting for kinetics transport and ribosome binding. We can then say that the probability of a ribosome being active is equal to the probability of a ribosome being *unbound*,

$$P_{\text{active}} = 1 - P_{\text{bound}} = 1 - \frac{c_{cm}}{c_{cm} + K_D^{cm}}.\tag{44}$$

As only active ribosomes will contribute to the accumulation of biomass, we must rewrite the dynamics as

$$\frac{dM}{dt} = \gamma(tRNA^c)M_{Rb}^{\text{active}} = \gamma(tRNA^c)P_{\text{active}}M_{Rb}.\tag{45}$$

To make the predictions shown in *Figure 3E and F*, we assumed that the chloramphenicol concentration in the growth medium is equal to the intracellular concentration and take $K_D^{cm} \approx 0.5$ nM.

## Incorporating effects of excess protein stress

We consider that the excess protein synthesis can be modeled as the introduction of a new protein class, which we consider to have a total mass of $M_X$. Following the allocation parameters of the flux-parity model as defined in *Equation 32*, we can introduce a new allocation parameter $\phi_X$ such that

$$\frac{dM_X}{dt} = \phi_X \frac{dM}{dt} \ ; \ \phi_O + \phi_{Mb} + \phi_{Rb} + \phi_X = 1.\tag{46}$$

In *Figure 3 I*, we show that a collection of data can be collapsed onto a single line that relates the relative change in growth rate as a function of the excess protein that is synthesized. While we cannot fully solve the flux-parity model analytically, we can derive an analytical expression of this relation. Specifically, we note that the steady-state growth rate in the absence of excess expression $\lambda$ follows the simple relation

$$\lambda = \gamma(tRNA^c)\phi_{Rb}\left(\frac{tRNA^c}{tRNA^u}\right) = \gamma_{max}(1 - \phi_O)\frac{tRNA^c}{tRNA^c + K_M^{tRNA^c}}\frac{\frac{tRNA^c}{tRNA^u}}{\frac{tRNA^c}{tRNA^u} + \tau}.\tag{47}$$

This can be easily extended to compute the growth rate under excess protein synthesis $\lambda_X$ as

$$\lambda_x = \gamma(tRNA^c)\phi_{Rb}\left(\frac{tRNA^c}{tRNA^u}\right) = \gamma_{max}(1 - \phi_O - \phi_X)\frac{tRNA^c}{tRNA^c + K_M^{tRNA^c}}\frac{\frac{tRNA^c}{tRNA^u}}{\frac{tRNA^c}{tRNA^u} + \tau}.\tag{48}$$

We can take the ratio of these growth rates to yield an expression for the collapse function

$$\frac{\lambda_X}{\lambda} = \frac{\gamma_{max}(1-\phi_O-\phi_X)\frac{tRNA^c}{tRNA^c+K_M^{tRNA^c}}\frac{\frac{tRNA^c}{tRNA^u}}{\frac{tRNA^c}{tRNA^u}+\tau}}{\gamma_{max}(1-\phi_O)\frac{tRNA^c}{tRNA^c+K_M^{tRNA^c}}\frac{\frac{tRNA^c}{tRNA^u}}{\frac{tRNA^c}{tRNA^u}+\tau}}. \tag{49}$$

If we assume that the excess protein synthesis affects *only* $\phi_X$, leaving all other parameters untouched, *Equation 49* reduces to the concise form

$$\frac{\lambda_X}{\lambda} = \frac{1-\phi_O-\phi_X}{1-\phi_O}, \tag{50}$$

which is the linear relation plotted in *Figure 3I*.

Aside from the collapse, we also show how the flux-parity model quantitatively predicts the growth rate as a function of excess protein for three different media (*Figure 3H*). In this case, we require some knowledge of what the metabolic rate $\nu_{max}$ is for those specific conditions. As the metabolic rate is an efficient rate incorporating the action of different metabolic reactions and serving as a proxy of the nutrient quality, it is not possible to make an a priori estimate of its value. To nevertheless estimate $\nu_{max}$ for each condition, we determined its value by using the simple allocation model as encoded in 'Derivation of analytical expressions,' assuming the growth rate $\lambda$ and the ribosomal content describes the allocation towards ribosomes $\phi_{Rb}$. Under the simple allocation model, we note that an expression for the metabolic rate can be solved from the steady-state precursor concentration $c_{pc}^*$ (*Equation 18*) to yield

$$\nu_{max} = \frac{\lambda(c_{pc}+1)}{1-\phi_O-\phi_{Rb}}. \tag{51}$$

The steady-state precursor concentration $c_{pc}^*$ can be solved from the definition of the steady-state growth rate and has the form

$$c_{pc}^* = \frac{K_D^{c_{pc}}\lambda}{\phi_{Rb}\gamma_{max}\left(1-\frac{\lambda}{\phi_{Rb}}\right)}. \tag{52}$$

Combining *Equations 51 and 52* yields an expression for the maximal metabolic rate $\nu_{max}$,

$$\nu_{max} = \frac{\lambda}{1-\phi_O-\phi_{Rb}}\left(\frac{K_D^{c_{pc}}\lambda}{\phi_{Rb}\gamma_{max}\left(1-\frac{\lambda}{\phi_{Rb}}\right)}+1\right). \tag{53}$$

Thus, given knowledge of the steady-state growth rate $\lambda$ and the allocation towards ribosomes $\phi_{Rb}$ (which are both measured quantities), the value of $\nu_{max}$ can be derived.

## Incorporating effects of nutrient upshifts

To model the dynamics of growth in fluctuating conditions, we asserted that a nutritional upshift is equivalent to an instantaneous change in the metabolic rate such that $\nu_{max}^{preshift} < \nu_{max}^{postshift}$. However, this is not completely sufficient to capture the phenomenology that is observed. It is becoming exceedingly clear that bacterial cells are non-optimal in *what* genes they express, with many proteins that are synthesized are ultimately useless in the specific condition (*Balakrishnan et al., 2021a*). This can have very important effects on the growth rate as any amount of conditionally useless protein that is synthesized consumes resources that could otherwise be partitioned to the proteins that need to be synthesized. To incorporate this effect, we introduce another protein class with an allocation parameter $\phi$. As the degree of conditionally useless expression is significantly more pronounced in slow rather than fast conditions (*Balakrishnan et al., 2021a*; *Belliveau et al., 2021*; *Schmidt et al., 2016*), we further asserted that the magnitude of this sector also changed in response to the nutritional upshift such that $\phi^{preshift} > \phi^{postshift}$. The precise value of this sector is less important than the difference in the pre- and post-shift condition and can be considered as an additional rescaling factor as described in Appendix 1 Neglecting the other proteins. Thus, for all nutritional shifts in this work, we considered that $\phi^{postshift} = 0$ and the value of $\phi^{preshift}$ to be linearly proportional to the difference in the growth rates between the pre- and post-shift conditions.

## Incorporating effects of nutrient depletion

Up to this point, we have explored the flux-parity model under the assumption that the nutrients in the environment were saturating, such that $\nu(c_{nt}) \approx \nu_{max}$. However, a dependence on the environmental nutrient concentration $c_{nt}$ can be easily included in the definition of the metabolic rate $\nu$ as

$$\nu(tRNA, c_{nt}) = \nu_{max} \left( \frac{tRNA^u}{tRNA^u + K_M^{tRNA^u}} \right) \left( \frac{c_{nt}}{c_{nt} + K_M^{c_{nt}}} \right), \tag{54}$$

where $K_M^{c_{nt}}$ is the Michaelis–Menten constant. We can then model the dynamics of the nutrient concentration $c_{nt}$ in a batch-culture system as

$$\frac{dc_{nt}}{dt} = - \frac{\nu(tRNA, c_{nt}) M_{Mb}}{Y}, \tag{55}$$

where $Y$ is the yield coefficient.

## Data sets

This work leverages a large collection of data, primarily from *E. coli*, to evaluate the accuracy of our model in describing biological phenomena. These data come from a range of studies spanning around 50 years of measurements from different groups and different geographical locations. Collecting and curating this large data set required the manual transcribing of data from papers as well as various standardization steps to ensure that measurements were truly comparable between studies, as is outlined in *Supplementary file 2*.

For proper referencing and attribution, we list the data sources here as follows: *Albertson and Nyström, 1994*; *Baracchini and Bremer, 1988*; *Basan et al., 2015*; *Bentley et al., 1990*; *Bremer and Dennis, 2008*; *Bren et al., 2013*; *Brunschede et al., 1977*; *Büke et al., 2022*; *Buckstein et al., 2008*; *Coffman et al., 1971*; *Dai et al., 2016*; *Dalbow and Young, 1975*; *Dong et al., 1995*; *Erickson et al., 2017*; *Forchhammer and Lindahl, 1971*; *Gausing, 1972*; *Hernandez and Bremer, 1990*; *Hernandez and Bremer, 1993*; *Imholz et al., 2020*; *Kepes and Beguin, 1966*; *Korem Kohanim et al., 2018*; *Lacroute and Stent, 1968*; *Lazzarini et al., 1971*; *Li et al., 2014*; *Li et al., 2018*;; *Mori et al., 2017*; *Morris and Hansen, 1973*; *Oldewurtel et al., 2021*; *Panlilio et al., 2020*; *Pedersen, 1984*; *Ryals et al., 1982*; *Sarubbi et al., 1988*; *Schmidt et al., 2016*; *Schleif, 1967*; *Schleif et al., 1973*; *Scott et al., 2010*; *Si et al., 2017*; *Skjold et al., 1973Sloan and Urban, 1976*; *Sokawa et al., 1975*; *Wu et al., 2022*; *You et al., 2013*; *Young and Bremer, 1976*; *Zhu and Dai, 2019*; *Bonven and Gulløv, 1979*; *Lacroute, 1973*; *Metzl-Raz et al., 2017*; *Paulo et al., 2015*; *Paulo et al., 2016*; *Riba et al., 2019*; *Siwiak and Zielenkiewicz, 2010*; *Waldron and Lacroute, 1975*; *Xia et al., 2021*; *Rohatgi, 2021*.

## Acknowledgements

We thank Uri Alon, Markus Arnoldini, Rachel Banks, Suzy Beeler, Nathan Belliveau, Terence Hwa, Soichi Hirokawa, Christine Jacobs-Wagner, Sergey Kryazhimskiy, Armita Nourmohammad, Manuel Razo-Mejia, Tom Röschinger, Jan Skotheim, Cat Triandafillou, and members of the Tadashi Fukami, Dmitri Petrov, Alfred Spormann, and JC research groups for extensive discussions and the critical reading of the manuscript. GC acknowledges support by the NSF Postdoctoral Research Fellowships in Biology Program (grant no. 2010807).

## Additional information

### Funding

| Funder | Grant reference number | Author |
| --- | --- | --- |
| National Science Foundation | 2010807 | Griffin Chure |
| Stanford Bio-X | | Jonas Cremer |

| Funder | Grant reference number | Author |
|---|---|---|

The funders had no role in study design, data collection and interpretation, or the decision to submit the work for publication.

## Author contributions

Griffin Chure, Conceptualization, Resources, Data curation, Software, Formal analysis, Supervision, Funding acquisition, Validation, Investigation, Visualization, Methodology, Writing - original draft, Project administration, Writing - review and editing; Jonas Cremer, Conceptualization, Resources, Supervision, Funding acquisition, Investigation, Methodology, Writing - original draft, Project administration, Writing - review and editing

## Author ORCIDs
Griffin Chure http://orcid.org/0000-0002-2216-2057
Jonas Cremer http://orcid.org/0000-0003-2328-5152

## Decision letter and Author response
Decision letter https://doi.org/10.7554/eLife.84878.sa1
Author response https://doi.org/10.7554/eLife.84878.sa2

# Additional files

## Supplementary files
• Supplementary file 1. Reference parameter set for *E. coli* used in this work. (a) Value was estimated as follows. The maximum tRNA synthesis rate was taken to be that in very rich media. In these conditions, each *E. coli* cell has ≈ 5–8 copies of its genome and each genome is assumed to have ≈ 64 copies of tRNA genes. At a transcription rate of 40 nt/s, each tRNA gene can produce ≈ 1 tRNA per second *Belliveau et al., 2021*. Together, this yields an estimate for the maximum rate of tRNA synthesis to be ≈ 5× genomes × 64 genes/genome.

• Supplementary file 2. *E. coli* data used in this work and the corresponding sources. (a) Concentration of ppGpp was computed relative to that of the sample with the closest growth rate of 1.0 hr$^{-1}$ to remain consistent with *Wu et al., 2022*. (b) Source data was not available, so data was determined from figures using WebPlotDigitizer (*Rohatgi, 2021*). (c) Converted to ribosome content by assuming 0.86 µg of rRNA per 1 µg RNA and 0.53 µg of ribosomal protein per 1 µg of rRNA. This yields a conversion factor of $M_{Rb} = 0.4558$RNA. (d) Original values were calculated assuming ≈ 70% of ribosomes were active. We assume all ribosomes are active and recalculated the values accordingly. (e) Growth rate immediately following the shift was calculated by averaging values within first 20 min after the shift to be consistent with procedure reported by *Korem Kohanim et al., 2018*. (f) Original data curated and standardized by *Belliveau et al., 2021*.

• Supplementary file 3. Reference parameter set for *S. cerevisiae* used in this work.

• Supplementary file 4. *S. cerevisiae* data used in this work and the relevant sources. (a) Original values were calculated assuming ≈ 70% of ribosomes were active. We assume all ribosomes are active and recalculated the values accordingly. (b) Source data was not available, so data was approximated from Figure using WebPlotDigitizer (*Rohatgi, 2021*). (c) Calculated given the total mass of protein per cell in Table 5 and the number of ribosomes in Table 4. A length of 11,984 AA per ribosome and an average amino acid mass of 110 Da /AA was used to calculate total ribosomal mass.

• MDAR checklist

## Data availability
All data is available via the paper GitHub repository (https://github.com/cremerlab/flux_parity; copy archived at *Chure et al., 2023*) and is registered in Zenodo via DOI: https://doi.org/10.5281/zenodo.5893799.

The following datasets were generated:

| Author(s) | Year | Dataset title | Dataset URL | Database and Identifier |
|---|---|---|---|---|
| Chure G, Cremer J | 2023 | Collated measurements of *E. coli* ribosomal mass fractions | https://github.com/cremerlab/flux_parity/blob/master/data/main_figure_data/ecoli_ribosomal_mass_fractions.csv | GitHub, ecoli_ribosomal_mass_fractions.csv |
| Chure G, Cremer J | 2023 | Collated measurements of *E. coli* translation rates | https://github.com/cremerlab/flux_parity/blob/master/data/main_figure_data/ecoli_peptide_elongation_rates.csv | GitHub, ecoli_peptide_elongation_rates.csv |
| Chure G, Cremer J | 2023 | Collated measurements of relative *E. coli* ppGpp concentrations | https://github.com/cremerlab/flux_parity/blob/master/data/main_figure_data/ecoli_relative_ppGpp.csv | GitHub, ecoli_relative_ppGpp.csv |
| Chure G, Cremer J | 2023 | Collated measurements of *E. coli* useless protein overexpression | https://github.com/cremerlab/flux_parity/blob/master/data/main_figure_data/Fig5B_overexpression_growth_rates.csv | GitHub, Fig5B_overexpression_growth_rates.csv |
| Chure G, Cremer J | 2023 | Collated measurements of nutrient upshift dynamics in *E. coli* | https://github.com/cremerlab/flux_parity/blob/master/data/main_figure_data/Fig5C_shift_magnitudes.csv | GitHub, Fig5C_shift_magnitudes.csv |
| Chure G, Cremer J | 2023 | All data is available via the paper is registered in Zenodo | https://doi.org/10.5281/zenodo.5893799 | Zenodo, 10.5281/zenodo.5893799 |

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

## Appendix 1

### Allocation models to study microbial growth

Over the past several decades, theoretical studies have introduced a huge variety of mathematical models to better understand how microbes grow. These range from simple phenomenological relations, first introduced by *Verhulst, 1845* and *Monod, 1942* to describe exponential growth and nutrient consumption, to more recent high-dimensional models which explicitly consider hundreds of cellular processes involved in biomass synthesis and growth. Here, we summarize different modeling approaches which integrate considerations of protein synthesis and metabolism to rationalize bacterial growth and quickly compare the construction and results to the model presented in the main text.

As outlined in the main text, the idea that protein synthesis constitutes a major limitation of microbial growth has a storied history. With an ever-improving characterization of both the composition and the major biochemical processes undertaken by microbes, it has become increasingly clear that the auto-catalytic nature of protein synthesis (i.e., 'ribosomes making ribosomes') and the corresponding allocation of ribosomal activity towards different proteins are of paramount importance. Therefore, it is imperative to consider protein synthesis and this allocation when modeling cell growth (*Hernandez and Bremer, 1993*; *Koch, 1988*). Over the years, very different approaches have been introduced to model protein synthesis and growth. To provide an overview, we roughly distinguish between higher-dimensional, coarse-grained, and low-dimensional approaches in the coming sections.

Higher-dimensional models build on the genetic annotations and reaction networks in *E. coli* to specifically account for hundreds (or even thousands) of molecular reactions. An important class of these are holistic cell-growth models which include thousands of reactions involved in protein synthesis (*Karr et al., 2012*; *Macklin et al., 2020*). These studies can describe a number of observations, including the concentration of different metabolites, as well as the relation between gene expression and the abundances of thousands of proteins. However, as it is difficult to estimate the many parameters involved, these models are typically constrained to only a few reference growth conditions like growth on glucose, often with moderate uncertainty. Moreover, the integration of hundreds of different reactions often counteracts the development of a more intuitive understanding of microbial growth and necessitates computationally costly analyses.

A second class of models are coarse grained models which substantially simplify cellular life, often by focusing on a subset of biochemical processes in a less detailed manner, yet still requiring dozens to hundreds of parameters. One example is the framework introduced by *Weiße et al., 2015* to analyze the relation between gene expression (transcription), protein synthesis (translation), and growth across varied conditions. Compared to the aforementioned holistic growth models, this is a lower-dimensional approach with only a few different types of reactions. A more recent example of different scope is the work by *Hu et al., 2020*. The authors formulated a modeling framework which focuses on protein synthesis and explicitly considers 274 reactions driven by equally many enzymes. The authors then studied how the allocation of protein synthesis towards the different enzymes affects growth. Numerical simulations show that an allocation behavior to reach optimal growth does not imply that the speed with which ribosomes translate is fast. Rather, optimal allocation and efficient growth is often reached even when translation speeds are substantially lower than the maximum in some conditions. We arrive at a similar conclusion and discuss this important point in the main-text when we introduce different regulatory 'scenarios.'

Different in scope to these 'medium-dimensional' coarse-grained approaches are those we truly call 'low-dimensional' which utilize only a few parameters, typically less than 10. In a now seminal work, *Molenaar et al., 2009* introduced such a low-dimensional approach to model protein synthesis and the growth-rate dependent switch between different metabolic pathways. Central to this model is a consideration of how protein synthesis resources are partitioned among four different protein classes, including ribosomal and metabolic proteins. Through enumerating coupled differential equations that relate the partitioning to growth rate, their model predicts, for example, that ribosome content needs to scale linearly with growth rate to ensure efficient growth, though comparison with experimental data is absent.

Another early and significant low-dimensional modeling approach was introduced by *Scott et al., 2014* only a few years later. This work is similar in spirit to that of Molenaar et al., but adds a deeper motivation of the low-dimensional allocation approach by building on the authors' previous experimental study (*Scott et al., 2010*) which introduced novel phenomenological relations describing ribosome content and its dependence on growth rate. In particular, the authors discuss how feedback in the regulation of ribosomes and metabolic proteins (end-product inhibition and supply-driven activation) can lead to the observed scaling of ribosome content with growth rate. As such, the article highlights the important question of global regulation beyond the control of single genes. However, the authors rationalize this regulatory scenario as being necessary to ensure a stable steady-state regime. This arises in part due to the authors' assumption that the dilution of metabolic precursors is negligible, a process we believe is physiologically critical to consider. Consequentially, the regulatory scheme which underlies the coordination of ribosomes and metabolic proteins is quite different to what other low-dimensional models have proposed and what we present here (see also Section 3 of this supplement).

Following studies by Molenaar et al. and Scott et al., other low-dimensional allocation models have been introduced with various extensions and modifications. The models presented by *Maitra and Dill, 2015* and *Giordano et al., 2016*, for example, follow a similar low-dimensional description as Scott et al. The explicit consideration of precursor dilution allowed them to derive analytical solutions describing steady state growth, similar to those we present in this work. *Dourado and Lercher, 2020* recently extended these results by providing analytical solutions for a generalized allocation model with the solution presented for a low-dimensional limit being similar to that of *Giordano et al., 2016* and what we used as the starting point for our analysis of optimal allocation.

Several studies also extended the allocation framework to more explicitly model the global regulation that may be at play. In particular, several studies have investigated how the global regulator guanosine tetraphosphate (ppGpp) may tune the allocation of ribosomes to obtain optimal growth across conditions, specifically in *E. coli* (*Giordano et al., 2016*; *Bosdriesz et al., 2015*). These models often take a quite fine-grained view of the kinetics of ppGpp synthesis, its degradation, its dependence on stalled ribosomes, and even its mechanism of regulation. In some cases, particular details of the kinetics of transcriptional initiation of ribosomal RNA genes is explicitly considered (*Bosdriesz et al., 2015*). A commonality between these models is (once again) an enumeration of a handful of coupled ordinary differential equations, though their precise functional forms are unique. These theoretical analyses suggested that the ppGpp-mediated regulation feedback can robustly tune ribosome content with encountered conditions to support optimal growth in steady conditions. However, the evaluation of these models and particularly the chosen approaches to relate tRNA charging levels (for which there is contradictory data; *Skjold et al., 1973*; *Bremer and Dennis, 2008*) to ppGpp concentrations remain limited as only a cursory and largely qualitative comparison with data is presented. In this work, we provide an unprecedented quantitative comparison between available experimental data and our own low-dimensional very coarse-grained approach to model tRNA charging and ppGpp regulation. The excellent match between theory and observation (*Figure 3C*) confirms the important role of tRNA charging and ppGpp in mediating allocation and growth as highlighted by Giordano et al. and Bosdriez et al.

Notably, however, we choose a different ansatz in relating tRNA charging levels to ppGpp which we find crucial to describe the diverse growth phenomena in changing conditions (see Section 8). Recently, *Wu et al., 2022* also presented a phenomenologically guided modeling approach to further understand the relation between ribosome content and growth rate, with a focus on the possible role of ppGpp in regulating ribosome *activity* in addition to expression. We think the derived picture is substantially different from that present in our work. Particularly, the authors contend that a condition-dependent translation rate is at odds with the principle of optimal allocation and cannot be understood without explicit consideration of an inactive ribosome pool. Following our analysis, including a condition-dependent translation rate *strongly supports* the observed relation between ribosome content and growth rate, strengthening a picture of optimal allocation without requiring an inactive pool of ribosomes for the majority of growth conditions. We discuss the topic of inactive ribosomes further in Appendix 1 - Inactive ribosomes.

Low-dimensional allocation models have further been used to study growth beyond exponential growth in steady conditions. Particularly, *Erickson et al., 2017* formulated an allocation model to

analyze growth when the abundance of the major growth supporting carbon source rapidly changes (up- and downshifts). Their approach utilizes (and enforces) linear phenomenological relations describing ribosomal and metabolic protein content observed during steady-state growth. As such, the study can describe certain transitions quite well, particularly the up- and downshift from growth on gluconate/glucose to growth on succinate. *Korem Kohanim et al., 2018* used a low-dimensional modeling approach to analyze the rapid increase in growth when conditions improve (nutritional upshift). The modeling approach particularly allowed them to explain the rapid increase of growth rates, with the increase depending on observed steady-state growth rates before and after the shift. *Mori et al., 2017* also utilized phenomenological relations and a low-dimensional allocation model to explore the possible benefits of maintaining a reserve of inactive ribosomes. *Balakrishnan et al., 2021a* further extended the allocation modeling approach to analyze the physiological origin of long lag times which frequently emerge during diauxic growth on different carbon sources. These works illustrate that there are many approaches one can take toward rationalizing growth out of steady state, each with different assumptions and behaviors in various limits.

Our approach and the extensive comparison between data and theory we present confirm the overall logic presented by these studies: growth transitions are largely determined by the way cells coordinate translation and gene expression during the shift. However, our approach is different in scope as the core of our model is the tRNA-dependent control of allocation as fundamental regulation scheme, rather than relying on linear phenomenological relations to describe the 'regulatory function.' This allows us to rationalize the emergence of many growth behaviors without assuming specific relations between growth rate and physiological quantities as a starting point. Furthermore, we also caution against the common assumption in these models that the offset in the approximately linear relation between steady growth rate and ribosome content represents inactive ribosomes which changes the overall perception of how cells operate to efficiently grow in steady conditions and when environments change (see discussion in Appendix 1 - Inactive ribosomes)

More recently, low-dimensional allocation models have further been extended to investigate additional aspects of cell physiology beyond biomass accumulation and growth. For example, *Roy et al., 2021* explicitly considered the relationship between protein and RNA synthesis and the interesting question how different autocatalytic cycles like those involved in protein and RNA synthesis have to be coupled to promote growth. We discuss in Appendix more specifically the role of RNA synthesis and why we refrained from modeling it explicitly. Additional studies have also extended the allocation framework to explicitly consider cell-size control and proteins involved in division (*Serbanescu et al., 2020*; *Bertaux et al., 2020*); however, a consistent description of growth and cell size is still missing and part of ongoing research efforts.

## The major simplifications of low-dimensional allocation models and why they might work

Cells are highly complex entities containing thousands of molecular players which interact in myriad ways varying over space and time. The allocation models boldly simplifies this reality. Some of the most important simplifications are:

1. All cells as being treated as identical and encountering the same condition, including the nutrients which they consume to build new cellular material.
2. Cells are not treated individually but the overall biomass is considered. As such, the approach ignores considerations of cell division and variations in cell physiology throughout the cell cycle.
3. Rather than considering the abundance of thousands of unique protein species within a cell, allocation models take a coarse-grained view of the composition of the proteome where proteins are pooled together. We pool specifically proteins being either ribosomal (i.e., synthesizing new protein from precursors such as charged-tRNA), metabolic (i.e., synthesizing precursors from nutrients), or being involved in other biological processes required for cellular growth and survival *Figure 1—figure supplement 1*.

Here we summarize our view of why some of the low-dimensional allocation models can develop such a predictive power despite all these major simplifications.

As mentioned, the low-dimensional allocation framework is completely ignorant to (i) the spatial arrangement of processes across the cell and (ii) neglects the existence of cells as a whole. Instead,

the description only considers the change of total protein biomass of the system (i.e., culture) over time. Despite this objectively major simplification, the model quantitatively predicts growth phenotypes across a broad range of conditions, as we presented in the main text. Remarkably, this is accomplished with one core set of parameters (such as the maximal speed of translation) which remains fixed across conditions. This is a surprising result given that cell size and the spatiotemporal arrangement of cellular components are known to be highly dependent on conditions, which also should affect major model parameters (such as the maximal speed of translation). The finding that the model can nevertheless capture observations across conditions thus suggests to us that many cellular processes, including those involved in cell size control, are highly coordinated by the cell such that translation and metabolism work efficiently and can be captured by simple rate equations; it is the complexity of the cellular machinery which allows us to formulate a simple but predictive model of growth.

To further illustrate this point, we consider specifically the density of macromolecules within the cell which strongly informs the rates of myriad biochemical reactions. A density that is too large, for example, will strongly hamper diffusion and thus slow many reaction rates, while a low density generally reduces binding rates (*Delarue et al., 2018*; *van den Berg et al., 2017*). Cells are thus expected to maintain macro-molecular density ranges within narrow ranges to operate efficiently (*van den Berg et al., 2017*). Experimental studies support this idea. For example, mass densities in *E. coli* appear stay within narrow ranges across growth conditions (*Oldewurtel et al., 2021*). Complex biophysical processes appear to also be in place to control the spatial arrangement of macromolecules, such as the strong mutual-exclusivity of DNA and ribosomes within the cytoplasm (*Gray et al., 2019*). If it were not for these processes, macromolecular densities and thus major cellular processes (like translation) would vary tremendously with growth conditions and a model based on simple rate equations would have very limited predictive power.

## Precursors concentrations and the importance of dilution by cell growth

In the main text, we consider that the translation rate $\gamma$ is dependent on the concentration of precursors $c_{pc}$. The tug-of-war between the metabolic processes (synthesizing precursors) and protein synthetic processes (consuming precursors) is what determines this value. With the protein density of cells being approximately constant , we consider the precursors concentration as the mass of precursors per total protein mass, $c_{pc}$. We describe the dynamics of this precursor concentration $c_{pc}$ as

$$\frac{dc_{pc}}{dt} = \nu(c_{nt})\frac{M_{Mb}}{M} - \gamma(c_{pc})\frac{M_{Rb}}{M} - \underbrace{c_{pc}\gamma(c_{pc})\frac{M_{Rb}}{M}}_{\text{dilution}},$$

(A1)

where $M$, $M_{Mb}$, and $M_{Rb}$ denote the masses of the total protein, metabolic protein, and ribosomal protein pools, respectively. The latter term in the above equation denotes the decrease in the precursor concentration due to the increase in biomass; that is, this term considers the decrease in precursor concentration due to dilution upon a growing cell volume. Through a change-of-variables from $m_{pc}$ to $c_{pc}$, it mathematically follows that

$$\frac{dc_{pc}}{dt} = \frac{d}{dt}\left(\frac{M_{pc}}{M}\right) = \frac{1}{M}\frac{dM_{pc}}{dt} - \underbrace{c_{pc}\frac{1}{M}\frac{dM}{dt}}_{\text{dilution}}.$$

(A2)

1In many previous allocation models (outlined in Section 1), it is assumed that the effect of dilution is negligible compared to the magnitude of metabolism and protein synthesis (*Towbin et al., 2017*; *Scott et al., 2014*; *Bosdriesz et al., 2015*). It is important to note, however, that the effect of dilution is *not* negligible when one considers the *difference* between the metabolic and translational processes as has been emphasized by *Giordano et al., 2016* and *Dourado and Lercher, 2020*. To illustrate this further, consider *Equation A1* which in steady state equates to zero. Upon some rearrangement we have

$$\nu(c_{nt})\phi_{Mb} - \gamma(c_{pc})\phi_{Rb} = c_{pc}\gamma(c_{pc})\phi_{Rb}.$$

(A3)

At steady state, $\frac{M_{Rb}^*}{M^*} = \phi_{Rb}^*$ and $\frac{M_{Mb}^*}{M^*} = \phi_{Mb}^*$. When the effect of dilution is neglected (i.e., setting the right-hand side of **Equation A3** equal to 0), it is required that steady state is only reached when the two fluxes are equal. Thus, if there is any perturbation to the precursor concentration, such as a sudden influx (**Appendix 1—figure 1A**) or efflux (**Appendix 1—figure 1B**) of precursors, the system has no recourse to re-establish a steady state (dashed lines in **Appendix 1—figure 1**). In the case where the effect of dilution is not neglected, the system relaxes back to the stable steady state (solid lines in **Appendix 1—figure 1**) as the influence of dilution increases or decreases in response to the change in precursor concentration.

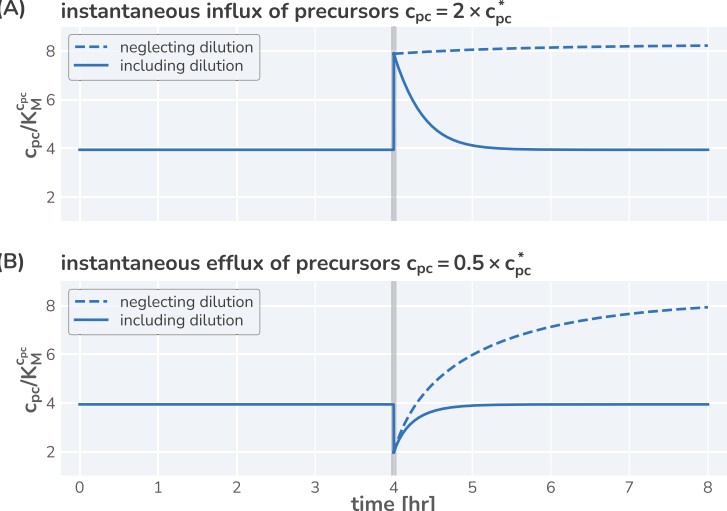

**Appendix 1—figure 1.** Neglecting effect of precursor dilution results in an unstable steady state. Integrated dynamics of the model equations including (solid lines) and neglecting (dashed lines) the dilution of the precursors with increasing biomass. At time $t = 4$ hr (gray vertical line), the precursor concentration is instantaneous increased (**A**) or decreased (**B**) by a factor of 2 and the system is allowed to respond. Neglecting the effect of dilution results in a monotonic increase in the precursor concentration, whereas including dilution allows a rapid return to the steady-state concentration.

To obtain biologically relevant analytical solutions, we thus explicitly include the dilution factor. Notably, keeping the dilution term around is also what allows analytical solutions to be derived (**Giordano et al., 2016**; **Dourado and Lercher, 2020**). In many previous studies (**Towbin et al., 2017**; **Scott et al., 2014**), it was necessary to include end-product inhibition as a regulatory element of the metabolic flux, one which we do not need to consider here. In mathematical terms, no additional parameters need to be included to define a dependence of the metabolic rate on the precursor concentration As such, the emergence of a steady state is less of a mystery: it does not require the integration of complex regulation schemes but readily emerges in steady environmental conditions and when the allocation of protein synthesis is constant.

## Additional considerations relevant at slow growth

While the optimal allocation model (scenario III) can describe the observed relation between ribosome abundance and growth rates remarkably well for fast growth, predictions and observations disagree when growth rates fall below $\lambda < 0.5$ hr$^{-1}$ (**Figure 1H and I**). The observed values are higher than what is predicted by the regulation scenarios II (constant translation) and III (optimal allocation) which might be attributed to a range of additional aspects, including (i) the active degradation of proteins, (ii) a comparison with data which has not yet reached steady-state, and (iii) an increase in the fraction of inactive ribosomes at slower growth. Here we discuss these additional aspects and conclude that additional experiments are needed before extending our modeling approach to additionally capture very slow growth conditions.

## Active protein degradation

The active degradation of proteins is an additional factor which might require a higher content of ribosomes than what is predicted by the model at slow growth. In formulating the allocation model,

we considered only protein synthesis but not degradation. Mathematically, we modeled the change in protein mass to depend solely on translation,

$$\frac{dM}{dt} = \gamma(c_{pc})M_{Rb},$$ (A4)

but did not include a degradation term, for example,

$$\frac{dM}{dt} = \underbrace{\gamma(c_{pc})M_{Rb}}_{\text{synthesis}} - \underbrace{k_{deg}M}_{\text{degradation}}$$ (A5)

This simplification is well justified when growth is fast since observed degradation half-lives are then very long compared to the timescale of growth. For *E. coli*, for example, measured degradation rates $k_{deg}$ remain below 0.02 hr⁻¹, substantially lower than observed growth rates (**Pine, 1973**). However, the timescales become comparable when growth is slow and an explicit consideration of protein degradation is needed. Furthermore, protein degradation rates appear to increase at very slow growth ($k_{degr} \approx 0.03$ hr⁻¹ for *E. coli*). For a more detailed discussion of peptide degradation and its possible affect on the ribosome content at slow growth we refer to a recent study from **Calabrese et al., 2021**.

## Observations not in steady state

Another aspect which might explain the derivation between available data and predictions is related to the specifics of the experimental culturing protocols: slow-growing cultures may take such a long time to reach steady state that the experimental measurements may not reflect the steady-state physiology, making an assessment of the model accuracy difficult in this regime. The experimental studies which we use to test our model (listed in **Supplementary file 2**) provide a description of the culturing conditions used, though the specifics of the procedure (such as culturing lengths) are given in broad terms. Most studies relied on a procedure more or less as follows:

1. A seeding culture is grown to exponential phase in a rich growth medium (such as LB, $\lambda \approx 2$ hr⁻¹).
2. The seeding culture is diluted into the experimental medium (termed the *preculture*) which is allowed to grow 'overnight' (which we take to be between 12 and 18 hr) to mid-exponential phase.
3. The preculture is then modestly diluted into the experimental medium and allowed to grow for one or two doublings ($\approx 1.5$ generations) before measurements are made.

While this is a robust protocol to ensure that a fast-growing cultures reach steady state, step 2 requires particular care when slow growth conditions are explored. As the seed culture is typically grown in rich media, the inoculum of cells will have a significantly large allocation towards ribosomes, such as $\phi_{Rb} \approx 0.2$ for a culture grown in LB. As the degradation of ribosomes is slow, this 'bolus of ribosomes' only decreases via dilution as the culture grows. For example, a poor growth medium which can support a growth rate of $\lambda \approx 0.2$ would require more than 20 hr to reach steady state within typical measurement errors of 0.1% (**Appendix 1—figure 2A and B**). Notably, this calculation reflects a best-case scenario where there is no growth arrest upon transfer from from the seeding culture to the preculture medium. In reality, long phases of growth arrest with lag-times on the order of hours is common (**Madar et al., 2013**). In conclusion, precultures have to maintained for a very long time, of the order of a day, when aiming for a steady-state culture in very poor conditions, substantially longer than what the overnight cultures commonly mentioned in the protocols would support.

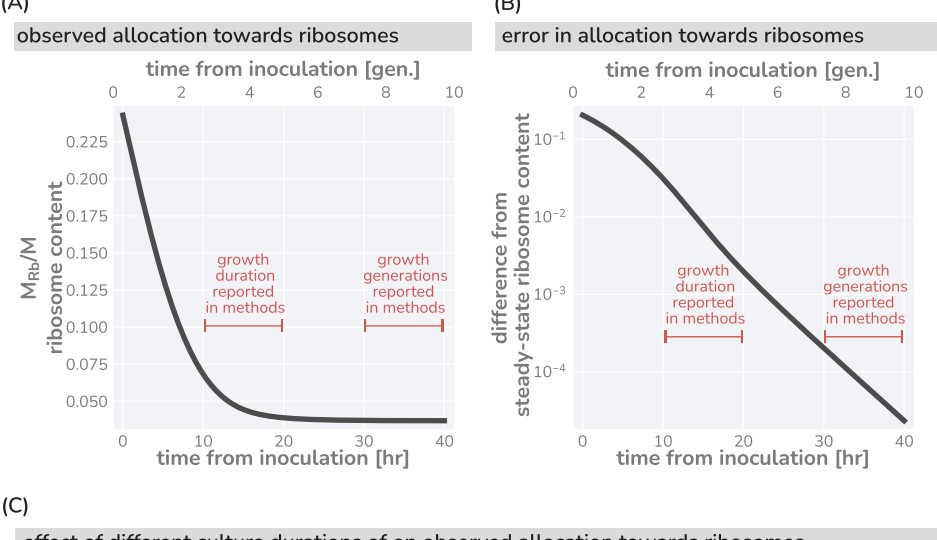

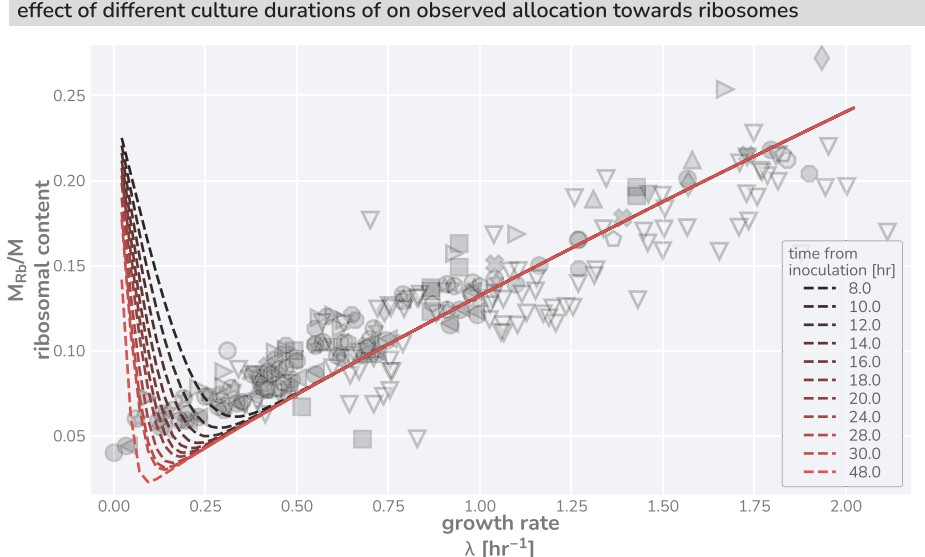

**Appendix 1—figure 2.** Effects of culturing time on observed ribosomal content at slow growth. (**A**) Predicted dynamics of equilibration to steady-state from a seeding culture with high ribosome content ($\phi_{Rb} \approx 0.25$, $\lambda \approx 2.0$ hr$^{-1}$, $\nu_{max} \approx 11$ hr$^{-1}$) into a poor growth medium with a low steady-state ribosome content ($\phi_{Rb} \approx 0.03$, $\lambda \approx 0.2$ hr$^{-1}$, $\nu_{max} \approx 0.03$ hr$^{-1}$). (**B**) The difference between the measurable ribosomal content and the steady-state value in the poor growth medium. Red brackets in (**A**) and (**B**) correspond to claimed culturing duration commonly seen in the literature. Culturing time of 'overnight' is taken to approximately mean 12 hr. (**C**) The effect of insufficient culturing time on the measurable ribosome content. Red dashed lines show model predictions assuming a seeding culture with same parameters as in (**A**) and (**B**) into different media with a range of metabolic rates. Dark to light colors correspond to short (8 hr) and long (48 hr) culturing conditions, respectively. Data and markers are the same as those shown in *Figure 1*.

To see how a too short preculture time could alter the predictions of our model we numerically integrated the flux-parity model (described in 'Methods') assuming a seed culture grown in LB with a large initial ribosome content (*Appendix 1—figure 2C*). From this analysis, it is plausible that some of the discrepancy between the model predictions and experimental measurements could be explained by harvesting cells before they have reached steady state. More detailed information would be needed (such as precise preculturing duration) to concretely assess the magnitude of this effect in the dataset we have assembled.

## Inactive ribosomes

Several studies have reported that microbes commonly maintain a pool of inactive ribosomes (*Metzl-Raz et al., 2017*; *Dai et al., 2016*; *Li et al., 2018*; *Müller et al., 2021*; *Bremer and Dennis, 2008*). In slow growth conditions, an inactive ribosome pool may promote quicker recovery once conditions improve, a phenomenon sometimes called a 'spare ribosome capacity' (*Li et al., 2018*; *Korem Kohanim et al., 2018*; *Mori et al., 2017*). In our model, we do not consider an inactive pool of ribosomes; rather, we assume that *all* ribosomes are active. However, in reality, it is likely that only 80–90% are active at any instant, as has been reported and assumed by many others (*Forchhammer and Lindahl, 1971*; *Dennis and Bremer, 1974*). In our rendering of the allocation models, this can be accommodated in our description of the total biomass synthesis dynamics $\frac{dM}{dt}$ by adding an active fraction prefactor $f_a$ as

$$\frac{dM}{dt} = f_a M_{Rb} \gamma(\text{tRNA}^c), \tag{A6}$$

which results in a translation of the model prediction curves by a small factor well within the variability of the measurements, as shown in *Appendix 1—figure 3*. Given that our quantitative conclusions are not substantively changed by inclusion of this parameter, we have opted to omit it for notational simplicity. This simplification that all ribosomes are active all the time might thus explain the derivation between the observed ribosomal content in cells and the model predictions when growth is very slow. However, direct evidence which supports the maintenance of an inactive ribosome fraction is sparse and many open questions remain. Here we summarize our perspective.

To thrive in fluctuating conditions, cells have to be capable of dynamically regulating the activity of ribosomes. For example, consider a strong downshift scenario where cells transition abruptly from very fast growth (supported by a high precursor flux) to very slow growth (supported by a much lower precursor flux). During the shift, a substantial fraction of the ribosomes needs to be immediately inactivated to avoid exhaustion of the remaining precursor pool leading to cessation of protein synthesis. Molecular studies have supported this hypothesis as proteins which trigger ribosome inactivation (termed 'hibernation factors') are synthesized relatively quickly to the downshift. *E. coli*, for example, uses ribosome modulation factor (RMF) among others which dimerizes ribosomes forming an inactive 100 s complex (*Prossliner et al., 2018*). Transcriptomic analysis has further show that RMF is heavily expressed during a downshift or during entry into starvation, confirming that cells can quickly change the fraction of active ribosomes. However, while we believe in the important role of ribosome inactivation during downshifts, we also believe that the role of ribosome inactivation during steady-state growth remains much less clear. Given our current experimental knowledge, we challenge the idea that cells actively maintain a large fraction of inactive ribosomes during slow growth to be prepared for a quick growth recovery once growth conditions resume. This view of 'spare capacity' is based on the reported large fraction of inactive ribosomes during growth in poor growth conditions (growth rate $\lambda \leq 0.5$ hr$^{-1}$). We see two problems with the derivation of this picture. First, while anticipatory behavior sounds plausible given the reported high fraction of inactive ribosomes, we should keep in mind that the fraction of inactive ribosomes is not based on direct experimental measurements. Instead, the fraction is commonly estimated by the difference between observed growth and measured translation rates (*Dai et al., 2016*), which assumes that measured translation rates correctly reflect the average translation rate of all active ribosomes. More direct measurements of active fractions are possible in principle, such as by ribosome or polysome profiling, but very hard to perform in practice with quantitative accuracy. Secondly, this view assumes that cultures have reached a steady state when the fraction of inactive ribosomes is estimated. As is discussed in Section 3, the time required for cultures to reach steady state is particularly long for poor growth conditions; for growth rates $\lambda < 0.5$ hr$^{-1}$, the times cultures need to spend in pre-culture states extend substantially beyond the 15–20 hr periods commonly used for overnight precultures. There is thus the possibility that reported inactive ribosomes in slow growth conditions are high because cultures are still adjusting to the encountered growth conditions, rather than being high because cells actively maintain a high fraction of inactive ribosomes during slow growth in steady conditions.

In conclusion, a quite involved combination of aspects might be at play at slow growth and we thus did not expand our model to better cover slow growth observations and explicitly include active protein digestion and inactive ribosomes. Further studies at both the experimental and theoretical level are needed to fully assess the role of inactive ribosomes in steady-state growth.

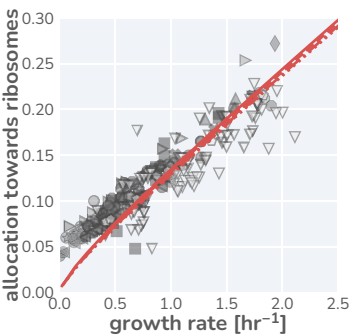 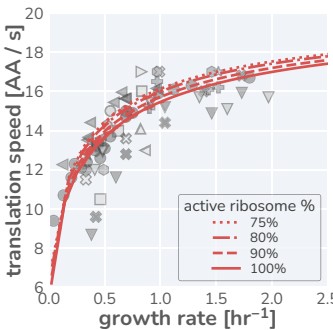

**Appendix 1—figure 3.** Influence of a growth rate independent inactive fraction of ribosomes on model predictions. Data for the ribosomal mass fraction (left) and translation rate (right) observed in steady-state growth are shown using symbols as defined in *Figure 1G*. Red lines indicate the model predictions assuming an active ribosome fraction of 75% (dotted line), 80% (dashed-dotted line), 85% (dashed line), and 100% (solid line).

## What makes the fraction of 'other' proteins?

In the specification of the simple ribosomal allocation model, we asserted that the entire proteome could be categorized into just three sectors: one for metabolic proteins, one for ribosomal proteins, and one for all 'other' proteins. In this section, we demonstrate that the precise value of $\phi_O$, the allocation parameter describing the synthesis of other proteins, is largely independent in predicting growth dynamics and we explore the experimental evidence which establishes and quantify this sector in *E. coli*.

## Defining the other proteins

In this work, we assign one sector of the proteome to be composed of ribosomal proteins. This sector is well defined and specifically contains the $\approx 50$ proteins that make up the 50S and 30S subunits of the ribosome. It is more difficult, however, to determine what proteins are 'metabolic' and which should be classified as 'others.' The past decade has seen a flurry of studies leveraging modern proteomic methods to measure the abundances and relative concentrations of the thousands of protein species which constitute bacterial cells (*Schmidt et al., 2016*; *Peebo et al., 2015*; *Li et al., 2014*; *Valgepea et al., 2013*; *Mori et al., 2021*). *Schmidt et al., 2016*, for example, measured the absolute abundances of 2041 individual proteins in *E. coli* across 22 growth conditions. This data set, coupled with the mountain of functional annotation available for *E. coli* (*Parker et al., 2020*), allows us to explore how different biological processes scale with the steady-state growth rate.

One method to do so is through Clusters of Orthologous Groups (COGs)(*Galperin et al., 2021*), which groups genes by their annotated functions into distinct 'classes' of proteins. *Appendix 1—figure 4A* shows the protein sector mass fraction for all proteins involved in ribosomal structure and biogenesis (gold; COG class J), general metabolism (purple; COG classes P, H, F, E, G, C), and all other processes (black; COG classes X, O, U, W, Z, N, M, T, V, Y, D, B, L, K, A, R, S). Exploring how the mass fractions of these very general annotations scale with growth rate reveals a strong anticorrelation between ribosomal and metabolic genes, with an approximate constant fraction of 'other" proteins. One approach is to rely on this annotation to determine the magnitude of the allocation towards other proteins and take $\phi_O \approx 0.3$.

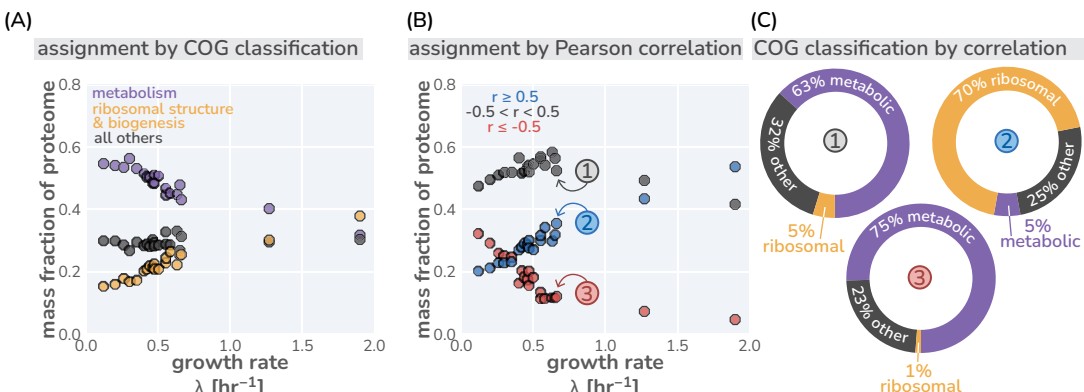

**Appendix 1—figure 4.** Different classification strategies of 'other' proteins. (**A**) Classification of proteins in mass spectrometry data (*Schmidt et al., 2016*) by their COG classification. 'Metabolism' includes all transport and metabolic processes (COG letters: P,H,F,E,G,C *Maitra and Dill, 2015*). (**B**) Classification of 'other' proteins by their growth rate dependence. Other proteins are defined as those with a Pearson correlation coefficient between –0.5 and +0.5. (**C**) The composition of each correlation-defined sector for one condition are shown as doughnut plots. Colors correspond to the COG classifications shown in (**A**) and circled numbers correspond to labeled points in (**B**).

An alternative approach is to determine how the abundance of each individual protein changes with the steady-state growth rate. For each protein in the data of Schmidt et al., we computed the Pearson correlation coefficient between the proteome mass fraction of each individual protein and the growth rate with Pearson's $r$ values of –1.0 and 1.0 showing perfect anticorrelation and correlation, respectively. With a measure of the correlation, we made the somewhat arbitrary decision that any protein with a Pearson's $r$ between $-0.5 < r < 0.5$ to be classified as 'constant,' having weak or no correlation with the growth rate at all. *Appendix 1—figure 4B* shows the results of this classification. Here, it appears that the constant (black points) sector hovers around a mass fraction of ≈0.5, another candidate value for $\phi_O$. To see if this classification scheme was reasonable, we examined what COG classes were represented in each sector defined by the Pearson correlation. *Appendix 1—figure 4C* shows a representative breakdown of each sector by the same COG classification as used in (A). Here, it becomes clear that metabolic proteins dominate both the 'constant' and 'negatively correlated' classes, whereas the 'positively correlated' sector contains predominantly ribosomal proteins. This illustrates that there exists a sizeable pool of proteins whose relative abundance is largely independent of growth rate, despite their classification as being involved in metabolism.

This exploration highlights a subtle yet important point in the classification of the proteome into sectors. While in the main text we specify proteins as being *involved* in metabolism or protein synthesis, we really mean that they can be classified as having a dependence on the growth rate, whether it be positive or negative. Hui and colleagues (*Hui et al., 2015*) recently explored in great depth how the *E. coli* proteome can be broken into six or seven sectors which have different correlations with the growth rate under different types of limitation. In this work, they arrived at an estimation that approximately one-half of the proteome is growth rate independent ($\phi_O = 0.55$) under the many conditions they examined. This value agrees with the simple growth-correlation classification presented above and we thus taken here $\phi_O = 0.55$ for *E. coli*. However, as we describe in the following section, the predictions made by our model is largely independent on the precise value of this parameter.

## Neglecting the other proteins

In *Equation 7* of the main text, we define the mass dynamics of the protein sectors ($M_{Rb}$, $M_{Mb}$, and $M_O$) as

$$\frac{dM_{Rb}}{M} = \phi_{Rb}\frac{dM}{dt} \; ; \; \frac{dM_{Mb}}{dt} = \phi_{Mb}\frac{dM}{dt} \; ; \; \frac{dM_O}{dt} = \phi_O\frac{dM}{dt}, \tag{A7}$$

where $\phi_{Rb}$, $\phi_{Mb}$, and $\phi_O$ denote allocation parameters for ribosomal, metabolic, and 'other' proteins, respectively. We further introduce the constraint that these three classes make up the entire composition of the proteome, meaning that

$$\phi_O + \phi_{Rb} + \phi_{Mb} = 1. \tag{A8}$$

As the majority of the predictions of this work are dependent on the allocation towards ribosomes, we can alternatively define the metabolic allocation factor given *Equation A8* as

$$\phi_{Mb} = 1 - \phi_{Rb} - \phi_O. \tag{A9}$$

Thus, so long as the values of $\phi_{Rb}$ and $\phi_O$ are known, the value of $\phi_{Mb}$ follows and thus the maximal metabolic output N can be defined as

$$N = \nu_{max}\phi_{Mb} = \nu_{max}\left(1 - \phi_{Rb} - \phi_O\right), \tag{A10}$$

where $\nu_{max}$ is the maximum metabolic rate for that particular condition and composition of the metabolic sector. However, suppose we *didn't* know the value of $\phi_O$ and *only* knew $\phi_{Rb}$. In this case, we could further reduce the dimensionality of the proteome by stating that all of the proteins are either ribosomal or are *not* ribosomal. In this case, the allocation parameters for this scenario become

$$\phi_{Mb}^{\dagger} + \phi_{Rb} = 1 \; ; \; \phi_{Mb}^{\dagger} = 1 - \phi_{Rb}, \tag{A11}$$

where the new metabolic allocation parameter includes an unknown 'other' allocation such that

$$\phi_{Mb}^{\dagger} = \phi_{Mb} + \phi_O. \tag{A12}$$

It then follows that the maximal metabolic output given this allocation $N^{\dagger}$ is calculated as

$$N^{\dagger} = \nu_{max}^{\dagger}\phi_{Mb}^{\dagger} = \nu_{max}^{\dagger}\left(1 - \phi_{Rb}\right). \tag{A13}$$

This structuring implies that the metabolic outputs are different whether one knows $\phi_O$ or not. However, these two scenarios can be made equivalent by a simple rescaling of the metabolic rate $\nu_{max}$. Setting *Equation A10* and *Equation A13* to be equivalent and solving for the metabolic rate $\nu_{max}$ where $\phi_O$ is known yields

$$\nu_{max} = \frac{\nu_{max}^{\dagger}\left(1 - \phi_{Rb}\right)}{\left(1 - \phi_O - \phi_{Rb}\right)}. \tag{A14}$$

Thus, one can achieve quantitatively identical predictions between the scenario where $\phi_O$ is known and that where $\phi_O$ is unknown by a simple rescaling of the metabolic rate, $\nu_{max}$. While this serves as a contrived example, it reveals that our estimation of $\phi_O$ having a constant allocation $\phi_O = 0.55$, as has been inferred from mass spectrometry studies (*Hui et al., 2015*) (see previous paragraph), to be largely inconsequential for the predictions made in this work. However, there are some scenarios where the precise value of $\phi_O$ does become important (such as in the case of excess protein stress).

## Inactive ribosomes are not needed to describe the linear relation between ribosome content and growth rate

In their simplest form, allocation considerations assume cells optimally control ribosome content such that protein synthesis by ribosomes occurs with a fixed translation speed or rate ($\gamma_0$). To have all ribosomes working with a constant rate, the allocation parameter $\phi_{Rb}$ controlling the fraction of ribosomes in the cell must then scale linearly with the growth rate, $\phi_R = \gamma_0\lambda$. This relation does not have an offset: at the extreme limit where metabolic proteins are hardly supporting any growth ($\lambda \to 0$), this linear scaling implies that the ribosome content drops to zero ($\phi_{Rb} \to 0$). However, this notion appears to disagree with experimental observations. When a linear regression is performed on the ribosomal content data (solid line in *Appendix 1—figure 5*) as has been done in other studies (*Scott et al., 2010*), one yields an "offset", $\phi_{Rb}^{(min)}$, yielding a linear relation of

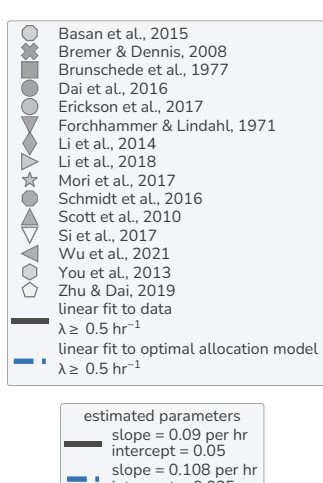
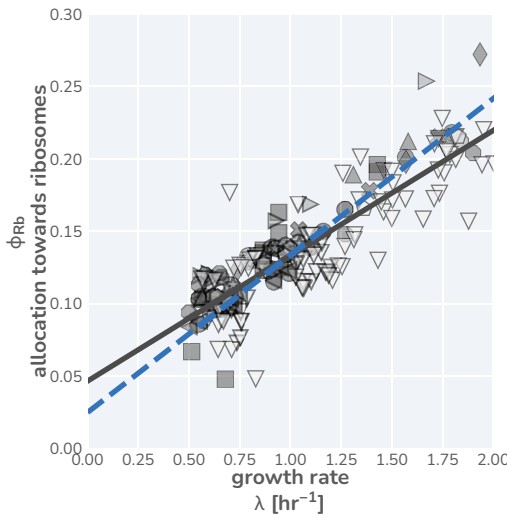

**Appendix 1—figure 5.** Linear regression on data and optimal allocation model in fast-growth regime yields comparable offsets. Data in right-hand panel corresponds to the same measurements used in the main text but restricted to only those data where the growth rate is $\lambda \geq 0.5$ hr$^{-1}$. A simple linear regression (using the SciPy python library) was performed on this data (solid black line) to yield a slope of $\approx 0.1$ hr$^{-1}$ and an intercept of $\phi_{Rb}^{(min)} \approx 0.05$, in line with parameter estimates from previous work (**Li et al., 2014**). Restricting the model predictions to growth rates $\lambda \geq 0.5$ hr$^{-1}$ yields a linear relation (dashed blue line) with a slope of $\approx 0.1$ hr$^{-1}$ and an intercept of $\phi_{Rb}^{(min)} \approx 0.02$.

$$\phi_{Rb}^* = \phi_{Rb}^{(*,min)} + \gamma_0 \lambda. \tag{A15}$$

Previous phenomenological studies have thus rationalized this offset as a growth-rate-independent abundance of inactive ribosomes which are not involved in translation (**Scott et al., 2010**; **Mori et al., 2017**). However, later measurements have confirmed that the translation rate is decidedly *not* constant and in fact increases with the growth rate, asymptotically approaching a maximal value (**Dai et al., 2016**). In **Dai et al., 2016**, the authors use this observation to hypothesize that most of the ribosomes remains active as long as growth rates are not slow ($\lambda \geq 0.5$ hr$^{-1}$). Consistent with this idea, our rendering of the optimal allocation model (scenario III in the main text) explains why this offset emerges from a linear regression without the introduction of any inactive ribosomes.

The strong correlation between the ribosomal content and bulk growth rate has long been hailed as a linear relation; however, there is no a priori rationale behind stating it must be linear. In fact, our optimal allocation model results in a *nonlinear*, yet still monotonically positive, correlation between the ribosomal content and the growth rate. While nonlinear, it is approximately linear in the regime of fast growth, $\lambda \geq 0.5$ hr$^{-1}$. Extending this approximately linear behavior yields a slope and an offset (dashed line in **Appendix 1—figure 5**) which is comparable to the empirically observed offset.

The fundamental reason for this observation is that a close-to-maximal translation rate requires very high precursor concentrations which are very resource demanding to sustain. In our model, this is described by a translation rate which is only met when the precursor concentrations $c_{pc}$ are substantially higher than the Michaelis–Menten constant $K_M^{c_{pc}}$. These dependence can be further explored via the interactive versions of the manuscript figures at our paper website (cremerlab. github.io/flux_parity).

## Application of the model to *Saccharomyces cerevisiae*

In the main text, we evaluated the model predictions by direct comparison with observations for *E. coli*, as appropriate data for this model organism is highly abundant. However, the model predictions should be applicable more broadly to any microbial organism whose growth rate is primarily dependent on the synthesis of protein biomass. The budding yeast *S. cerevisiae* is one such microbe where our approach may be applied and used to quantitatively explore aspects of eukaryotic microbial physiology, and we here provide a parameterization of our model.

We first surveyed the literature to identify and assign a priori values to the major model parameters, which are listed in **Supplementary file 3**. Of particular note is the substantial difference between *E. coli* and *S. cerevisiae* in the proteinaceous mass of a single ribosome ($m_{Rb} = 7459$ AA and $m_{Rb} = 11984$ AA, respectively) and the reported maximum translation speed ($v_{tl,max} \approx 20$ AA/s and $v_{tl,max} \approx 10$ AA/s, respectively) which lead to a substantial difference in the maximum translation rates ($\gamma_{max} = v_{tl,max}/m_{Rb}$). We further note that the fraction of the proteome occupied by the 'other' protein class has not received sufficient characterization in yeast. However, this is not of relevance when comparing model predictions and data during steady-state growth as a variation of $\phi_0$ merely leads to a rescaling of the maximum metabolic rate $\nu_{max}$ which we vary anyway to scan growth rates (see Section 5).

To evaluate the applicability of our model, we further explored the *S. cerevisiae* literature for basic physiological measurements including ribosomal content and translation speeds across growth conditions. To our surprise, we found that these fundamental physiological quantities have been scarcely measured, despite *S. cerevisiae* being a heavily characterized model organism. This is true particularly for the translation speed, for which there are only four or five reported measurements. Nevertheless, we assembled a collated data set from 10 independent studies that we were able to find and appropriately vet and compared their values to the model predictions (**Supplementary file 4**).

Specifically, we evaluated the regulatory scenarios II and III in which the translation rate is either held constant at $\approx 90\%$ of its maximum value or the allocation towards ribosomes is tuned such that growth rate is optimized (**Appendix 1—figure 6**). While the paucity of the data precludes us from making any concrete assessments, it is plausible that *S. cerevisiae* also follows an scheme of optimal regulation of ribosomal allocation (scenario III, blue line). More study is needed, particularly of the growth rate dependence of translation speed, to evaluate the applicability of this approach to yeast.

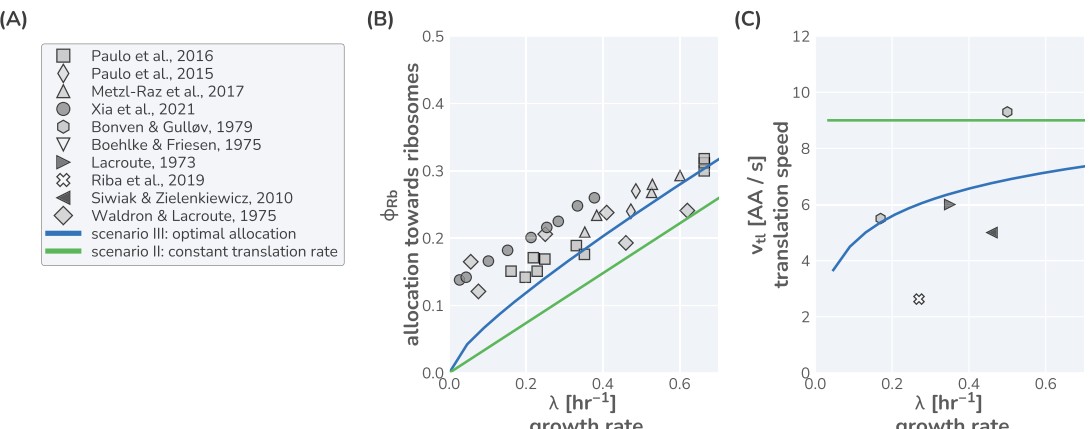

**Appendix 1—figure 6.** Comparison of model predictions to data from *Saccharomyces cerevisiae*. (**A**) List of literature sources reporting measurement of ribosomal content and/or translation speed measurements. Experimental data for (**B**) ribosomal content and (**C**) translation speed are shown as function of growth rate. Green and blue lines correspond to model predictions for scenarios II and III, respectively, using the model parameters defined in **Supplementary file 3**. For scenario II, a constant translation rate of 90% of $v_{tl,max}$ was used.

## Parameter dependence of the flux-parity model

In the main text, we present a solution of the flux-parity model which nearly identically matches the solution for scenario III in which optimal allocation was ensured by hand (**Figure 2**, **Figure 3A-C**). Here, we discuss the parametric sensitivity of this matching and comment on our rationale for choosing specific values.

In 'Methods,' we defined the equations of the flux-parity model. In comparison with the simplistic model where the allocation towards ribosomes is a parameter, we have introduced two Michaelis–Menten parameters ($K_M^{tRNA^u}$ and $K_M^{tRNA^c}$), one ppGpp-specific sensitivity parameter ($\tau$) and a maximal uncharged-tRNA synthesis rate ($\kappa_{max}$). While we can use in vivo and in vitro studies for estimates of these parameters, it is useful to explore how sensitive the model predictions are to precise values.

We first explore how different combinations of parameter values for the Michaelis–Menten constants impact the predictions. We chose to evaluate the steady-state conditions of the flux-parity model for pairwise combinations of a range of $K_M$ values spanning three orders of magnitude from $\approx 10^{-5}$ ($\approx$20 µM) to $\approx 10^{-2}$ ($\approx$20 mM), which covers typical physiological ranges of Michaelis–Menten constants. With the steady-state solutions in hand, we the computed the absolute difference in the steady-state allocation towards ribosomes $\phi_{Rb}$ from that predicted by the optimal allocation (scenario III) of the simple allocation model where allocation parameters are set by hand [Fig 7(A)]. Mathematically, this is defined as

$$\Delta\phi_{Rb} = \left| \phi_{Rb}^{\text{flux-parity}} - \phi_{Rb}^{(III)} \right|. \tag{A16}$$

We found that the precise value of either Michaelis–Menten parameter was less important than their relative values. In fact, we found that a near identical match to the optimal allocation emerged when the parameters were of approximately equal value, $K_M^{tRNA^u} \approx K_M^{tRNA^c}$. This makes sense from a theoretical perspective as both metabolism and translation are feeding into each other's precursor pools. If one $K_M$ was significantly larger than another, the sensitivity of the ppGpp system to the charged- to uncharged-tRNA ratio would also need to be significantly adjusted to accommodate the drastically different kinetics. At particularly large values ($K_M^{tRNA^c} \approx K_M^{tRNA^u} \approx 10^{-3}$), this one-to-one ratio breaks down with an optimal solution emerging when $K_M^{tRNA^u} > K_M^{tRNA^c}$. However, this difference is small and maintaining a $K_M^{tRNA^u} \approx K_M^{tRNA^c}$ deviates from the optimal allocation by $\leq 1\%$.

Like the Michaelis–Menten constants, there is also a strong interdependent relationship between the value of the uncharged-tRNA synthesis rate $\kappa_{max}$ and the ppGpp sensitivity parameter $\tau$, which sets the charged- to uncharged-tRNA ratio at which $\phi_{Rb}$ is half-maximal. We also did a wide pairwise parameter value scan over the range $\tau \in [10^{-5}, 10]$ and $\kappa_{max} \in [10^{-5}, 1]$ hr$^{-1}$, which spans reasonable physiological values [*Appendix 1—figure 7(B)*]. We again see a region of parameter space where the precise values are largely unimportant, so long as the magnitude of $\tau$ is approximately three times larger than $\kappa_{max}$. At particularly low values of $\kappa_{max}$ ($\leq 10^{-4}$ hr$^{-1}$), this dependence again breaks down with $\tau \approx 1.5$ yielding approximately optimal results.

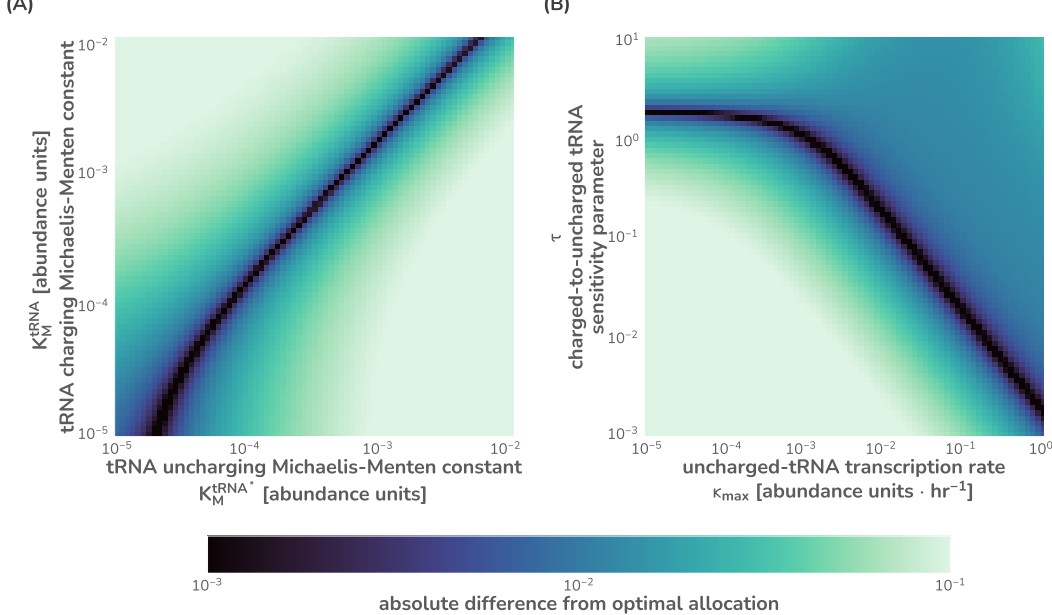

**Appendix 1—figure 7.** Sensitivity analysis of the flux-parity model for key parameters. To demonstrate the sensitivity of the flux-parity model to yield approximately identical predictions to optimal allocation, parameter values spanning several orders of magnitude were compared for the (**A**) Michaelis–Menten constants and (**B**) the tRNA synthesis rate $\kappa$ and ppGpp sensitivity parameter $\tau$. Both (**A**) and (**B**) were evaluated at a single metabolic

*Appendix 1—figure 7 continued*
rate $\nu_{max} = 4.5$ hr⁻¹. The absolute difference from the flux-parity determined allocation and the optimal allocation was computed and is shown on a logarithmic color scale from near-identical (purple) to significantly different (light green). All parameter values except for those being explored were kept the same as listed in **Supplementary file 1**

The inverse relationship between these two parameters also makes sense from a biological perspective. There is only one way by which charged-tRNAs can be synthesized (via metabolism), but two ways in which uncharged-tRNAs can be synthesized (via translation or via transcription). Thus, if the transcription rate of uncharged-tRNAs is very large compared to the synthesis rate by translation, it becomes difficult for the charged- to uncharged-tRNA ratio to become $\geq 1$. To appropriately adjust the allocation towards ribosomes $\phi_{Rb}$, $\tau$ must be at a lower value to remain responsive to changes in the charged- to uncharged-tRNA ratio.

This sensitivity analysis demonstrates a large amount of parametric degeneracy in the flux-parity allocation model. Thus, there is a large parameter space of physiologically feasible values where flux-parity can operate to effect an optimal allocation strategy. This degeneracy suggests that an optimal allocation strategy could more easily evolve once the basic regulation strategies are in place as it does not rely on the simultaneous fine-tuning of every parameter describing the different processes. Given this degeneracy, one can also reduce the dimensionality of the model even further by asserting that the $K_M$'s must be approximately equal,

$$K_M^* = K_M^{\text{tRNA}^u} \approx K_M^{tRNA^c} \tag{A17}$$

and that the magnitude of $\kappa_{max}$ must be one-third that of the sensitivity parameter $\tau$,

$$\kappa_{max} \approx \frac{\tau}{3}\frac{1}{\text{hr}}. \tag{A18}$$

While this reduces the flux-parity model to only four critical parameters ($\tau$, $K_M^*$, $\gamma_{max}$, and $\nu_{max}$), we chose to keep all parameters independent and assigned their values as described in **Supplementary file 1**.

## Flux-parity prediction of total tRNA abundance

A centerpiece of the flux-parity ribosomal allocation model is the separation of the precursor pool into species of charged- and uncharged-tRNA, the concentration ratio of which defines the ribosomal allocation as well as the total tRNA content (tRNA$^c$+tRNA$^u$). To further test the veracity of the flux-parity model, we can compare the predicted steady-state concentrations of total tRNA to those reported in the literature. Specifically, the amount of total tRNA relative to the total number of ribosomes, the results of which are shown in **Appendix 1—figure 8**. We believe that the predicted abundance of tRNA relative to the predicted ribosome content modestly agrees with quantitative measurements, but not to the level of accuracy found in all other comparisons in this work.

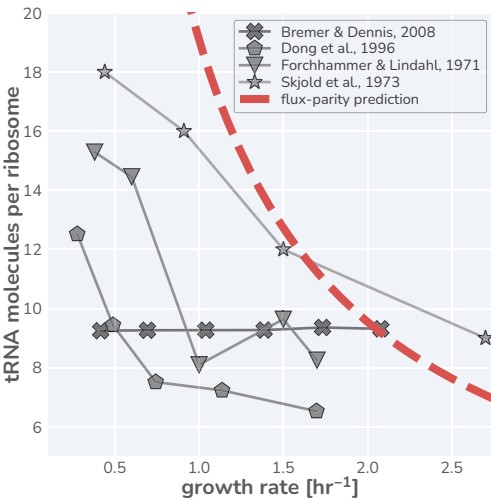

**Appendix 1—figure 8.** Literature measurements for the number of tRNA molecules per ribosome in steady-state growth. Glyphs are shown with connecting lines to more clearly demonstrate the observed trend in each dataset. Red dashed-line represents the estimated tRNA abundance per ribosome resulting from the flux-parity allocation model.

It is important to note, however, that there is a large amount of uncertainty that is present in these types of measurements (quantitative limitations discussed in *Reue, 1998*). Furthermore, there remains quantitative disagreement between studies that directly measure total tRNA content (*Skjold et al., 1973*; *Dong et al., 1996*; *Forchhammer and Lindahl, 1971*) with the calculated abundance (*Bremer and Dennis, 2008*). In the latter, the tRNA per ribosome ratio was calculated following a number of assumptions, namely, a growth-rate independent factor of total RNA that is tRNA (see table 1, $f_s, f_t$, and table 2, note $q$ in *Bremer and Dennis, 2008*). As the majority of stable RNA is rRNA, the assumption of this constant factor a priori enforces a growth rate-independent value for the total number of tRNA per ribosome. Due to the disagreement between these reported values, the large degree of measurement uncertainty, and the approximations needed to convert our tRNA-concentration and total ribosomal mass based model, we emphasize that further quantitative measurements are needed to accurately assess these model predictions.

## Estimating the number of ribosomes within the cell

In the main text, we make the assertion that the accumulation of biomass is dependent on two factors: (i) the number of ribosomes present in the cell and (ii) the speed at which they make peptide bonds. Here we clarify how we estimate the number of ribosomes in the cell.

Ribosomal assembly is an impressively complex process in which ≈ 50 individual proteins and three large rRNAs self-assemble into two major subunits with high efficiency (*Reuveni et al., 2017*). In this work, we thus assume that assembly is instantaneous with the total number of ribosomes given by the total mass of ribosomal proteins, $M_{Rb}/m_{Rb}$, where $m_{Rb}$ is the proteinaceous mass of a single ribosome. In reality, ribosomes can only begin translation once they are assembled. Therefore, a proper accounting of the mass of *functional* ribosomes is

$$N_{Rb} = \left\lfloor \frac{M_{Rb}}{m_{Rb}} \right\rfloor \leq \frac{M_{Rb}}{m_{Rb}}, \tag{A19}$$

where the brackets $\lfloor \ldots \rfloor$ denote the floor function (i.e., rounding down to the nearest integer). Given number of ribosomes per cell is typically large (between ≈5000 and ≈20, 000 depending on the condition *Belliveau et al., 2021*), the fraction of incomplete ribosomal mass is comparatively small, allowing us to make the approximation

$$\left\lfloor \frac{M_{Rb}}{m_{Rb}} \right\rfloor \approx \frac{M_{Rb}}{m_{Rb}}. \tag{A20}$$

If cells could consist exclusively of ribosomes (meaning, $M_{Rb} = M$) which are translating at their maximal rate, $\gamma_{max} = v_{tl,max}/m_{Rb}$, the total biomass dynamics would become

$$\frac{dM}{dt} = \frac{dM_{Rb}}{dt} = \gamma_{max}M_{Rb},$$ (A21)

which can be solved as an exponential relation with a doubling time of

$$t_{double} = \log 2 \frac{1}{\gamma_{max}} = \log 2 \frac{m_{Rb}}{v_{tl,max}}.$$ (A22)

Notably, this approximation only holds as ribosomes consist of many short ribosomal proteins (each ≈200–500 amino acids in size) which are quickly translated. With many different ribosomes translating, all ribosomal proteins required to form a novel ribosome can be translated very quickly (*Reuveni et al., 2017*). Conversely, if the proteinaceous components of ribosomes was a *single* protein with ≈7500 amino acids, then the shortest doubling time would instead be the time it takes to translate a protein with mass $m_{Rb}$ (i.e., $t_{double} = m_{Rb}/v_{tl,max}$), and a description with a simple rate equation (*Equation A19*) would substantially overestimate protein synthesis. This is again emphasizing that complex cellular processes need to be in place for a simple rate equation formulation to work.

## Ribosomes making ribosomes: A consideration of rRNA synthesis

The allocation model presented in the main text considers exclusively protein synthesis as the determinant of microbial growth. Yet, as the cell contains a substantial mass of RNA, microbes clearly must allocate some fraction of their ribosomes towards the synthesis of RNA polymerases (RNAP) such that the required RNA species (rRNA, mRNA, and $tRNA^u$) can accumulate to the appropriate levels. This question was analyzed in more detail by *Roy et al., 2021*. The modeling frameowkr presented by the authors allows a more detailed investigation of RNA and protein synthesis and how these two auto-catalytic cycles are couples. However, in order to maintain a simple, low-dimensional modeling framework we decided to not follow such a more detailed approach. This is movitated by two observations. First, recent order-of-magnitude work has shown that the abundance of RNAP (and the corresponding $\sigma$-factors) is not limiting for growth of *E. coli* across many conditions (*Belliveau et al., 2021*). Second, we here additionally show that RNA synthesis is not associated with a huge protein cost and RNAP synthesis (and correspondingly, RNA synthesis) has thus only a minor effect on the growth rate in nutrient replete conditions. We stress that the situation can be different in nutrient deplete conditions, for example, when phosphate to support RNA synthesis is limiting growth. For such scenarios, a more explicit modeling of RNA synthesis such as the approach introduced by Roy et al. is needed.

Ribosomal RNA (rRNA) accounts for the vast majority of RNA in the cell (≈85%; BNID: 106421 *Milo et al., 2010*) and we therefore only consider the synthesis of rRNA to estimate the demand for RNAP. Ribosomes consist of three large rRNA species which together account for a large fraction of the ribosomal mass and are responsible for the catalysis of peptide bonds. It thus follows that rRNA accounts for a large fraction of the cellular dry mass. One may therefore expect rRNA synthesis to be an important determinant of the time it takes to replicate a ribosome with a strong consequence on growth. However, a comparison of transcription and translation speeds shows that the synthesis of rRNA is far more rapid. *E. coli*, for example, harbors three rRNAs species per ribosome (5S, 16S, and 23S) with a sum total length of 4566 nucleotides (nt). With a transcription speed of ≈ 40 nt/s a single RNAP needs only ≈115 s to synthesize these rRNAs. Given that an RNAP contains ≈4100 AA (significantly less than a ribosome, $m_{Rb} = 7459$ AA), the synthesis of required RNAP does not require a large pool of resources compared to what is required to synthesize the ribosomal proteins. In the following, we extend this logic and calculate the required allocation of ribosomes towards the synthesis of ribosomal proteins and rRNA synthesizing RNAP.

To most clearly introduce the logic of the calculations, we present here only a hypothetical scenario in which precursor supply is unlimited and cells do not have to synthesize metabolic proteins but only consist of ribosomes, rRNA, and the RNAP required to synthesize rRNA. However, similar calculations can be performed when considering the full allocation model and the metabolic proteins required to supply precursors. The mass of (ribosomal) proteins $M_{Rb}$ is proportional to the total number of ribosomes and depends on their elongation rate $\gamma_{max}$, which we assume in this hypothetical scenario to be always maximal. As we are only considering a cell with ribosomal and RNAP proteins, we can state that a certain fraction of the ribosomes $\phi_{Rb}$ are synthesizing ribosomal proteins whereas the rest $1 - \phi_{Rb}$ are generating RNAP. Mathematically, we can enumerate these dynamics as

$$\frac{dM_{Rb}}{dt} = \phi_{Rb}\gamma_{max}M_{Rb}, \tag{A23}$$

for the ribosomal protein biomass dynamics and

$$\frac{dM_{Po}}{dt} = (1 - \phi_{Rb})\gamma_{max}M_{Rb}, \tag{A24}$$

for RNAP protein biomass dynamics, where $M_{Po}$ is the total mass of all RNAP.

We consider that all RNAP are synthesizing rRNA to support ribosomal biogenesis, with the amount of rRNA nucleotides depending on the number of RNAP ($N_{Po}$) and the speed of transcription ($v_{tr}$),

$$\frac{drRNA}{dt} = v_{tr}N_{Po} \equiv \kappa_{tr}M_{Po}, \tag{A25}$$

where we have defined $\kappa_{tr} \equiv \frac{v_{tr}}{m_{Po}}$ with $m_{Po}$ being the mass of a single RNAP. As ribosomes can only work when a sufficient amount of rRNA is present, we next consider the number of rRNA nucleotides per ribosomal amino acids, $r_{nt} = rRNA/M_{Rb}$. The dynamics is then given by

$$\frac{dr_{nt}}{dt} = \kappa_{tr}\frac{M_{Po}}{M_{Rb}} - r_{nt}\gamma_{max}\phi_{Rb}. \tag{A26}$$

In steady-state growth ($dr_{nt}/dt = 0$ and $\frac{M_{Po}}{M_{Rb}} = \frac{1-\phi_{Rb}^*}{\phi_{Rb}^*}$), one obtains a quadratic equation for the fraction $\phi_{Rb}^*$ with the solution:

$$\phi_{Rb}^* = \frac{\kappa_{tr}}{(2r_{nt}\gamma_{max})}\left(-1 \pm \sqrt{1 + \frac{4r_{nt}\gamma_{max}}{\kappa_{tr}}}\right) \tag{A27}$$

For a ribosome to function, rRNA nucleotides and amino acids need to be present at a specific ratio, $r_{nt} = rRNA/m_{Rb}$. Taking this ratio and the known rates of transcription and translation, we can estimate the fraction $\phi_{Rb}$ for *E. coli* yielding $\phi_{Rb} \approx 0.90$. This indicates that only $\approx 10\%$ of the total ribosome pool are needed for RNAP synthesis. It then follows that the upper bound of the growth rate considering the requirements of rRNA synthesis, $\lambda = \gamma(\phi_{Rb,tot} - \phi_{Rb\rightarrow RNAP})$, is different from the exclusively proteinaceous growth rate $\gamma\phi_{Rb}$ by only 10%.

