## [Editor Report]

This valuable study provides a synthesis of sector models for cellular resource partitioning in microbes and shows how a simple flux balance model can quantitatively explain growth phenomena from numerous published experimental data sets. The evidence is convincing, and the study should be of interest to the microbial physiology community.

---

## [Decision Letter]

**Decision letter after peer review:**

Thank you for submitting your article "An Optimal Regulation of Fluxes Dictates Microbial Growth In and Out of Steady-State" for consideration by *eLife*. Your article has been reviewed by 2 peer reviewers, and the evaluation has been overseen by a Reviewing Editor and Aleksandra Walczak as the Senior Editor. The reviewers have opted to remain anonymous.

Essential revisions:

1) Please streamline the presentation of the model and the underlying assumptions, and clearly mention parameter values, and whether they are adjusted or fixed, in the main text. It would also be useful to provide units and typical values in *E. coli* when parameters are introduced, as well as to comment on the realism of parameter values for those that are varied. All this is important, especially for the reader to gain intuition on the model (see reviewer 3's detailed comments).

2) Please briefly discuss additional mechanisms that could be of interest, such as ppGpp binding RNAP and ternary complexes (see reviewer 3's detailed comments).

3) Please address the reviewers' suggestions, esp. regarding clarifying the manuscript, and comparing to previous results in the literature.

*Reviewer #1 (Recommendations for the authors):*

– I find the notations for the allocation parameters (ϕ in the text) misleading. It seems to me that this symbol is often reserved in the literature for mass fractions. Although these two quantities are the same at the steady state (in the framework presented), the authors should at least write a sentence to make the reader aware of that.

– On page 5 the authors discuss the presence of *γ_max_* in scenario I. I might miss something, but the authors could also comment somewhere about the presence of a *γ_min_* (slow growth seems not to be accessible in this regime).

– For the sake of clarity, it could be better to write the equations in Figure 1B in terms of the allocation parameters (not mass fraction).

– Page 6, 'reate' > 'rate'.

– Beginning of the section "Optimal allocation results from…". Here it should be clearly stated that now the authors explore a different model, in which *v* is different from *ν_max_* (as before).

– In Figure 2B, ribosome allocation and transcription rate could be highlighted as they 'constitute' the flux-parity regulator (if my interpretation is correct). They could have a red box.

– In my opinion, the comparison with scenario III on page 8 after Equation (2) is misleading. I would probably put Fig2D-E as a supplementary figure and just plot ϕRb∗ (flux parity). Also, I do not understand the colormap of Fig2D and the fact that the values are multiplied by ϕRb(III) in the plot.– The order of Table S1 and Table S2 should be changed.

*Reviewer #2 (Recommendations for the authors):*

In its current form, the presentation of the model and the underlying assumptions is convoluted. The reader has to do a lot of work to figure out parameter values, which assumptions have been made in which section, and so forth; a revised narrative which is easy to follow would help a lot. Throughout the supplemental material and appendix, references to Figures and Tables are missing (they appear as double question marks) and there are a number of typos; please check those issues.

Some specific comments follow:

– Please explain the proteome constraint ϕRb+ϕMb+ϕO=1 for the benefit of the reader, even though this constraint has been used in prior work.

– Inclusion of ribosome activity later in the paper is confusing after the discussion of how ribosome inactivity is a "puzzle".

– Bottom panel of Appendix 1, Figure 2 – the model behaves very differently near the origin; are there data to confirm this behavior?

– In Figure 1(A), it would help to add units alongside descriptions, e.g. does *c_pc_* have dimensions of charged tRNAs per unit volume, and does *v_max_* have units of nucleotides/sec?

– Please check that notation is consistent, e.g. at bottom of p. 4, ϕR is used instead of ϕRb, and similarly for ϕM.

– Relevant references are missing: In the main text on p. 11 (last line), add Metzl-Raz et al. *eLife* 2017 along with Karpinets et al. 2006; on p. 6 in the discussion regarding *S. cerevisiae* in the penultimate paragraph, add Kostinski and Reuveni Phys. Rev. Res. 2021, and also their 2020 PRL work analyzing the rRNA bound on bacterial growth rates in Methods on p. 15, last paragraph of “Synthesis of Proteins”; in the discussion of a non-zero ribosomal protein fraction in the limit that growth rate goes to zero, include Koch’s hypothesis that this is advantageous for fluctuating growth conditions [A.L. Koch, Adv. Microb. Physiol. 1971].

– Appendix 1, Figure 6: Could a mathematical expression for the 'absolute difference from optimal allocation' be spelled out?

– Proteome sectors in supplemental Figure S1(B) are not to scale: 'other proteins' fraction should be much larger (~55% from authors' numbers) and r-protein fraction (~7% to 25% by mass in *E. coli*) is smaller than represented.

How do the results of this manuscript compare to prior observations of tRNA charging? For example, consider this quote from Bremer and Dennis below:

"There are about nine tRNA molecules per ribosome in exponentially growing *E. coli*, and this ratio shows little variation for growth rates above 0.6 doubling/h. Since the peptide chain elongation rate approaches 22 amino acids per second, each tRNA is required to cycle through the ribosome on average about two times per second. Ikemura [T. Ikemura, 1981, J Mol Biol 146:1-21] has quantitated over 70% of the total tRNA population into 26 separate species, at least one for each amino acid except for proline and cysteine. For each of these 18 different amino acids, there is at least one major tRNA isoacceptor, which is present at a molar ratio of 0.15 to 0.60 copy per ribosome. The aminoacyl-tRNA synthetases are present at about 0.1 copy per ribosome; each synthetase molecule is therefore required to aminoacylate about 10 molecules of its cognate tRNA every second ( = 1 cognate tRNA per second per ribosome) to sustain protein synthesis at a rate of about 20 amino acids per second per ribosome."

---

## [Author Response]

Reviewer #1 (Recommendations for the authors):– I find the notations for the allocation parameters (ϕ in the text) misleading. It seems to me that this symbol is often reserved in the literature for mass fractions. Although these two quantities are the same at the steady state (in the framework presented), the authors should at least write a sentence to make the reader aware of that.

We appreciate the reviewers’ notational concern. In working on the manuscript, we found ourselves frequently confused with notation and its general inconsistency between works. For example, work from the Hwa lab ([1–4]) use *ϕ* to denote mass fractions and *ψ* to denote allocation parameters whereas Giordano et al. [5] uses *α* to denote allocation parameters and capital roman letters to denote mass fractions. Other works, such as that of Korem-Kohanim et al. [6] and Towbin et al. [7] use completely different notational schemes. We chose to use *f* to denote allocation parameters and ratios to denote mass fractions (e.g. *M_Rb_/M* for ribosomal mass fraction) to maintain internally consistent within our own work. Following the reviewers’ suggestion, we have included a more clear explanation of our notation and have adjusted figures to avoid any ambiguity. Furthermore, we have specified in the figures if we are only considering the steady-state regime in which the mass fraction is equivalent to allocation.

– On page 5 the authors discuss the presence of γ_max_ in scenario I. I might miss something, but the authors could also comment somewhere about the presence of a γ_min_ (slow growth seems not to be accessible in this regime).

Upper and lower bounds in these various parameters is something we internally discussed extensively. As *γ* represents the rate of translation (with dimensions of inverse time), it is biochemically well motivated to assume that there is a maximum rate at which the reaction can proceed. Similarly, there should exist a minimal translation rate below which translation stops. Simply because concentrations of required amino acids or other precursors are too low to support an efficient translation, or because efficient translation relies in general on a sufficiently high speed. In our model, we consider explicitly the dependence on the precursors concentrations which we describe by a Michaelis-Menten kinetics. Thus, mathematically the minimum rate we assume in our model is *γ_min_* = 0. But it is important commented on this rationale in the main text.

– For the sake of clarity, it could be better to write the equations in Figure 1B in terms of the allocation parameters (not mass fraction).

While presenting the equations in terms of the allocation parameters *ϕ* is valid for steady growth and furthermore leads to less cluttered equations, it does not hold true beyond this regime. As we want to introduce the model in its most general form (which includes out of steady state dynamics) we state here the equations using the mass fractions.

– Page 6, 'reate' > 'rate'.

We corrected this and other typos, and thank the reviewer again for their close read of the manuscript.

– Beginning of the section "Optimal allocation results from…". Here it should be clearly stated that now the authors explore a different model, in which v is different from v_max_ (as before).

We’ve modified the first two paragraphs of this section to clarify that we are now assuming that nutrients in the environment are saturating and that the value of the metabolic rate *n* depends solely on the uncharged-tRNA concentration tRNAu.

– In Figure 2B, ribosome allocation and transcription rate could be highlighted as they 'constitute' the flux-parity regulator (if my interpretation is correct). They could have a red box.

We appreciate the close read given by the reviewer and have made these changes as suggested.

– In my opinion, the comparison with scenario III on page 8 after Equation (2) is misleading. I would probably put Fig2D-E as a supplementary figure and just plot ϕRb∗ (flux parity). Also, I do not understand the colormap of Fig2D and the fact that the values are multiplied by ϕRb(III) in the plot.

We apologize to the reviewer that these plots were hard to interpret as presented and muddled the point that we were trying to make. Our intent was to do a direct comparison between the “optimal” allocation (scenario III in Figure 1) resulting from analytical maximization of the growth rate to the allocation that emerges from flux-parity. In essence, we believe that showing one can achieve equivalent results between these two models is important to illustrate that growth rate maximization can naturally emerge from the flux-parity model. To hopefully make this more clear, we have modified Figure 2E of the revised text to show the numerical difference between the allocation resulting from flux-parity and the optimal solution. We have retained a plot of this quantity as a function of the metabolic rate to emphasize that the flux-parity regulatory circuit is capable of quantitatively matching the optimal scenario across effectively all physiologically relevant conditions. We have also removed what was Figure 2E in the submitted manuscript which is now shown (in improved form) in the appendix (Figure A6).

– The order of Table S1 and Table S2 should be changed.

We appreciate the reviewer catching that these tables were shuffled due to a LaTeX error. The order has now been fixed.

Reviewer #2 (Recommendations for the authors):In its current form, the presentation of the model and the underlying assumptions is convoluted. The reader has to do a lot of work to figure out parameter values, which assumptions have been made in which section, and so forth; a revised narrative which is easy to follow would help a lot. Throughout the supplemental material and appendix, references to Figures and Tables are missing (they appear as double question marks) and there are a number of typos; please check those issues.Some specific comments follow:– Please explain the proteome constraint ϕRb+ϕMb+ϕO=1 for the benefit of the reader, even though this constraint has been used in prior work.

We agree that it is very useful to better introduce this constraint in the text. We have particularly included a textual explanation that this constraint enforces that all new protein synthesis is accounted for and all proteins can be classified into one of these three sectors.

– Inclusion of ribosome activity later in the paper is confusing after the discussion of how ribosome inactivity is a "puzzle".

Following also the reviewer’s previous remarks on ribosome inactivity being a puzzle (which we do not state anymore in the revised manuscript), we have revisited this section of the manuscript and have clarified it at several points that we hope relieves the confusion.

– Bottom panel of Appendix 1, Figure 2 – the model behaves very differently near the origin; are there data to confirm this behavior?

The model shown in the bottom panel of Figure 1 demonstrates the potential overestimation of steady-state ribosome content arising from insufficient culturing time to reach a true steady state. The hockey-stick shaped response results from the reduction in ribosome content from a seeding culture (grown in LB with ϕRb≈0.2) solely through dilution when transferred into a minimal medium. We are not aware of any extant measurements of ribosome content as a function of time from inoculation from a seeding culture, though we plan to perform experiments in our group.

– In Figure 1(A), it would help to add units alongside descriptions, e.g. does c_pc_ have dimensions of charged tRNAs per unit volume, and does v_max_ have units of nucleotides/sec?

Inline with the reviewers comments and the editors suggested revisions, we have redesigned Figures 1 and 2 of the main text to include tables which summarize the major parameters of each model along with their physical dimensions, approximate values in *E. coli*, and the appropriate references. We hope that this will help clarify to the reviewer’s and the reader the units of all model variables and parameters. In response to the reviewer, our rendering of the metabolic rate *n* has dimensions of inverse time with the range of 0.01 – 20 hr^-1^ used to cover conditions from nutrient poor to nutrient rich conditions. The precursor concentration *c_pc_* has units of concentration, however we convert this concentration to a relative mass abundance, an approximation explain in the Methods section “Approximation concentration via relative abundance.”

– Please check that notation is consistent, e.g. at bottom of p. 4, ϕR is used instead of ϕRb, and similarly for ϕM.

We appreciate the close read the reviewer gave to the manuscript and we have fixed these typos.

– Relevant references are missing: In the main text on p. 11 (last line), add Metzl-Raz et al. eLife 2017 along with Karpinets et al. 2006; on p. 6 in the discussion regarding *S. cerevisiae* in the penultimate paragraph, add Kostinski and Reuveni Phys. Rev. Res. 2021, and also their 2020 PRL work analyzing the rRNA bound on bacterial growth rates in Methods on p. 15, last paragraph of "Synthesis of Proteins"; in the discussion of a non-zero ribosomal protein fraction in the limit that growth rate goes to zero, include Koch's hypothesis that this is advantageous for fluctuating growth conditions [A.L. Koch, Adv. Microb. Physiol. 1971].

We appreciate the reviewer’s suggestion to include these references. We have included all of them at appropriate locations.

– Appendix 1, Figure 6: Could a mathematical expression for the 'absolute difference from optimal allocation' be spelled out?

We agree with the reviewer that this should be included and have added a new equation (Equation A15) explaining how this quantity is calculated.

– Proteome sectors in supplemental Figure S1(B) are not to scale: 'other proteins' fraction should be much larger (~55% from authors' numbers) and r-protein fraction (~7% to 25% by mass in *E. coli*) is smaller than represented.

We appreciate the reviewer’s attention to detail and have modified the figure to be more representative of the values used in the work. We chose the original representation to be qualitatively pedagogical for how a proteome could be broken down but agree that an illustration in better alignment with observations is better.

How do the results of this manuscript compare to prior observations of tRNA charging? For example, consider this quote from Bremer and Dennis below:"There are about nine tRNA molecules per ribosome in exponentially growing *E. coli*, and this ratio shows little variation for growth rates above 0.6 doubling/h. Since the peptide chain elongation rate approaches 22 amino acids per second, each tRNA is required to cycle through the ribosome on average about two times per second. Ikemura [T. Ikemura, 1981, J Mol Biol 146:1-21] has quantitated over 70% of the total tRNA population into 26 separate species, at least one for each amino acid except for proline and cysteine. For each of these 18 different amino acids, there is at least one major tRNA isoacceptor, which is present at a molar ratio of 0.15 to 0.60 copy per ribosome. The aminoacyl-tRNA synthetases are present at about 0.1 copy per ribosome; each synthetase molecule is therefore required to aminoacylate about 10 molecules of its cognate tRNA every second ( = 1 cognate tRNA per second per ribosome) to sustain protein synthesis at a rate of about 20 amino acids per second per ribosome."

We appreciate the reference to this quote from Bremer Dennis, a work we reference routinely ourselves. In developing and evaluating the flux-parity model, we originally sought to see if our rendering of the flux-parity allocation model could quantitatively agree with literature measurements of tRNA abundance, the results of which are shown in Appendix 1—figure 8. We believe that the predicted abundance of tRNA relative to the predicted ribosome content modestly agreed with quantitative measurements, but not to the level of accuracy found in the rest of the manuscript. When digging into the data, however, we became aware of the large amount of uncertainty that is present in these types of measurements (quantitative limitations discussed in Ref. [24]) paired with uncertainty in the absolute abundance of the number of ribosomes per cell. Furthermore, we were concerned with the profoundly different behaviors between direct measurements of this quantity (Forchhammer and Lindahl 1971, Skjold et al. 1973, and Dong et al. 1996) with the calculated abundance as reported in the Bremer and Dennis work. In the latter, the tRNA per ribosome ratio was calculated following a number of assumptions, namely, a growth-rate independent factor of total RNA that is tRNA (See table 1, *f_s_*, *f_t_*, and table 2, note *q* in [10]). As the majority of stable RNA is rRNA, the assumption of this constant factor a priori enforces a growth-rate independent value for the total number of tRNA per ribosome. Due to the disagreement between these reported values, the large degree of measurement uncertainty, and the approximations needed to convert our tRNA-concentration and total ribosomal mass based model, we erred on the side of caution and did not include this analysis into the final manuscript. We have now included this analysis in the revised appendix as Section 9.

These points aside, our work remains in agreement with the quote mentioned by the reviewer when one considers the regime where growth is rapid. In this limit, one can assume that the concentration of tRNAs is sufficient such that the translation speed is approximately maximal, ≈20 AA / second. If we assume that ≈10 tRNA are present per ribosome with 50x10^4^ ribosomes per cell, one can estimate the charged tRNA mass abundance to be

tRNA∗≈50×103ribosomescell×1AAtRNA×1cell109AA≈5×10−3charged−tRNA massprotein\ biomass Numerically solving for the rapid growth regime in the flux-parity allocation model yields a steady-state charged-tRNA mass abundance of ≈ 2 x 10^-3^, within a factor of a few from the order of magnitude estimate, and placing us in agreement with the calculation performed by Dennis and Bremer.

References